



# A New-Generation Internal Tide Model Based on 30 Years of Satellite Sea Surface Height Measurements

Zhongxiang Zhao [1, 2]

[1]Applied Physics Laboratory, University of Washington, Seattle, WA, USA
[2]School of Oceanography, University of Washington, Seattle, WA, USA

**Correspondence:** Zhongxiang Zhao  (zzhao@uw.edu)

**Abstract.** An internal tide model ZHAO30yr is developed using 30 years of satellite altimetry sea surface height measurements from 1993 to 2022 by a newly improved mapping technique that consists of two rounds of plane wave analysis with a spatial bandpass filter in between. Prerequisite wavelengths are obtained using climatological annual-mean hydrographic profiles in the World Ocean Atlas 2018. The model contains 12 internal tide constituents: 8 mode-1 constituents ($M_2$, $S_2$, $N_2$, $K_2$, $K_1$, $O_1$, $P_1$, and $Q_1$) and 4 mode-2 constituents ($M_2$, $S_2$, $K_1$, and $O_1$). Model errors are estimated to be lower than 1 mm in amplitude, thanks to the new mapping technique and long data record. The model is evaluated by making internal tide correction to independent altimetry data in 2023. Ten constituents (but for $K_2$ and $Q_1$) can reduce variance on global average; $K_2$ and $Q_1$ can cause variance reductions in their source regions. The model decomposes the multiconstituent multimodal multidirectional internal tide field into a series of plane waves at each grid point. The decomposition reveals unprecedented features previously masked by multiwave interference. The model divides each internal tide constituent into components by propagation direction. The directionally-decomposed components reveal numerous long-range internal tidal beams associated with notable topographic features. The semidiurnal beams off the Amazon shelf and the diurnal beams in the Arabian Sea are examined in detail. ZHAO30yr is available for download at http://doi.org/10.6084/m9.figshare.28078523 (Zhao, 2024).

## 1 Introduction

Internal tides (internal gravity waves at tidal frequencies) are inherent wave motions in the interior of stratified ocean (Wunsch, 1975; Munk and Wunsch, 1998; Garrett and Kunze, 2007). Internal tides are mainly generated by barotropic tidal currents flowing over variable topography (Egbert and Ray, 2000; Smith and Young, 2002; Nycander, 2005). They propagate over hundreds to thousands of km and redistribute the converted tidal energy in the open ocean (Ray and Mitchum, 1996; Alford, 2003; Zhao, 2014; MacKinnon et al., 2017). Internal tides gradually lose their coherence (phase locking) with the barotropic tidal forcing in long-range propagation through the time-varying ocean. Fortunately, a fraction of internal tides remain coherent and thus detectable by multiyear time series from field moorings, acoustics thermometry, and satellite altimetry (Ray and Mitchum, 1996; Alford, 2003; Dushaw, 2022). In recent years, internal tides have drawn great research interest, because they play an important role in various ocean processes including tracer transport, acoustic transmission, coral bleaching, primary productivity, and ocean mixing (Jayne and St. Laurent, 2001; Tuerena et al., 2019; Whalen et al., 2020; Zhang et al., 2021;



Dushaw, 2022; Jacobsen et al., 2023; Guan et al., 2024). In particular, internal tides can be used for monitoring global ocean changes, in that their speed changes in long-range propagation contain important information on ocean stratification (Zhao, 2016). Internal tides may be unwanted noise for some researchers and should be accurately corrected (Morrow et al., 2019; Yadidya et al., 2024). In the past decade, a few empirical internal tide models have been constructed from satellite altimetry (Ray and Zaron, 2016; Zhao et al., 2016; Zaron, 2019; Zhao, 2023b; Zaron and Elipot, 2024). On the other hand, numerical internal tide models have been developed by numerical simulations driven by atmosphere forcing and tidal potential (Simmons et al., 2004; Müller et al., 2012; Shriver et al., 2012; Buijsman et al., 2017; Arbic et al., 2018; Li and von Storch, 2020; Arbic, 2022). In this paper, I will present a new internal tide model developed using 30 years of satellite altimetry sea surface height (SSH) measurements from 1993 to 2022.

Satellite altimetry observes internal tides via their small SSH fluctuations and thus provides a unique tool for mapping internal tides on a global scale. However, their weak SSH signals are usually overwhelmed by leaked mesoscale signals (mesoscale contamination). Previous studies mainly focused on the first baroclinic modes of $M_2$, $S_2$, $K_1$, and $O_1$ constituents (Carrere et al., 2021). Previous internal tide models contain considerable errors, but do not provide an error estimate. The oceanographic community needs accurate and complete internal tide models in various researches such as quantifying coherent and incoherent internal tides, internal tide-induced ocean mixing, and internal tide-eddy interactions. These questions require a better knowledge of the global internal tide field. Previous advances are due mainly to the accumulation of multiyear multimission altimetry data, because the longer data record may lead to lower errors. Some recent altimetry missions (phases) are operated along nonrepeat tracks (Zhao, 2022a). For example, CryoSat-2 has a long repeat period of 369 days and samples the ocean along 10668 ground tracks (Wingham et al., 2006). Haiyang-2A has 386 ground tracks in the exact-repeat phase and 4630 ground tracks in the geodetic phase. The nonrepeat ground tracks greatly improve spatial resolution, because the denser ground tracks allow us to map internal tides in smaller fitting windows (Zhao, 2022a, Figure 2).

We have been improving our mapping technique over the past years to construct better and better internal tide models (Zhao and Alford, 2009; Zhao, 2014; Zhao et al., 2016; Zhao, 2022a). Our core technique is plane wave analysis that resolves waves in different horizontal directions. Our mapping technique has been adapted to nonrepeat altimetry missions. Previous pointwise harmonic analysis cannot extract internal tides from nonrepeat altimetry missions, because the SSH time series at any given point is too short to reliably extract tidal harmonics. The point-wise harmonic analysis in Zhao et al. (2016) has been replaced with plane wave analysis, and thus our new mapping procedure calls plane wave analysis twice. The along-track one-dimensional bandpass filter in Zhao et al. (2016) has been replaced with spatial two-dimensional bandpass filter to avoid removing internal tides having a large angle with ground tracks. The newly improved mapping technique thus consists of two rounds of plane wave analysis with a spatial bandpass filter in between (Zhao, 2022a, b). It filters the internal tide field by three rounds of temporal and spatial filtering, utilizing known tidal periods and wavelengths of the target internal tides.

This paper reports a new internal tide model developed by applying the newly improved mapping technique to 30 years of satellite altimetry data from 1993 to 2022. The new internal tide model is labeled ZHAO30yr. The model decomposes the internal tide field and thus reveals numerous long-range internal tidal beams. In contrast, all previous internal tide models give the multiwave summed internal tide fields and do not resolve internal tidal beams (Carrere et al., 2021). As shown in this paper,





the resolved internal tidal beams contain key information on their generation, propagation, and dissipation. ZHAO30yr has the following outstanding features:

1. The model provides model errors that are estimated by background internal tides. The combination of new mapping technique and long data record reduces model errors down to lower than 1 mm; therefore, we can extract the much weaker minor and mode-2 internal tide constituents.

2. The model contains 12 internal tide constituents: eight mode-1 constituents ($M_2$, $S_2$, $N_2$, $K_2$, $K_1$, $O_1$, $P_1$, and $Q_1$) and four mode-2 constituents ($M_2$, $S_2$, $K_1$, and $O_1$). It contains more constituents than all previous empirical internal tide models published in Carrere et al. (2021, Table 1).

3. The model resolves the 12 internal tide constituents each by 5-wave decomposition. It thus decomposes the global multiconstituent multimodal multidirectional internal tidal field into 60 simple plane waves at each grid point. The decomposition gives many new features that are previously masked by multiwave interference.

4. The model contains directionally decomposed components, which reveal numerous well-defined long-range internal tidal beams associated with notable topographic features. The beams are characterized by large amplitudes, linear-increasing phases, and across-beam co-phase lines.

The remainder of this paper is arranged as follows. Section 2 briefly describes the satellite altimetry data and ocean stratification data used in this study. Section 3 gives a detailed description of mapping methods and key mapping parameters. Section 4 estimates model errors and evaluates the model using independent altimetry data. Section 5 examines the decomposed components and shows numerous internal tidal beams. Section 6 examines in detail the internal tidal beams off the Amazon shelf and in the Arabian Sea. Section 7 is a summary. Appendix **??** shows scaling factors of 4 pairs of internal tide constituents.

## 2 Data

### 2.1 Satellite altimetry data

The internal tide model is developed using 30 years of satellite altimetry along-track SSH measurements from 1993 to 2022 (Figure 1, red box). The data are pre-processed and distributed by the Copernicus Marine Service (https://doi.org/10.48670/moi-00146). The SSH measurements are made by 15 nadir altimetry missions, along both exact-repeat and nonrepeat tracks. The merged data record is about 120 years long, including all altimetry data from 1993 to 2022 distributed by the Copernicus Marine Service. The 30-year long data lead to higher spatial resolution (Zhao, 2022a, b). The data have been pre-processed for standard geophysical corrections including atmospheric effects, surface wave bias, geophysical effects, barotropic tide, polar tide, solid Earth tide and loading tide (Pujol et al., 2016; Taburet et al., 2019). The altimetry data from seven nadir altimetry missions in 2023 are reserved for model evaluation (Figure 1, blue box).

The SSH signals of mesoscale eddies are about one order of magnitude greater than the internal tide signals. Direct mapping internal tides without mesoscale correction would lead to large model errors (Ray and Zaron, 2016; Zhao et al., 2016).

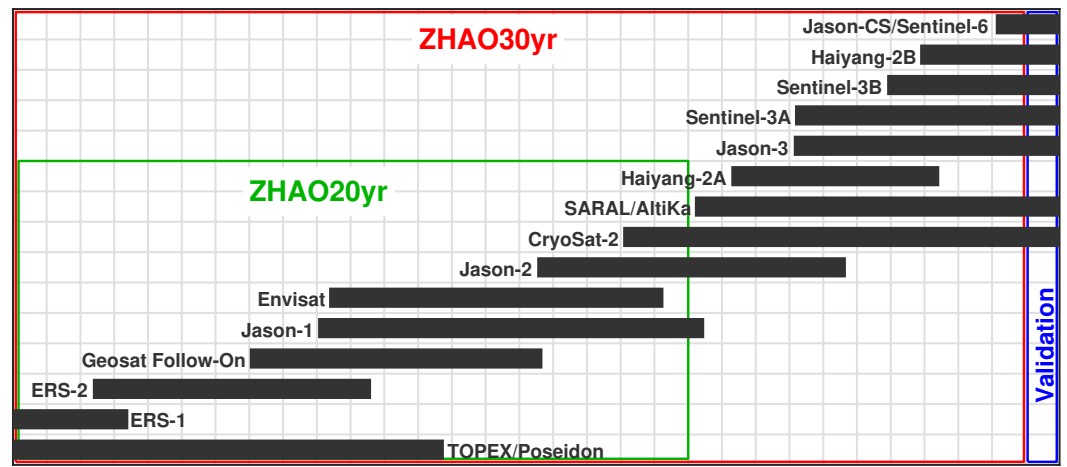

**Figure 1.** Satellite altimetry data. ZHAO20yr and ZHAO30yr are developed using 20 (1993–2012) and 30 (1993–2022) years of altimetry data, respectively. Altimetry data in 2023 are reserved for model evaluation.

Mesoscale correction was brought up by Ray and Byrne (2010) and has been employed in a number of studies (Ray and Zaron, 2016; Zaron, 2019; Zhao, 2022b). In this study, prior mesoscale correction is made using the two-dimensional (2D) gridded SSH fields distributed by the Copernicus Marine Service (https://doi.org/10.48670/moi-00148). The fields are gridded daily in time and $0.25°$ by $0.25°$ in the horizontal. Prior to mesoscale correction, the gridded SSH fields are 2D low-pass filtered to remove internal tide signals (Ray and Zaron, 2016; Zaron, 2017). Cutoff wavelengths of 200 km (300 km) are used for data sets prepared for mapping semidiurnal (diurnal) internal tides. The mesoscale signals are then interpolated and removed from the along-track SSH data (Ray and Byrne, 2010; Ray and Zaron, 2016; Zaron, 2019; Zhao, 2022b). The mesoscale correction cannot perfectly remove mesoscale signals, but it is an indispensable step to suppress mesoscale contamination.

### 2.2 Internal tide wavelengths

Our mapping technique requires tidal periods and wavelengths of the target internal tides. The periods (frequencies) of internal tides are astronomical constants that have been well documented in classic textbooks (e.g., Pugh and Woodworth, 2014) and software packages (e.g., Egbert and Erofeeva, 2002; Pawlowicz et al., 2002). Table 1 gives the tidal periods of eight principal constituents studied in this paper ($M_2$, $S_2$, $K_1$, $O_1$, $N_2$, $K_2$, $P_1$, and $Q_1$). There are two pairs of internal tide constituents that are separated by 2 cycles per year (cpy). One pair contains $K_1$ (23.9345 hours) and $P_1$ (24.0659 hours). The other pair contains $S_2$ (12 hours) and $K_2$ (11.9672 hours). For barotropic tides, at least 6-month hourly data record is needed to separate each pair. It is found in this study that 30 years of altimetry data with irregular sampling rates can separate both constituent pairs.

The wavelengths of internal tides are calculated using the climatological annual-mean hydrographic profiles in the World Ocean Atlas 2018 (WOA18) provided by the NOAA National Centers for Environmental Information (https://www.nodc.noaa.gov/OC5/woa18/). The WOA18 hydrography is at a spatial grid of $0.25°$ by $0.25°$. For a given ocean

**Table 1.** Properties and Empirical Mapping Parameters of the 12 Internal Tide Constituents.

| Constituent | Period (hour) | Wavelength[a] (km) | Window Size (step 1) | Bandpass Width (step 2) | Window Size (step 3) | Final Grid |
|---|---|---|---|---|---|---|
| mode-1 $M_2$ | 12.4206 | 137.3 | 120 km | [0.75, 1.50] | 120 km | 0.05° |
| mode-1 $S_2$ | 12 | 129.5 | 120 km | [0.80, 1.25] | 120 km | 0.05° |
| mode-1 $K_1$ | 23.9345 | 294.1 | 120 km | [0.75, 1.50] | 160 km | 0.05° |
| mode-1 $O_1$ | 25.8193 | 345.6 | 120 km | [0.75, 1.50] | 160 km | 0.05° |
| mode-2 $M_2$ | 12.4206 | 71.9 | 120 km | [0.75, 1.50] | 80 km | 0.05° |
| mode-2 $S_2$ | 12 | 67.9 | 120 km | [0.80, 1.25] | 80 km | 0.05° |
| mode-2 $K_1$ | 23.9345 | 163.8 | 120 km | [0.75, 1.25] | 120 km | 0.05° |
| mode-2 $O_1$ | 25.8193 | 191.2 | 120 km | [0.75, 1.25] | 120 km | 0.05° |
| mode-1 $N_2$ | 12.6583 | 142.1 | 160 km | [0.80, 1.25] | 120 km | 0.05° |
| mode-1 $K_2$ | 11.9672 | 127.9 | 160 km | [0.80, 1.25] | 120 km | 0.05° |
| mode-1 $P_1$ | 24.0659 | 294.3 | 160 km | [0.75, 1.50] | 160 km | 0.05° |
| mode-1 $Q_1$ | 26.8584 | 404.5 | 160 km | [0.75, 1.50] | 160 km | 0.05° |

[a] global mean wavelength within ±60° (semidiurnal) and ±26.5° (diurnal)

depth and stratification profile, the vertical structures and eigenvalue speeds of discrete baroclinic modes are obtained by solving the Sturm-Liouville orthogonal problem (Gill, 1982; Kelly, 2016; Zhao et al., 2016)

$$\frac{d^2\Phi(z)}{dz^2} + \frac{N^2(z)}{c_n^2}\Phi(z) = 0, \tag{1}$$

subject to free-surface and rigid-bottom boundary conditions, where $N(z)$ is buoyancy frequency profile, $\Phi(z)$ and $c_n$ are eigenvector and eigenvalue, and the subscript $n$ is modal number, respectively. With Earth's rotation, wavelength $\lambda_n$ can be calculated from the eigenvalue speed $c_n$ following $\lambda_n = c_n/\sqrt{\omega^2 - f^2}$, where $\omega$ and $f$ ($\equiv 2\Omega\sin(\text{latitude})$, where $\Omega$ is Earth's rotation rate) are the tidal and inertial frequencies, respectively. The resulting global wavelengths for the semidiurnal and diurnal internal tide constituents are shown in Figures S1 and $S_2$ (Supplementary Materials), respectively. It is well known that wavelengths are a function of location, in particular, latitude. Table 1 gives their global mean wavelengths (within ±60° for semidiurnal constituents and ±26.5° for diurnal constituents). For the eight mode-1 constituents, the mean wavelengths range from 129.5 km for $S_2$ to 404.5 km for $Q_1$. The mode-2 $K_1$ and $O_1$ internal tides have mean wavelengths of 163.8 and 191.2 km, respectively, longer than mode-1 semidiurnal constituents. However, the mode-2 $M_2$ and $S_2$ constituents have wavelengths shorter than 80 km. Note that mode-1 $K_1$ and $P_1$ have close wavelengths (294.1 km and 294.3 km). Due to their close tidal periods and wavelengths, 30 years of altimetry data are needed to separate them apart.





## 3 Methods

The newly improved mapping procedure consists of two rounds of plane wave analysis with a spatial bandpass filter in between (Zhao, 2022a, b). An example of the 3-step mapping procedure can be found in Zhao (2022a, Figure 3). In the first step, one target internal tide constituent is mapped by plane wave analysis. At each grid point, five internal tidal waves of arbitrary propagation directions are determined (Section 3.1). The vector sum of these five waves gives the internal tide solution. This step yields a global internal tide field at a regular latitude-longitude grid from the sparse satellite along-track SSH data. In the second step, the regularly gridded internal tide field is further cleaned by spatial bandpass filtering (Section 3.2). The target internal tide field is converted to the 2D wavenumber spectrum by Fourier transform. The spectrum is truncated by bandpass width times the local wave number. Empirical bandpass widths are given in Table 1. In the third step, plane wave analysis is called again to decompose the filtered internal tide field into five internal waves at each grid point. The second-round plane wave analysis is the same as the first-round plane wave analysis, but that the input is the filtered internal tide field in the second step. In the end, the resulting five waves are saved with their respective amplitudes, phases and directions. The 5-wave decomposition makes it possible to separate internal tides of different propagation directions.

### 3.1 Plane wave analysis

The core technique for mapping internal tides from satellite altimetry data is plane wave analysis developed in a series of previous studies (Zhao and Alford, 2009; Zhao, 2014; Zhao et al., 2016; Zhao, 2022a). Examples can be found in Zhao (2014, Figure 3) and Zhao et al. (2016, Figure 2). Plane wave analysis extracts internal tides from SSH measurements in a square fitting window. At each grid point, the internal tide solution is mapped using along-track altimetry data in a fitting window centered at the grid point. Each fitting window contains a large number of independent SSH data. The target internal tidal wave $\eta(A, \phi, \theta)$ has three parameters to be determined: amplitude $A$, phase $\phi$, and propagation direction $\theta$. There are multiple waves of arbitrary propagation directions at one site; therefore, 5 internal tidal waves are fitted following

$$\eta(A, \phi, \theta; x, y, t) = \sum_{m=1}^{5} A_m f(t) \cos(\frac{2\pi}{T} t + \phi_m + u(t) - \frac{2\pi}{\lambda} x \cos\theta_m - \frac{2\pi}{\lambda} y \sin\theta_m), \qquad (2)$$

where $x$ and $y$ are the east and north Cartesian coordinates, $t$ is time, $T$ and $\lambda$ are the period and wavelength of the target wave, and $f(t)$ and $u(t)$ are nodal factor and phase for the 18.6-year cycle, respectively. The lunar nodal cycle is taken into account (Pugh and Woodworth, 2014), because the altimetry data are longer than 18.6 years. Using SSH data in one given fitting window, the amplitude $A$, phase $\phi$, and propagation direction $\theta$ of one target internal tidal wave are determined by least-squares fit. In each compass direction (angular increment is 1°), the amplitude and phase of one single plane wave are determined. When the resultant amplitudes are plotted as a function of direction in polar coordinates, an internal tidal wave appears to be a lobe. The amplitude and direction of the first wave are thus determined from the largest lobe. The signal of the determined wave is predicted and removed from the original SSH data. An iterative algorithm has been developed to extract an arbitrary number of waves (Zhao et al., 2016). Five internal tidal waves are determined in this study. They are sorted





with decreasing amplitudes. In the end, each wave is refitted with other waves temporally removed to reduce the wave-wave interference. The vector sum of the five waves gives the internal tide solution at the grid point.

## 3.2   Spatial bandpass filtering

Using the spatially regular internal tide field obtained by plane wave analysis, the internal tide field can be converted to a 2D wavenumber spectrum by Fourier transform in one overlapping 850 by 850 km window. Different windows have been tested
and found that the filtering is insensitive to the window size. The 2D wavenumber spectrum reveals that the variance is mainly around the theoretical wavenumber. The variance falling outside the theoretical wavenumber range is nontidal errors. Thus, the 2D wavenumber spectrum is truncated and converted back to the internal tide field by inverse Fourier transform. The width of the bandpass filter (e.g., cutoff wave numbers) is empirically determined (Table 1). They reflect the spectral peaks of the target internal tide constituent determined by the length of the data record. The bandpass width is affected by the fitting window
employed in plane wave analysis and noise level. An example of the spatial 2D bandpass filter can be found in Zhao (2019, Figure 4).

## 3.3   Special issues

There are some issues in the model development that require special attention. First, $S_2$ has an tidal aliasing issue with Sun-synchronous altimetry missions (e.g., ERS-1/2 and Haiyang-2A/2B). Previous studies usually map $S_2$ internal tides excluding
Sun-synchronous missions (Zhao, 2018; Zaron, 2019). However, Ubelmann et al. (2022) found that $S_2$ internal tides can be mapped including data from Sun-synchronous missions. The nontidal signals caused by solar radiance have longer spatial scales than internal tides; therefore, the wavenumber-frequency filtering can remove solar radiance and other nontidal signals. Here mode-1 and mode-2 $S_2$ internal tides are mapped using all altimetry missions including Sun-synchronous missions (Figure 1, red box). The results show that the new $S_2$ constituents have higher spatial resolutions and lower errors (Figure 6).
Second, care is needed to separate two pairs of internal tide constituents. The first pair contains mode-1 $K_1$ and $P_1$ with tidal periods of 23.9345 and 24.0659 hours, respectively. The second pair contains mode-1 $S_2$ and $K_2$ with tidal periods of 12 and 11.9672 hours, respectively. To separate $K_1$ and $P_1$, mode-1 $K_1$ internal tides are firstly constructed. Then $P_1$ internal tides are mapped using the temporally $K_1$-corrected altimetry data (e.g., predict and subtract $K_1$ internal tides from the original data). In the end, the mode-1 and mode-2 $K_1$ internal tides are re-mapped using the $P_1$-corrected altimetry data. Likewise, $S_2$ and
$K_2$ can be separated following the same procedure. The results show that this method can successfully suppress cross-talk and separate the two pairs of constituents.

Third, the larger mode-1 constituents may affect the smaller mode-2 constituents. Both mode-1 and mode-2 constituents are mapped for $M_2$, $S_2$, $K_1$, and $O_1$. For each tidal constituent, modes 1 and 2 have the same tidal period, but the mode-1 wavelengths are about twice the mode-2 wavelengths (Table 1). For each case, the mode-1 constituent may affect the mode-2
constituent, but the mode-2 constituent does not affect the mode-1 constituent. Therefore, the mode-1 constituent is firstly mapped and removed from the original data. Then the mode-2 constituent is mapped using the corrected data. This measure leads to clean mode-2 internal tide constituents.



### 3.4 Mapping parameters

We extract 12 internal tide constituents from the 30 years of satellite altimetry data one by one following the same 3-step
mapping procedure. They are eight mode-1 constituents ($M_2$, $S_2$, $N_2$, $K_2$, $K_1$, $O_1$, $P_1$, and $Q_1$) and four mode-2 constituents
($M_2$, $S_2$, $K_1$, and $O_1$). Table 1 lists the 12 internal tide constituents and their key empirical mapping parameters. In this study,
semidiurnal internal tide constituents are mapped from 60° S to 60° N and diurnal constituents from 30° S to 30° N. In the
first round of plane wave analysis, a fitting window of 120 km is used for major constituents and 160 km for minor constituents
(Table 1). In the second round of plane wave analysis, a fitting window of 160 km is used for diurnal constituents, 120 km for
mode-1 semidiurnal constituents, and 80 km for mode-2 $M_2$ and $S_2$ constituents. In the spatial bandpass filtering, [0.75, 1.50]
is used for diurnal constituents and $M_2$ constituents, and [0.80, 1.25] for other semidiurnal constituents. All constituents are
finally at a spatial grid of 0.05° by 0.05°. These mapping parameters are empirically chosen after testing several reasonable
choices. Our previous studies show that these mapping parameters will not affect the results much on a global scale (Zhao,
2022a, b); however, the model can be further optimized region by region and constituent by constituent. Figure 2 shows the
resulting 12 internal tide constituents. Internal tides with amplitudes lower than 1 mm are shown in light blue (model errors).
The regions with large model errors due to mesoscale contamination are indicated by black contours. The results show that the
mode-1 $M_2$ and $K_1$ internal tides have largest amplitudes greater than 25 mm, while the mode-1 $K_2$ and $Q_1$ internal tides have
lowest amplitudes lower than 3 mm.

### 3.5 Multiwave decomposition

The global internal tide field is a superposition of multiconstituent multimodal multidirectional internal waves. The multiwave
superposition leads to complicated spatial interference and makes it difficult to detect individual internal tidal waves and track
their generation, propagation, and dissipation. In the new model, the internal tide field is decomposed into a series of simple
plane waves. In frequency, eight principal internal tide constituents are extracted ($M_2$, $S_2$, $N_2$, $K_2$, $K_1$, $O_1$, $P_1$, and $Q_1$). In
the vertical direction, two lowest baroclinic modes are extracted for the four major constituents ($M_2$, $S_2$, $K_1$ and $O_1$). In the
horizontal direction, each internal tide constituent is decomposed into 5 waves of arbitrary directions at each grid point. All
together, 60 simple plane waves are determined at each grid point in the model. The 12 internal tide constituents and their 5
wave components are shown in Figures S3–S14 (Supplementary Materials). The multiwave decomposition gives a new view
of the global internal tide field.

For each constituent, the 5-wave summed field shows obvious interference features such as half-wavelength fluctuations
in amplitude and phase. In contrast, the 5-wave resolved components are not affected by multiwave interference. Therefore,
the decomposed results reveal a lot of new features that are previously masked by multiwave interference. In particular, the
first wave components (panels (b) in Figures S3–S14) show the largest waves at each grid point. They have large and smooth
amplitudes, so that individual long-range internal tidal beams can be clearly identified. Around the Hawaiian Ridge, there are
outgoing internal tidal beams in all 12 internal tide constituents. Because the Hawaiian Ridge is generally in west-east direction,
the internal tide radiation is dominantly southward and northward. Around the south-north aligned Izu–Bonin–Mariana Arc,

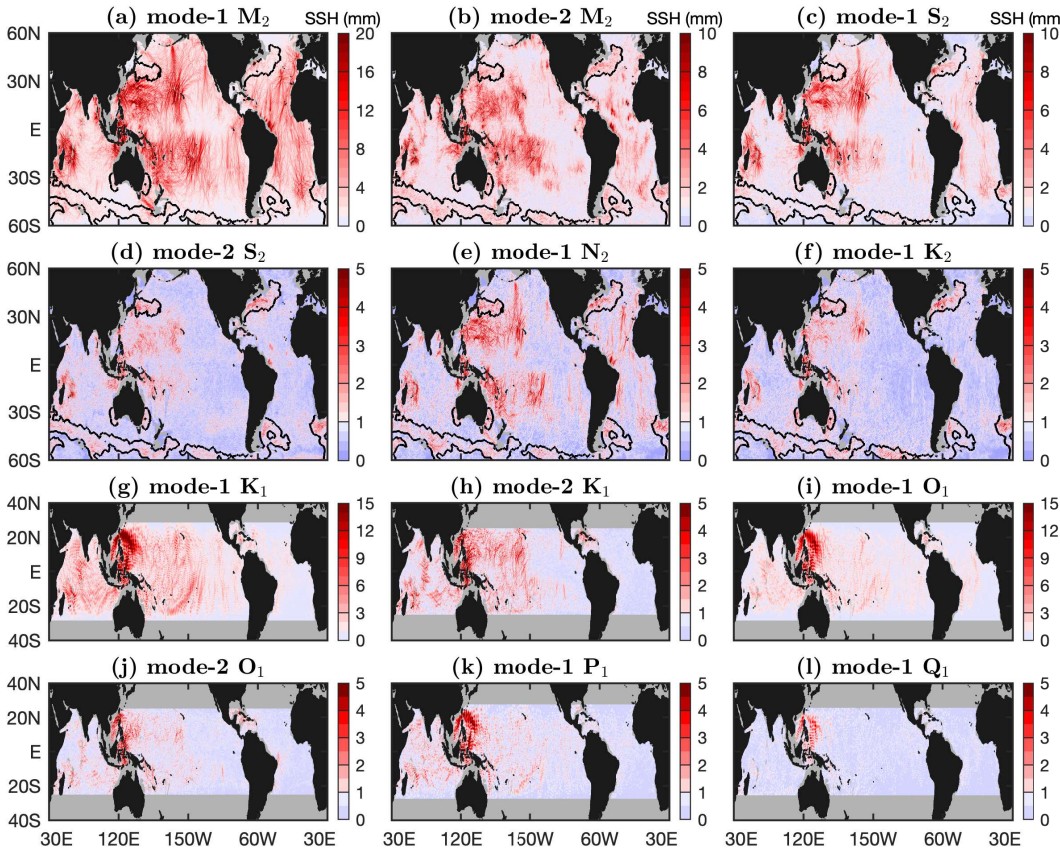

**Figure 2.** Twelve internal tide constituents in ZHAO30yr. Internal tides with amplitudes lower than 1 mm are shown in light blue. Black contours indicate regions of large model errors.

there exist westward and eastward internal tidal beams. In the Madagascar–Mascarene region, there are outgoing internal tidal beams in all directions. However, internal tidal beams shown in Figures S3–S14 may mix internal tidal waves from different generation sites. An alternative way to show internal tidal beams is using decomposed components by propagation direction (Section 5).

## 4 Errors and Evaluation

### 4.1 Model errors

Model errors are estimated using background internal tides following Zhao (2023a, b). Background internal tides are extracted from the same altimetry data following the same procedure but using tidal periods slightly different from the eight principal constituents. In other words, model errors are indicated by internal tide signals where internal tides do not exist. In principle,



model errors are determined by the given altimetry data and the mapping technique used to extract internal tides. Background internal tides do not vary much over the semidiurnal or diurnal frequency bands (Zhao, 2023b, Figure 5). This study estimates errors in semidiurnal internal tides using 12.3373 hours ($M_2$ minus 5 minutes) and errors in diurnal internal tides using 23.8511 hours ($K_1$ minus 5 minutes). Both mode-1 and mode-2 internal tides are estimated for the semidiurnal and diurnal constituents. The resulting model errors are shown in Figure 3. In regions of extremely high mesoscale eddies, the semidiurnal errors

are dominantly larger than 1 mm due to mesoscale contamination (mesoscale correction in Section 2 is not enough). These regions include the Kuroshio extension region, the Gulf Stream, the East Australian Current, the Agulhas Current, the Brazil Current, the Leeuwin Current, the loop current in the Gulf of Mexico, and the Antarctic Circumpolar Current. These regions are highlighted by black contours following Zhao et al. (2016). Our internal tide solutions in these regions are discarded. Fortunately, in most of the global ocean, model errors are very low (Figure 3, blue patches). On global average, the model

errors in all semidiurnal and diurnal constituents are lower than 1 mm. The low model errors allow us to reliably map the much weaker mode-2 constituents and minor constituents.

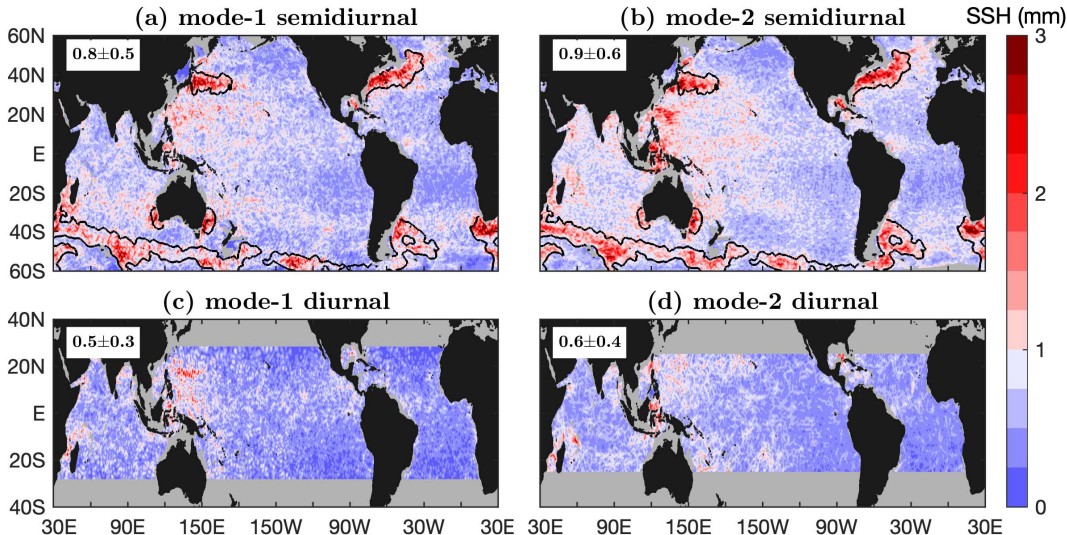

**Figure 3.** Model errors. Global means and standard deviations are given. Black contours indicate regions of large model errors due to mesoscale contamination.

### 4.2   Model evaluation

The new internal tide model is evaluated using independent nadir altimetry data in 2023 (Figure 1, blue box). Once the harmonic constants (amplitudes and phases) in the model are determined, one can predict internal tides $a(x, y, t)$ following

$$245 \quad a(x, y, t) = \sum_{n=1}^{12} A_n(x, y) \, f_n(t) \cos\left[\frac{2\pi}{T_n} t + \phi_n(x, y) + u_n(t)\right], \tag{3}$$





where $(x, y)$ indicate location, $t$ is time, $T$ is the tidal period, $f(t)$ and $u(t)$ are the nodal factor and phase of the 18.6-year cycle, $A(x, y)$ and $\phi(x, y)$ are the amplitude and phase of one internal tide constituent, and the subscript $n$ indicates the serial number of the 12 constituents, respectively. One can predict internal tides for any individual constituent or combination of constituents. Here Equation (3) only predicts the SSH displacements of internal tides. To predict their subsurface properties, one should convert the SSH displacements to subsurface properties following their baroclinic modal structures (Kelly, 2016; Zhao et al., 2016).

For each SSH measurement $\eta(x, y, t)$ in the independent data with known location $(x, y)$ and time $t$, the internal tide signal can be predicted following Equation (3) and subtracted from the original data. The variance reduction $\sigma^2(x, y, t)$ is the variance difference before and after the internal tide correction following

$$\sigma^2(x, y, t) = \eta^2(x, y, t) - [\eta(x, y, t) - a(x, y, t)]^2. \tag{4}$$

All SSH measurements in the independent altimetry data are corrected following Equation (4). The resulting variance reductions are then binned into $1°$ by $1°$ boxes. The 12 constituents are validated respectively following the same procedure. Special measures are needed to take care of the cross-talk between constituents (Section 3.3). To validate the mode-1 $P_1$ constituent, one should first predict and remove the mode-1 and mode-2 $K_1$ constituents, and vice versa. The same measure is taken in the evaluation of the $S_2$–$K_2$ pair. In addition, to validate each of the four mode-2 constituents ($M_2$, $S_2$, $K_1$, and $O_1$), one need to first predict and correct the corresponding mode-1 constituent.

The global maps of variance reductions explained by the 12 constituents are shown in Figure 4. The real internal tides reduce variance by $\sigma^2_{sig}$ but model errors increase variance by $\sigma^2_{err}$. When internal tides are larger than model errors ($\sigma^2_{sig} > \sigma^2_{err}$), positive variance reductions are obtained. Otherwise, negative variance reductions are obtained. The results show that all the constituents can cause regional positive variance reductions, because they can overcome model errors in these regions. However, these constituents also cause negative variance reductions in some regions, where the weak internal tides are lower than model errors. Figure 4 shows that negative variance reduction occurs in regions with weak internal tides, such as the equatorial and southern Pacific Ocean. Their global area-weighted mean variance reductions are 17.97 (mode-1 $M_2$), 2.26 (mode-2 $M_2$), 2.70 (mode-1 $S_2$), 0.52 (mode-2 $S_2$), 0.40 (mode-1 $N_2$), $-0.26$ (mode-1 $K_2$), 4.30 (mode-1 $K_1$), 0.39 (mode-2 $K_1$), 2.29 (mode-1 $O_1$), 0.13 (mode-2 $O_1$), 0.03 (mode-1 $P_1$), and $-0.002$ mm$^2$ (mode-1 $Q_1$), respectively. Ten constituents (but for $K_2$ and $Q_1$) cause overall positive values, because they can overcome model errors. The four minor constituents are overall weak, but they are relatively strong in the western Pacific Ocean (the Indonesian Seas, the South China Sea and the Philippine Sea). Their area-weighted mean variance reductions in the region ranging $105°$–$160°$ E, $10°$ S–$30°$ N (Figure 4, green boxes) are 1.52 ($N_2$), 0.47 ($K_2$), 1.57 ($P_1$), and 0.61 ($Q_1$) mm$^2$, respectively. The results suggest that the minor internal tide constituents are reliable in the western Pacific Ocean.

### 4.3 Comparison of ZHAO20yr and ZHAO30yr

Model evaluation shows that ZHAO30yr greatly improves over ZHAO20yr, an old model presented in Carrere et al. (2021). ZHAO20yr was constructed using 20 years of altimetry data from 1993–2012 by the obsolete mapping procedure (Zhao et al.,

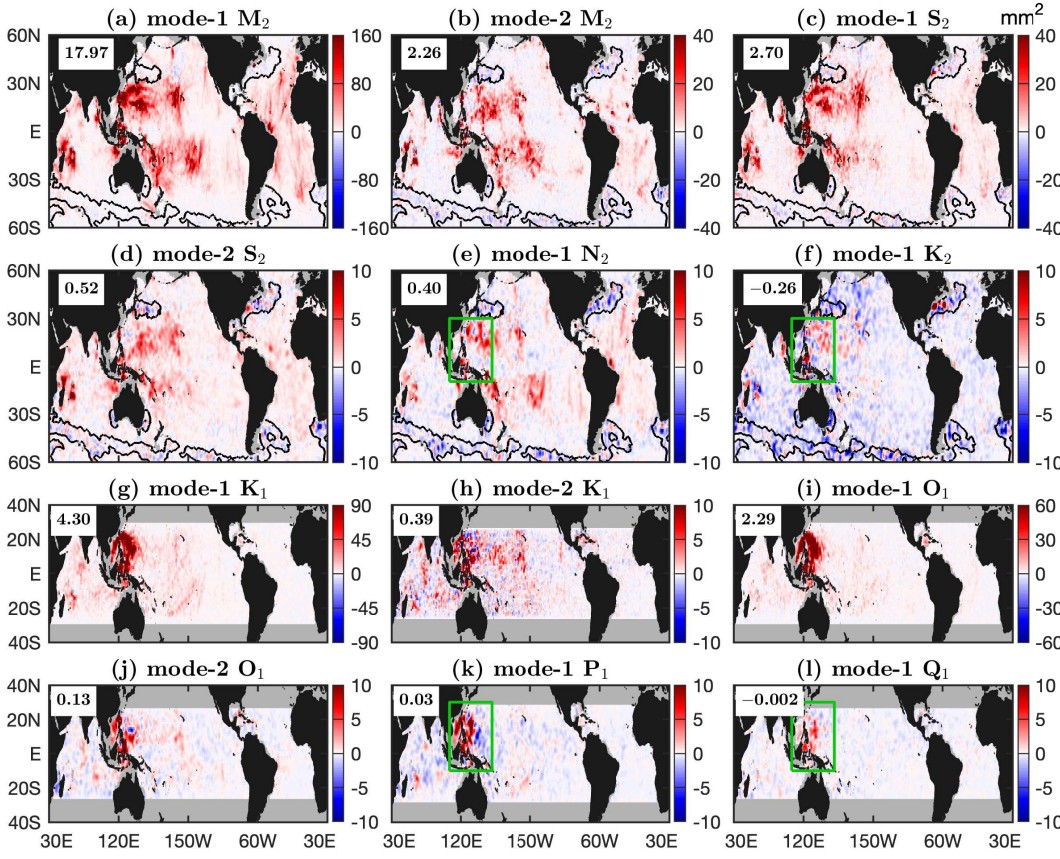

**Figure 4.** Model evaluation. Shown are variance reductions obtained in making internal tide correction to independent altimetry data in 2023. Black contours indicate regions of large model errors. Global area-weighted mean variance reductions are given. Green boxes indicate regions where minor constituents have strong signals.

2016). ZHAO20yr contains only 4 mode-1 constituents: $M_2$, $S_2$, $K_1$ and $O_1$. Both ZHAO20yr and ZHAO30yr are evaluated
using the altimetry data in 2023 following the same procedure. The resulting global variance reduction maps (not shown) are
similar to Figures 4. It is straightforward to calculate the global area-weighted mean variance reductions caused by the two
models (Figure 5). It shows that ZHAO30yr reduces more variance than ZHAO20yr for all 4 constituents. The improvement
can be quantified by the change rate of variance reduction following $(\sigma_{30yr}^2 - \sigma_{20yr}^2)/\sigma_{20yr}^2 \times 100\%$. They are 32% ($M_2$), 80%
($S_2$), 45% ($K_1$) and 36% ($O_1$), respectively. The comparison shows that ZHAO30yr significantly improves over the old model.
The improvement is mainly because ZHAO30yr is constructed using a longer data record and an improved mapping technique.

Comparison of the mode-1 $S_2$ internal tides in ZHAO20yr and ZHAO30yr reveals their differences (Figure 6). The mode-1
$S_2$ constituent is chosen because it is improved the most (80%). The improvement is also because they are constructed using
different altimetry data records. ZHAO20yr uses 20 years of altimetry data from 1993 to 2012 excluding Sun-synchronous
missions. The data record is only about 40 years long. ZHAO30yr uses all satellite altimetry data from 1993 to 2022 including





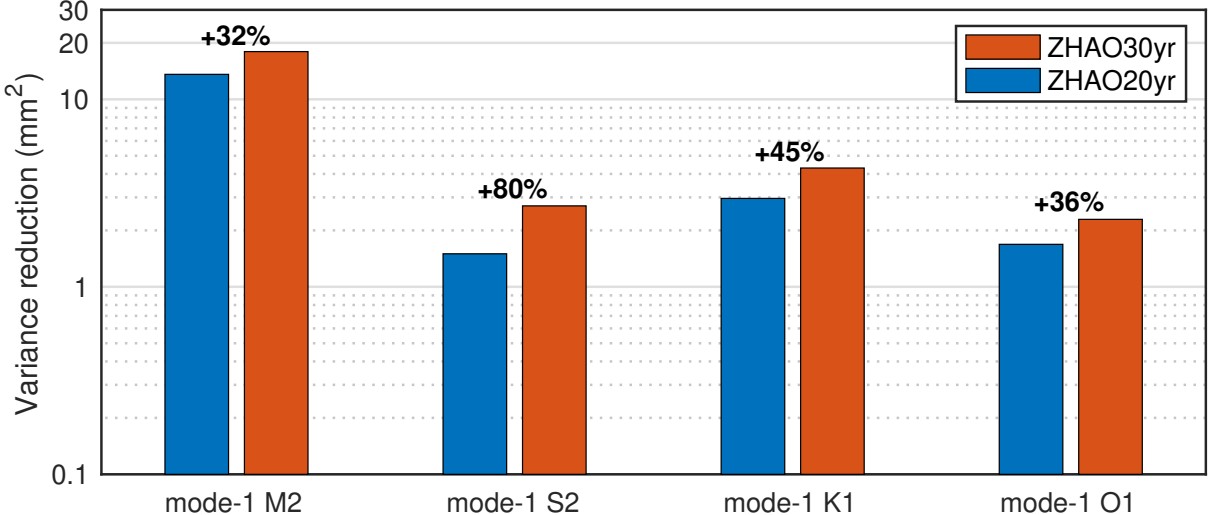

**Figure 5.** Histogram of global area-weighted mean variance reductions caused by ZHAO20yr and ZHAO30y. ZHAO30yr improves over ZHAO20yr by different percentages for different constituents.

the Sun-synchronous missions (Figure 1). The data record is about 120 years long, about 3 times larger. Due to their different data densities, the $S_2$ constituent is mapped using fitting windows of 120 km and 250 km, respectively. As a result, ZHAO30yr has a much higher spatial resolution, which allows us to detect internal tidal beams. In addition, ZHAO30yr has low model errors, as evidenced by the weak signals in the regions of large model errors.

## 5 Decomposed Components and Internal Tidal Beams

This section presents the 12 internal tide constituents, each of which has been divided into two components by propagation direction. The decomposed components reveal numerous well-defined long-range internal tidal beams. A picture is worth a thousand words. An interested reader may study the decomposed internal tide components for more features.

### 5.1 Mode-1 and mode-2 $M_2$ constituents

Figure 7 shows the mode-1 $M_2$ constituent and its northward (0°–180°) and southward (180°–360°) components. The 5-wave summed $M_2$ internal tide field (Figure 7a) shows significant small-scale spatial variations caused by multiwave interference. Internal tides with amplitudes lower than 1 mm (model errors) are shown in light blue. Figure 7a shows that mode-1 $M_2$ internal tides are dominantly greater than model errors and barely affected by model errors. Therefore, previous satellite investigations of internal tides mainly focused on the mode-1 $M_2$ constituent. In this figure, bottom topographic features are indicated by the 3000-m isobath contours (green lines). Figure 7a shows that strong mode-1 $M_2$ internal tides occur around the Hawaiian Ridge, around the French Polynesian Ridge, in the western Pacific Ocean, in the Madagascar–Mascarene region, and in the

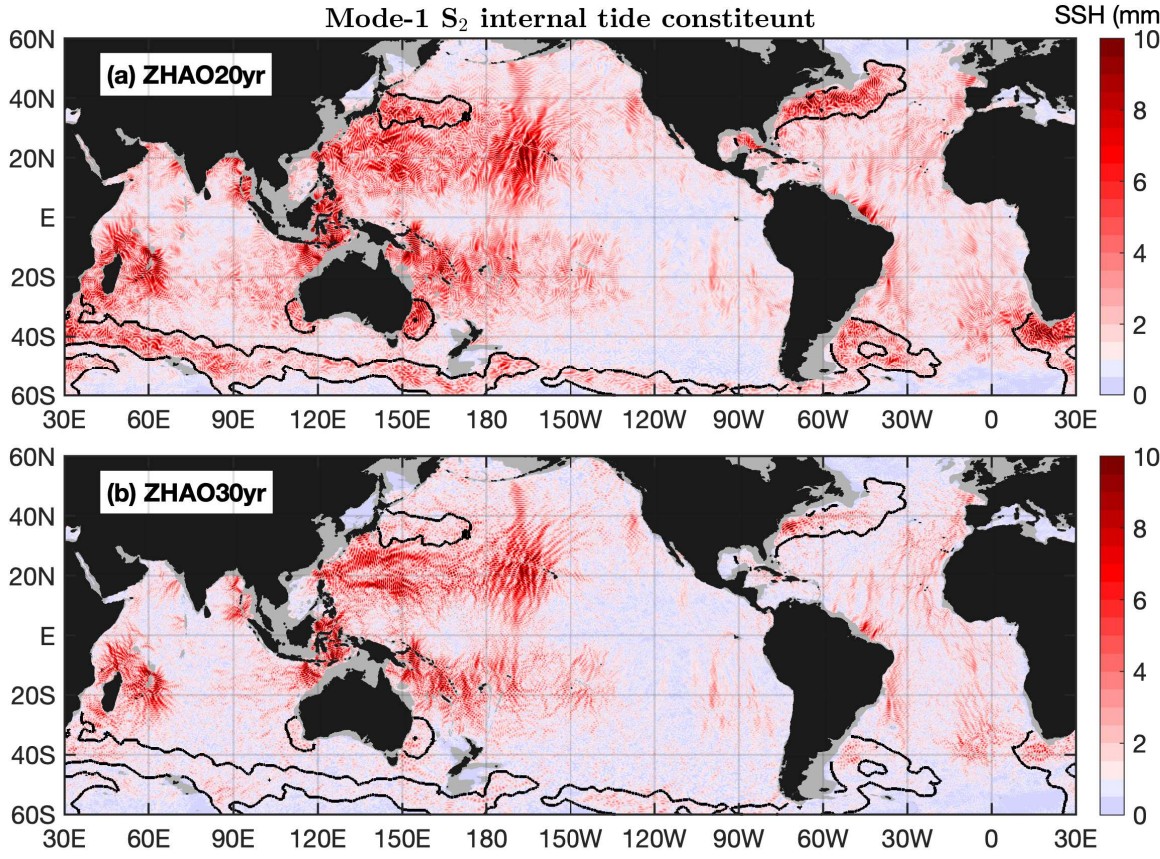

**Figure 6.** Mode-1 $S_2$ internal tide constituents in (a) ZHAO20yr and (b) ZHAO30yr. Black contours indicate regions of large model errors. ZHAO30yr has a higher spatial resolution and lower model errors.

Indonesian Seas. These regions have long been recognized in previous studies by satellite altimetry (Ray and Zaron, 2016; Zhao et al., 2016; Zaron, 2019).

We divide the mode-1 $M_2$ constituent into northward and southward components by propagation direction. The northward and southward components contain the largest waves at each grid point with propagation directions ranging $0°–180°$ and $180°–360°$, respectively. Such a division can separate long-range internal tidal beams in different directions. But the eastward $(−90°–90°)$ and westward $(90°–270°)$ decomposition can better resolve semidiurnal internal tidal beams in regions such as the western Pacific Ocean. In the decomposed components (Figures 7b, c), internal tides with amplitudes lower than 2/3 mm are shown in light blue, which is around model errors for individual internal tidal waves, in contrast to 1 mm for the 5-wave sum. Our colormap adjustment is to highlight internal tidal beams. The northward and southward components show numerous well-defined long-range internal tidal beams, which are featured by large amplitudes, linear-increasing phases, and across-beam co-phase lines (Section 6). The results are consistent with our satellite observations using short data records (Zhao and Alford,





2009; Zhao, 2014; Zhao et al., 2016), because mode-1 $M_2$ internal tides are less affected by model errors. Nevertheless, low model errors are key to track their along-beam amplitude and phase changes (Zhao, 2016).

All mode-1 $M_2$ beams are associated with notable topographic features. In the Pacific Ocean, mode-1 $M_2$ beams radiate from
320 the Hawaiian Ridge, the Line Islands Ridge, the French Polynesian Ridge, the Izu–Bonin–Mariana Arc, the Luzon Strait, the Amukta Pass (Alaska), the Mendocino Ridge, the Macquarie Ridge, and the Eastern Pacific Rise. In the Indian Ocean, mode-1 $M_2$ beams are from the Mascarene Plateau, the Ninety East Ridge, the Andaman Islands chain, and the Indian western shelf. In the Atlantic Ocean, mode-1 $M_2$ beams are from the Mid-Atlantic Ridge, the Amazon shelf, the Vitória-Trindade Ridge, the Walvis Ridge, the Great Meteor Seamount, and the Cape Verde. In addition, there are many short-range mode-1 $M_2$ beams
from the narrow straits in the Indonesian Seas, the Coral Sea, and the Caribbean Sea. Section 6.1 has shown mode-1 $M_2$ beams off the Amazon shelf.

Figure 8 shows the mode-2 $M_2$ constituent and its northward ($0°$–$180°$) and southward ($180°$–$360°$) components. The mode-2 $M_2$ constituent is weak; however, its SSH amplitudes may be up to 15 mm, larger than minor constituents $N_2$ and $K_2$. Mode-2 $M_2$ internal tides mainly occur at low latitudes, which is likely determined by the latitudinal structure of ocean stratification.
The mode-2 $M_2$ constituent is divided into northward and southward components following the same method. The northward and southward internal tides are to the north and south of notable topographic features, respectively (Figures 8b, c), suggesting that they are well separated. The decomposed components show numerous well-defined mode-2 $M_2$ beams. Among them, the mode-2 $M_2$ beams off the Amazon shelf will be studied in Section 6.1.

In the Pacific Ocean, the following remarkable generation sources have been recognized: the Hawaii region in the North
Pacific, seamounts, island chains and ridges in the western Pacific Ocean, the western South Pacific including the Coral Sea, the French Polynesian Ridge in the South Pacific, the Eastern Pacific Rise, the Alaskan shelf, and the Indonesian Seas. The Indian ocean has the following remarkable sources: the Madagascar–Mascarene region, the Indian western shelf, the Bay of Bengal, the Andaman Sea, and the Australian Northwest Coast. In the Atlantic Ocean, mode-2 $M_2$ beans are associated with the Mid-Atlantic Ridge, the Amazon shelf, the Walvis Ridge, and several scattered seamounts. In Figure 8, some mode-2 $M_2$
beams are highlighted using blue circles. Note that ZHAO30yr presents a better mode-2 $M_2$ field than Zhao (2017), due to the longer data record and improved mapping technique (Sections 2 and 3).

Mode-1 and mode-2 $M_2$ beams have the following different features. (1) Mode-2 $M_2$ beams are shorter than mode-1 $M_2$ beams. Mode-2 $M_2$ internal tides can travel over hundreds of km, in contrast to thousands of km for mode-1 $M_2$ internal tides. (2) Mode-2 $M_2$ beams are narrower than mode-1 $M_2$ beams. That is why the altimetry data along nonrepeat tracks are
345 important in mapping mode-2 internal tides. (3) There are more mode-2 $M_2$ beams than mode-1 $M_2$ beams, because mode-2 $M_2$ beams can be induced by small-scale topographic features such as isolated seamounts (Zhao, 2017). Because mode-2 $M_2$ beams are narrower and shorter, one can easily pinpoint their sources over topographic features. For example, some isolated mode-2 beams are unambiguously associated with known topographic features (Figure 8, blue circles). As reported in Zhao (2017), mode-2 beams are usually generated over isolated seamounts and trenches. Note that the mode-1 and mode-2 $M_2$
beams shown in Figures 7 and 8 contain a lot of important information. A detailed examination of the $M_2$ internal tides region by region or beam by beam can deepen our understanding of their generation, propagation, and dissipation.

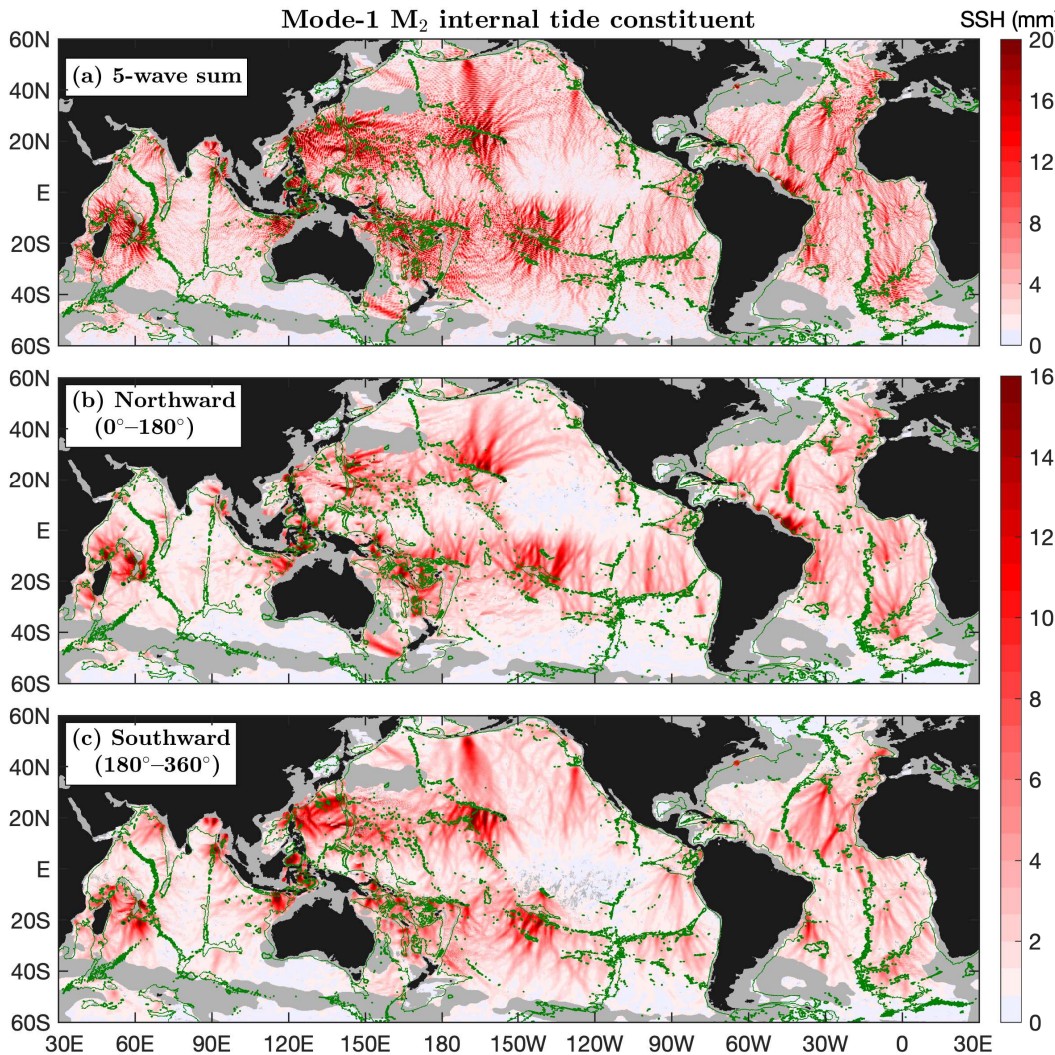

**Figure 7.** Mode-1 $M_2$ internal tide constituent. (a) The 5-wave sum. (b) Northward component. (c) Southward component. Internal tides in regions of large model errors or shallower than 1000 m in depth are shown in gray. Green contours indicate the 3000-m isobath. Numerous well-defined long-range internal tidal beams are associated with notable topographic features.

## 5.2 Mode-1 and mode-2 $S_2$ constituents

Figure 9 shows the mode-1 $S_2$ constituent and its northward (0°–180°) and southward (180°–360°) components. Similar to the mode-1 $M_2$ constituent, the decomposed components (Figures 9b, c) reveal numerous long-range mode-1 $S_2$ internal tidal beams. It shows that the northward mode-1 $S_2$ beams from the Hawaiian Ridge reaching the Alaskan shelf and the southward mode-1 $S_2$ beams from Amukta Pass (Alaska) reaching the Hawaiian Ridge. These long-range mode-1 $S_2$ beams have been



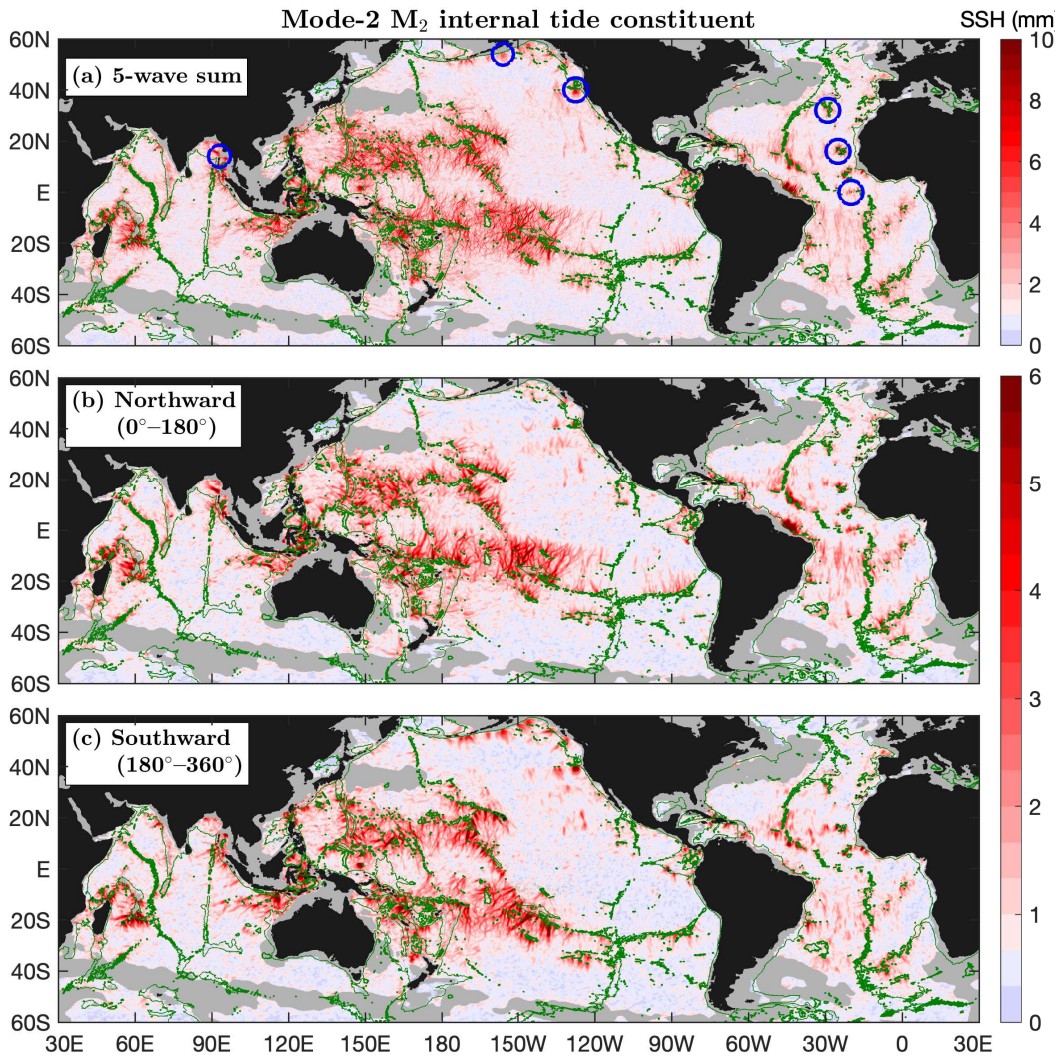

**Figure 8.** As in Figure 7 but for the mode-2 $M_2$ internal tide constituent. Well-defined long-range internal tidal beams are associated with notable topographic features. Blue circles mark some isolated beams. Mode-2 $M_2$ beams are narrower and shorter than mode-1 $M_2$ beams. [too many circles]

observed in Zhao (2017). Additionally, mode-1 $S_2$ beams are observed to radiate from notable topographic features such as the Mendocino Ridge, the French Polynesian Ridge, the Izu–Bonin–Mariana Arc, the Australian Northwest shelf, the Lombok Strait, the Andaman Islands chain, and the Mascarene Plateau. In the Atlantic Ocean, mode-1 $S_2$ beams radiate from the Amazon shelf, the Vitória-Trindade Ridge, the Walvis Ridge, and isolated seamounts along the Mid-Atlantic Ridge. Compared to Zhao (2017); however, ZHAO30yr can better resolve mode-1 $S_2$ beams, due to its higher spatial resolution and lower noise level.

In general, the mode-1 $M_2$ and $S_2$ constituents show similar internal tidal beams. For example, one notable topographic feature (e.g., the Hawaiian Ridge) usually generate both $M_2$ and $S_2$ internal tidal beams. But they have the following different

features. (1) Mode-1 $S_2$ beams are usually narrower and shorter than corresponding mode-1 $M_2$ beams, likely because the $S_2$ internal tides are weak and easily masked by model errors. (2) Mode-1 $S_2$ internal tides in the southern Pacific Ocean (e.g., the French Polynesian Ridge, the Tasman Sea, and the Coral Sea) are relatively weaker, which is caused by the relatively weaker $S_2$ barotropic tides in this region (Zhao, 2017).

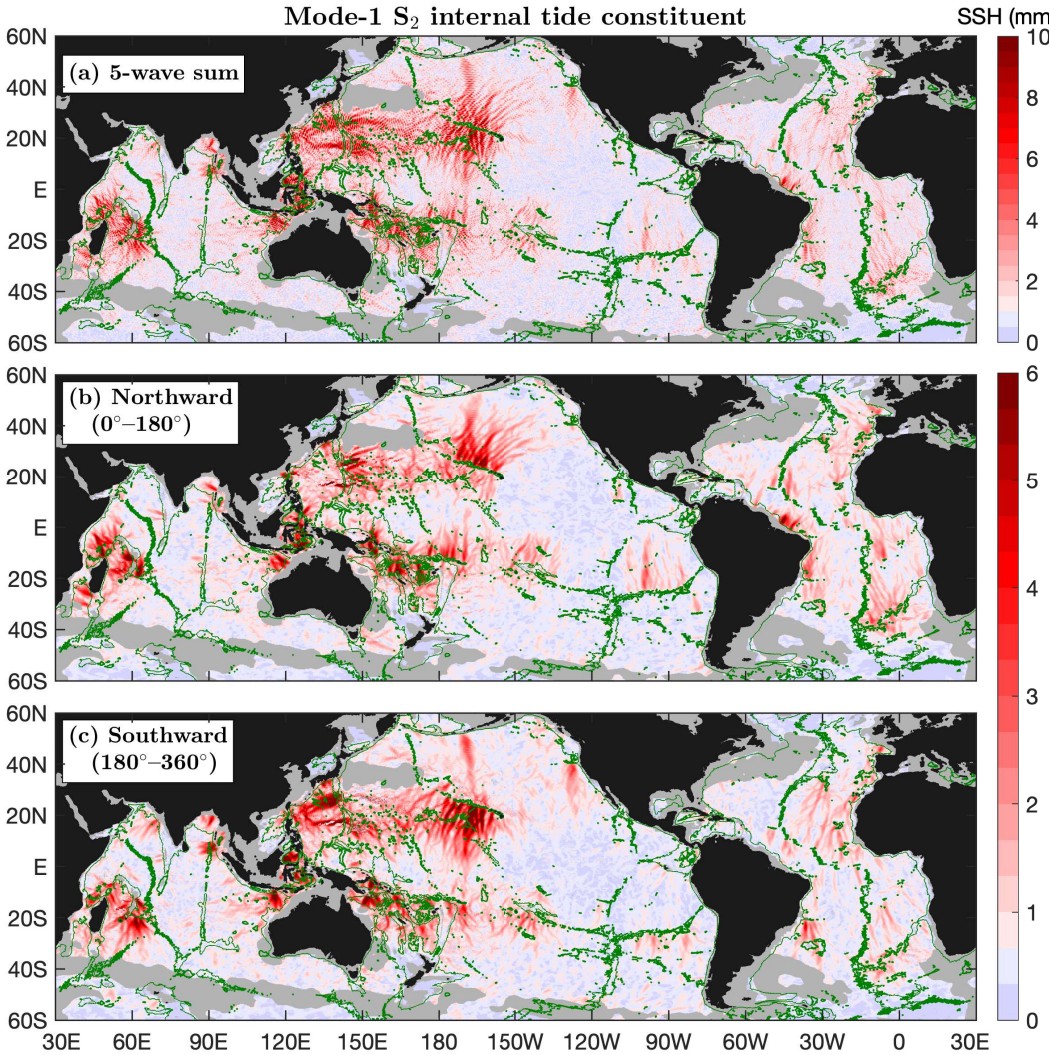

**Figure 9.** As in Figure 7 but for the mode-1 $S_2$ internal tide constituent.

Figure 10 shows the mode-2 $S_2$ constituent and its northward ($0°$–$180°$) and southward ($180°$–$360°$) components. The

370 mode-2 $S_2$ constituent is much weaker. Its amplitudes are usually lower than 4 mm. Therefore, in most of the global ocean,



mode-2 $S_2$ internal tides are overwhelmed by model errors. Fortunately, mode-2 $S_2$ internal tides are strong enough to overcome model errors in their source regions such as the Hawaiian Ridge, the western Pacific Ocean, the Indonesian Sea, the Coral Sea, the Madagascar–Mascarene region, and the Amazon shelf. Note that the 2–3-mm mode-2 $S_2$ internal tides are real signals, because they can cause positive variance reductions in making internal tide correction to independent altimetry data (Figure 4d). Likewise, the decomposed components (Figures 10b, c) show numerous well-defined mode-2 $S_2$ beams associated with topographic features. For example, one mode-2 $S_2$ beam radiates from the Andaman islands chain (Figure 10b, blue circle). In addition, the mode-2 $S_2$ beams off the Amazon shelf will be studied in Section 6.1. In conclusion, the mode-1 and mode-2 $S_2$ constituents are weak; however, our multiwave decomposition can still resolve well-defined internal tidal beams.

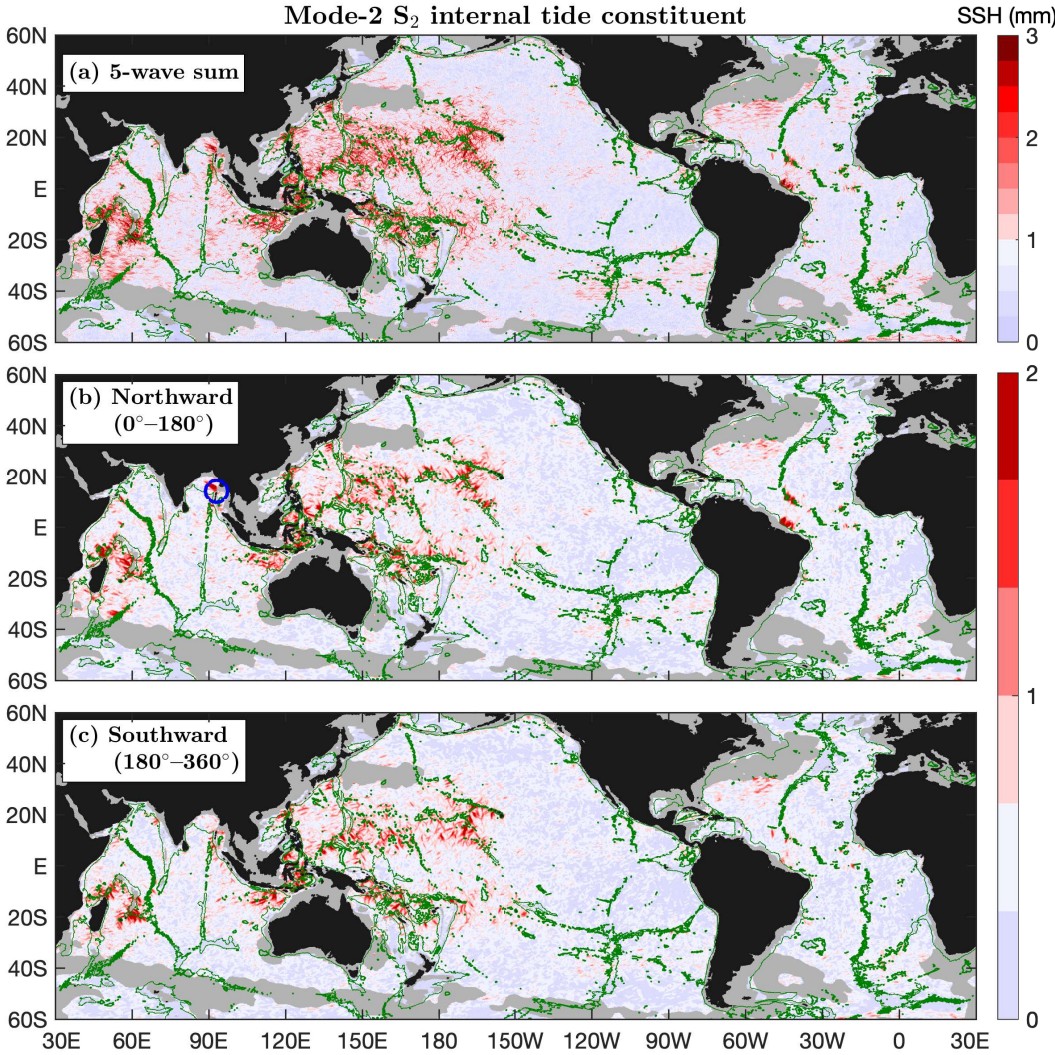

**Figure 10.** As in Figure 8 but for the mode-2 $S_2$ internal tide constituent.



### 5.3 Mode-1 $N_2$ constituent

Figure 11 shows the mode-1 $N_2$ constituent and its northward (0°–180°) and southward (180°–360°) components. The mode-1 $N_2$ constituent has amplitudes up to 6 mm and can overcome model errors in most of the global ocean. In the 5-wave summed filed (Figure 11a), mode-1 $N_2$ beams are masked by multiwave interference; however, the decomposed components (Figures 11b, c) reveal numerous well-defined long-range mode-1 $N_2$ beams. For example, one can observe both northward and southward $N_2$ beams radiating from the Hawaiian Ridge, the French Polynesian Ridge, and the Macquarie Ridge. As a

pioneering work, Zhao (2023b) mapped the mode-1 $N_2$ constituent using 27 years of altimetry data (1993–2019) by the same mapping technique used in this study. A comparison shows that the two models are almost same, because 90% of the two data records are the same (27-year vs 30-year). Zhao (2023b) gave a detailed description of the mode-1 $N_2$ constituent and compared with the mode-1 $M_2$ constituent. It was reported that mode-1 $N_2$ internal tides can travel from the Hawaiian Ridge to the Alaska and that the southward beams from the Mendocino Ridge can travel over 2000 km. Zhao (2023b) showed that the

mode-1 $M_2$ and $N_2$ internal tides have similar spatial patterns and that the $N_2$ amplitudes are about 1/5 of the $M_2$ amplitudes. The mode-1 $N_2$ beams off the Amazon continental shelf will be studied in Section 6.1. To avoid repetition, an interested reader is referred to Zhao (2023b) for a detailed description of the mode-1 $N_2$ constituent.

### 5.4 Mode-1 $K_2$ constituent

Figure 12 shows the mode-1 $K_2$ constituent and its northward (0°–180°) and southward (180°–360°) components. The weak

mode-1 $K_2$ constituent usually has amplitudes lower than 4 mm; therefore, it is masked by model errors in most of the global ocean. However, the mode-1 $K_2$ constituent can overcome model errors in its source regions such as the Hawaiian Ridge, the Indonesian Seas, and the western Pacific Ocean, where positive variance reduction is obtained in making internal tide correction to independent altimetry data. The decomposed components show numerous well-defined mode-1 $K_2$ beams. For example, one can see southward and northward beams from the Hawaiian Ridge. Note that mode-1 $K_2$ beams from the Hawaiian Ridge can

be tracked over 1000 km before disappearance. In addition, mode-1 $K_2$ beams radiate from the Mascarene Plateau, the Luzon Strait, the Mendocino Ridge, the Lombok Strait, the Vitória-Trindade Ridge, and the Amazon shelf. Among them, the mode-1 $K_2$ beams off the Amazon shelf will be studied in Section 6.1.

### 5.5 Mode-1 and mode-2 $K_1$ constituents

Figure 13 shows the mode-1 $K_1$ constituent and its eastward (−90°–90°) and westward (90°–270°) components. In this study,

all diurnal internal tide constituents are divided into eastward and westward components, because such a division can better resolve internal tidal beams in most of the global ocean. The eastward and westward components contain the largest wave at each grid point with propagation direction ranging −90°–90° and 90°–270°, respectively. All diurnal internal tide constituents are limited within about ±30°, poleward of which there are no propagating diurnal internal tides. The 3000-m isobath contours are overlain to show topographic features. The most remarkable feature of the mode-1 $K_1$ constituent and other diurnal

constituents is the asymmetry between the western and eastern hemispheres: All diurnal constituents are strong in the eastern



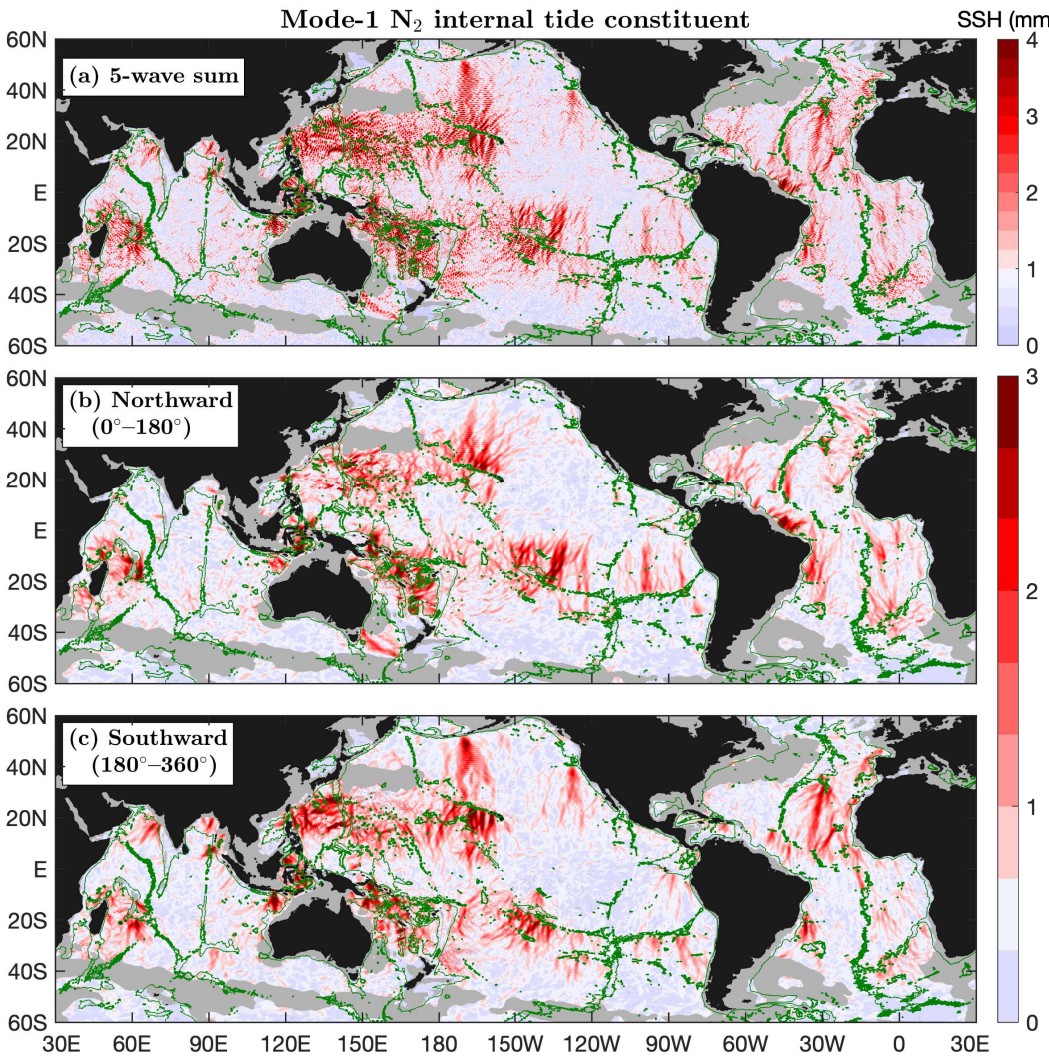

**Figure 11.** As in Figure 7 but for the mode-1 $N_2$ internal tide constituent.

hemisphere and weak in the western hemisphere. This feature stems from the asymmetry of the diurnal barotropic tides (Egbert and Ray, 2003). It has long been known that diurnal barotropic tides are weak in the Atlantic Ocean.

The Luzon Strait is the strongest source of mode-1 $K_1$ internal tides, and radiates mode-1 $K_1$ beams eastward into the western Pacific Ocean and westward into the South China Sea. In their long-range propagation, both beams refract toward the equator due to the beta effect (Zhao, 2014). The Tonga–Kermadec Ridge radiates another long-range mode-1 $K_1$ beam that propagates northeastward over 3000 km and refracts toward the equator due to the beta effect (Figure 13b). The decomposed components (Figures 13b, c) show numerous well-defined long-range mode-1 $K_1$ beams. In the Indian Ocean, mode-1 $K_1$ beams radiate from the Indian western shelf, the Ninety East Ridge, the Andaman Islands chain, and the Mascarene Plateau. In the Pacific

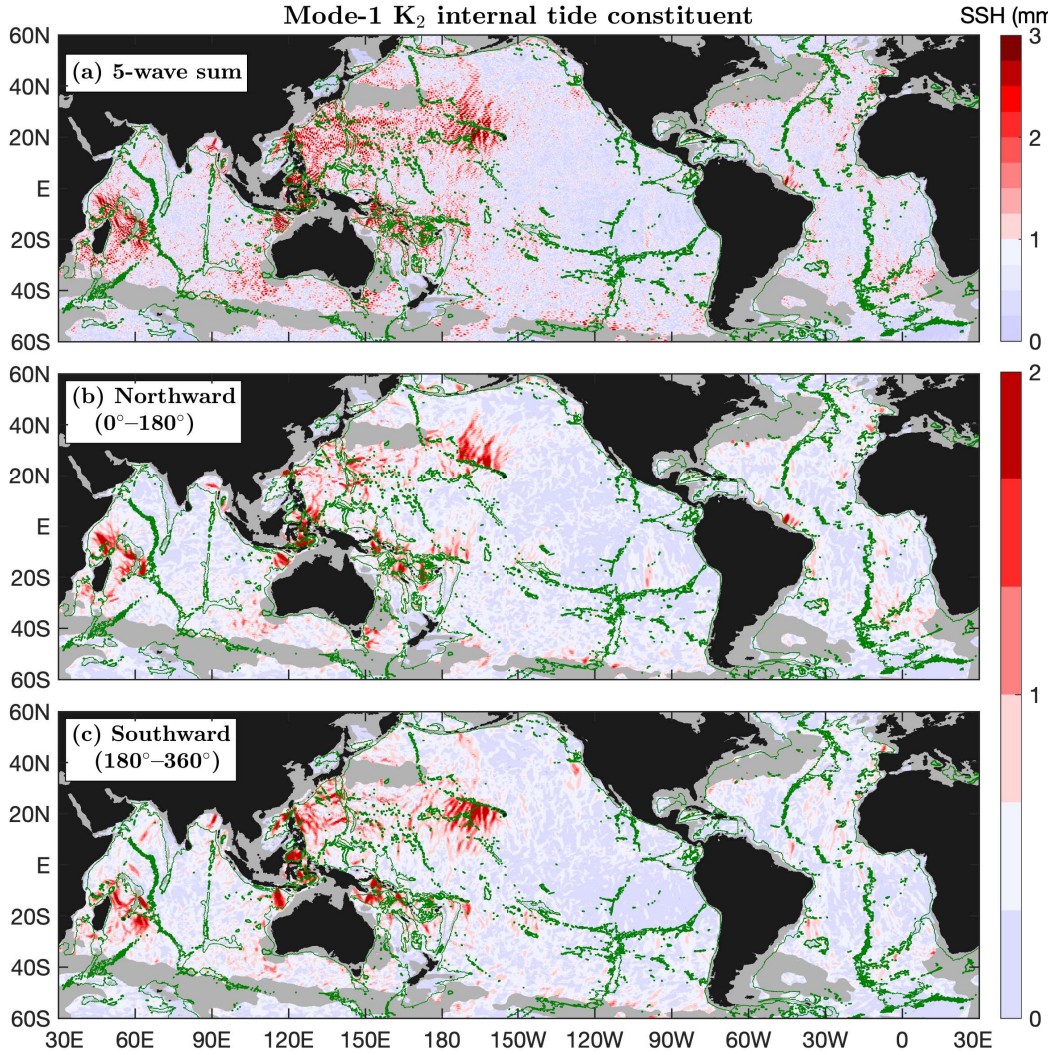

**Figure 12.** As in Figure 7 but for the mode-1 $K_2$ internal tide constituent.

Ocean, mode-1 $K_1$ beams are from the Luzon Strait, the Indonesian Seas, the Hawaiian Ridge, the Line Islands Ridge, the

420 French Polynesian Ridge, and the eastern Pacific Rise. In the Atlantic Ocean, weak but well-defined mode-1 $K_1$ beams are observed in the Caribbean Sea and from topographic features. Among them, the $K_1$ internal tidal beam from the Mona Passage (Figure 13b, circle) has been well studied (Dushaw, 2006; Dushaw and Menemenlis, 2023). One can see weak mode-1 $K_1$ beams offshore from the Amazon shelf and northward from the Vitória-Trindade Ridge (Figure 13b, circles). Among them, the westward mode-1 $K_1$ beams in the Arabian Sea will be examined in Section 6.2.

Figure 14 shows the mode-2 $K_1$ constituent and its eastward ($-90°$–$90°$) and westward ($90°$–$270°$) components. The weak mode-2 $K_1$ constituent has amplitudes up to 5 mm. The Luzon Strait is a strong source of mode-2 $K_1$ internal tides. The

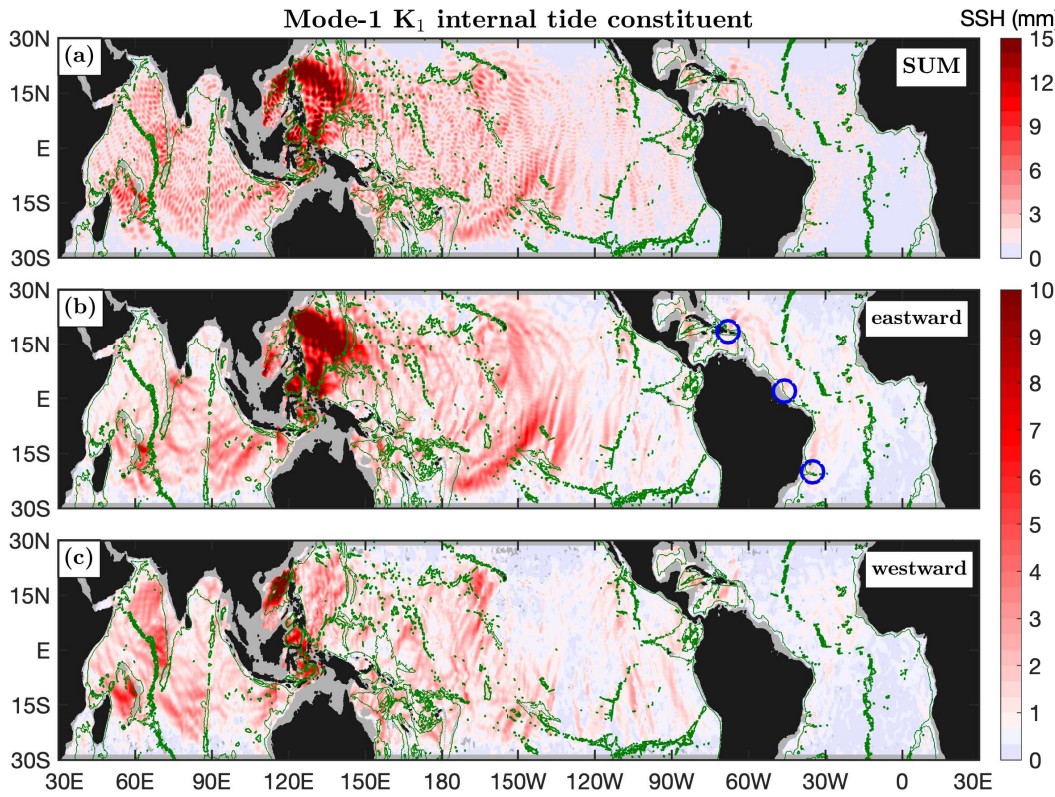

**Figure 13.** Mode-1 $K_1$ internal tide constituent. (a) The 5-wave sum. (b) Eastward component ($-90°$–$90°$). (c) Westward component ($90°$–$270°$). Internal tides in regions shallower than 1000 m in depth are shown in gray. Well-defined long-range mode-1 $K_1$ beams are associated with notable topographic features. Blue circles mark isolated mode-1 $K_1$ beams in the Atlantic Ocean.

decomposed components (Figures 14b, c) show numerous mode-2 $K_1$ beams associated with notable topographic features. For example, westward mode-2 $K_1$ beams occur in the Arabian Sea (Section 6.2), where the beams are from the Chagos–Laccadive Ridge and the western shelf of India. In the Indian Ocean, mode-2 $K_1$ beams radiate from the Ninety East Ridge, the Andaman Islands chain, and the Mascarene Plateau. In the Pacific Ocean, mode-2 $K_1$ beams are from the Luzon Strait, the Indonesian Seas, the Hawaiian Ridge, the Line Islands Ridge, the French Polynesian Ridge, and the eastern Pacific Rise. In the Atlantic Ocean, the mode-2 $K_1$ internal tides are very weak.

Both mode-1 and mode-2 $K_1$ constituents are strong in the eastern hemisphere and weak in the western hemisphere. The mode-1 and mode-2 $K_1$ constituents have different spatial patterns. Mode-2 $K_1$ beams are shorter and narrower than mode-1 $K_1$ beams. The short and narrow mode-2 $K_1$ beams allow us to pinpoint their generation sites. As an example, there are two isolated mode-2 $K_1$ beams in the eastern Pacific Ocean (Figure 14, blue circles). They originate at two notable seamounts and propagate northward over 1000 km. Answering this question will improve our understanding of the generation of internal tides. In addition, the mode-1 and mode-2 $K_2$ beams have different spatial patterns (Figures 13 and14).

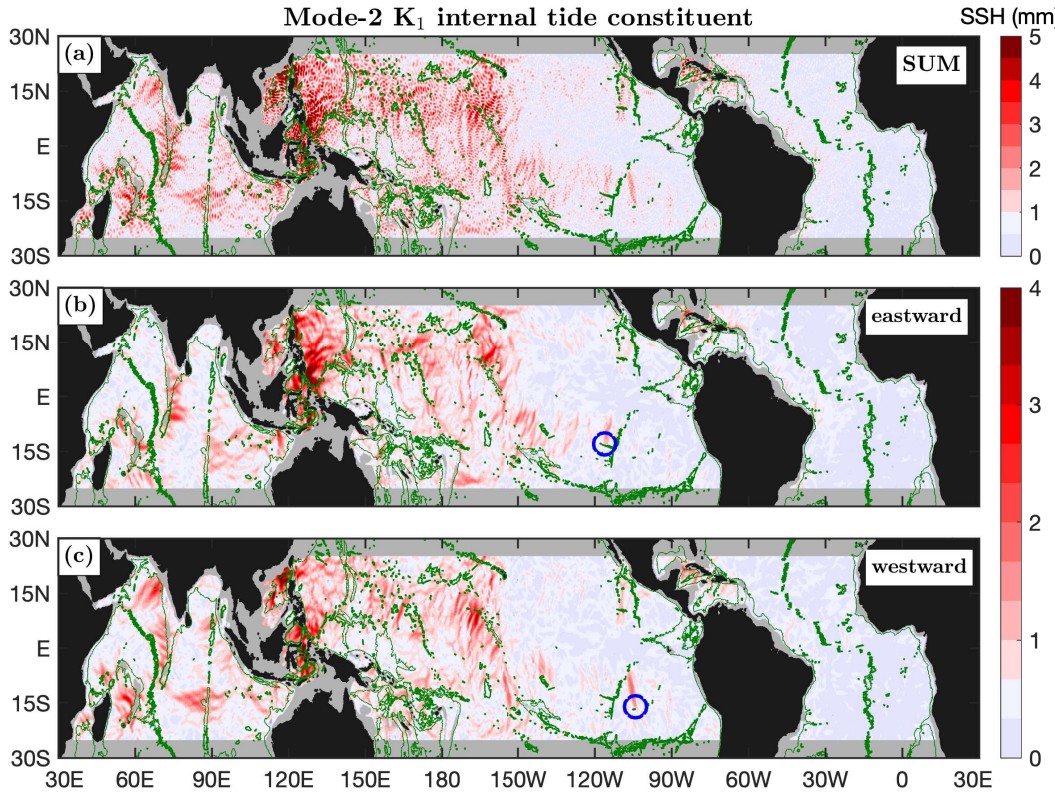

**Figure 14.** As in Figure 13 but for the mode-2 $K_1$ internal tide constituent.

### 5.6 Mode-1 and mode-2 $O_1$ constituents

Figure 15 shows the mode-1 $O_1$ constituent and its eastward ($-90°$–$90°$) and westward ($90°$–$270°$) components. The mode-1 $O_1$ and $K_1$ constituents have similar spatial patterns (Figures 13 and 15; correlation coefficient is 0.79), but that the $K_1$ constituent is about 50% stronger in amplitude. The decomposed components (Figures 15b, c) reveal numerous well-defined long-range mode-1 $O_1$ beams. The Luzon Strait generates the strongest mode-1 $O_1$ internal tides, which propagate eastward into the western Pacific Ocean and westward into the South China Sea. Like mode-1 $K_1$ beams, the two long-range beams

refract in propagation due to the beta effect. The Tonga–Kermadec Ridge radiates a strong northeastward mode-1 $O_1$ beam, which travels over 2000 km and refracts in propagation. Another similar feature is that mode-1 $O_1$ internal tides are also weak in the Atlantic Ocean and eastern Pacific Ocean, and strong in the western Pacific Ocean and Indian Ocean. In the Indian Ocean, mode-1 $O_1$ beams are from the Indian western shelf, the Chagos–Laccadive Ridge, the Mascarene Plateau, and the Ninety East Ridge. Although the $O_1$ amplitudes are low, well-defined mode-1 $O_1$ beams are observed from the Amazon shelf,

the Vitória-Trindade Ridge, and the Sierra Leone Rise in the Atlantic Ocean (Figure 15, blue circles).



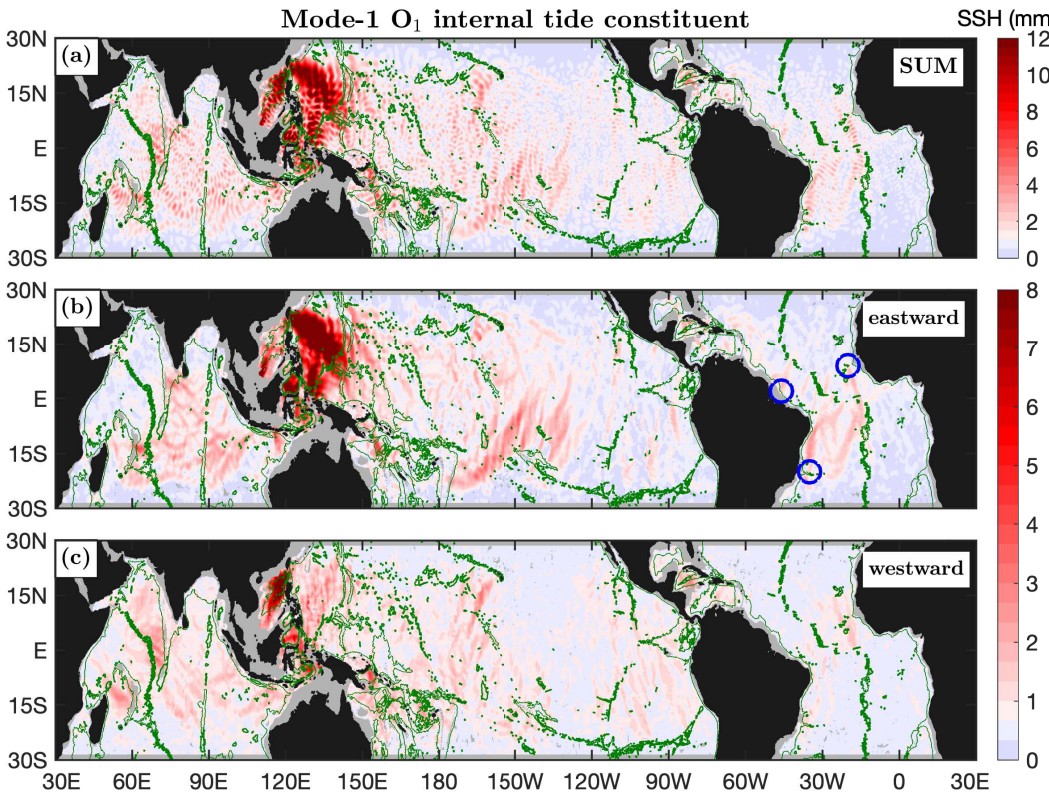

**Figure 15.** As in Figure 13 but for the mode-1 $O_1$ internal tide constituent.

Figure 16 shows the mode-2 $O_1$ constituent and its eastward ($-90°$–$90°$) and westward ($90°$–$270°$) components. The mode-2 $O_1$ and $K_1$ constituents have some degree of similarity, with a correlation coefficient of 0.56. The mode-2 $O_2$ amplitudes are up to 5 mm around the Luzon Strait. Like other diurnal constituents, the mode-2 $O_1$ constituent is strong in the eastern hemisphere and weak in the western hemisphere. Strong mode-2 $O_1$ internal tides occur in the Indonesian Seas and around the Luzon Strait. Mode-2 $O_1$ beams are also from the Line Islands Ridge and the Izu–Bonin–Mariana Arc. In the eastern Pacific Ocean, there are two singular mode-2 $O_1$ beams (Figure 16, blue circles), overlapping with the two mode-2 $K_1$ beams (Figure 14, blue circles). In the Indian Ocean, mode-2 $O_1$ beams are from the Indian western shelf, the Chagos–Laccadive Ridge, the Mascarene Plateau, and the Ninety East Ridge (Section 6.2). In the Atlantic Ocean, there is one outstanding beam propagating northward along the Brazilian shelf. In summary, the mode-1 and mode-2 $O_1$ constituents are similar to the mode-1 and mode-2 $K_1$ constituents, respectively, but the $K_1$ constituents are about 50% larger than the $O_1$ constituents. The decomposed components show numerous well-defined long-range mode-1 and mode-2 $O_1$ beams.

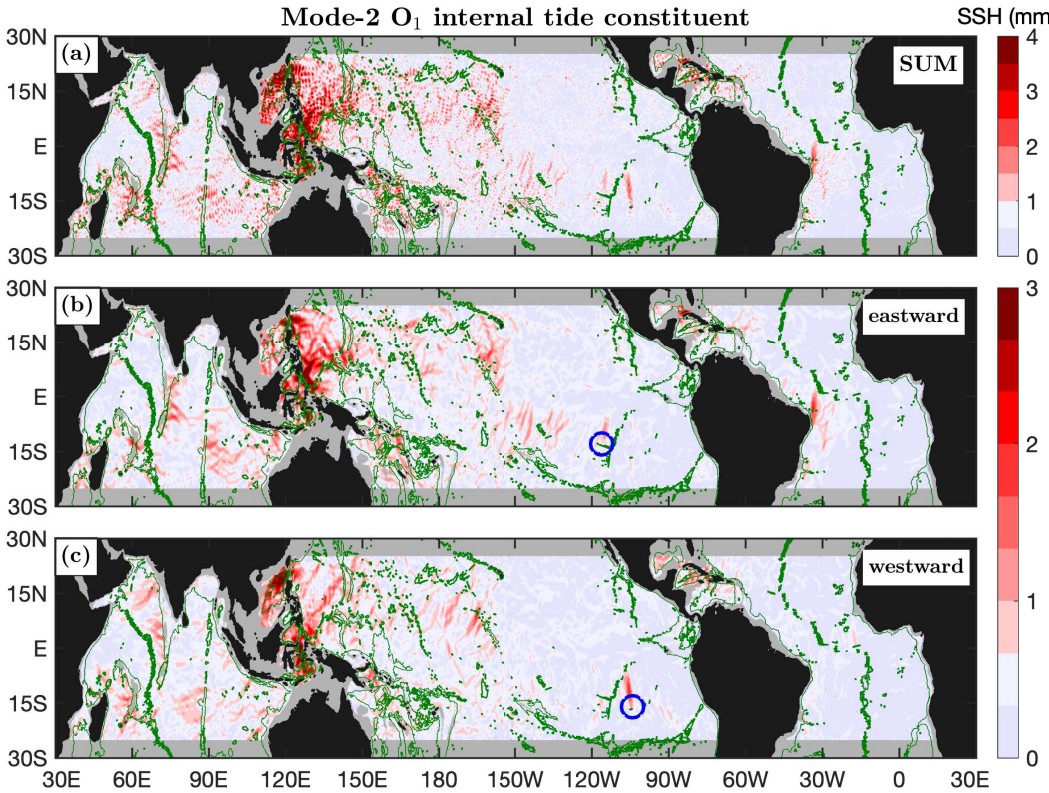

**Figure 16.** As in Figure 14 but for the mode-2 $O_1$ internal tide constituent.

### 5.7 Mode-1 $P_1$ constituent

Figure 17 shows the mode-1 $P_1$ constituent and its eastward ($-90°$–$90°$) and westward ($90°$–$270°$) components. The mode-1 $P_1$ amplitudes can be up to 5 mm. The Luzon Strait is the strongest generation source and radiate mode-1 $P_1$ beams eastward into

the western Pacific Ocean and westward into the South China Sea. The Tango–Kermedac Ridge is another strong generation source and radiates one $P_1$ beam northeastward. Additionally, mode-1 $P_1$ beams are observed from the French Polynesian Ridge and the Hawaiian Ridge. In the Indian Ocean, eastward and westward beams radiate from the Ninety East Ridge. Note that the mode-1 $P_1$ and $K_1$ constituents have similar spatial patterns and their amplitudes have a scaling factor of 1/3. In addition, $K_1$ and $P_1$ are different by 2 cpy in frequency. Their superposition forms a semiannual cycle, which should be accounted in the

study of their seasonal variations.

### 5.8 Mode-1 $Q_1$ constituent

Figure 18 shows the mode-1 $Q_1$ constituent and its eastward ($-90°$–$90°$) and westward ($90°$–$270°$) components. The mode-1 $Q_1$ constituent is very weak and its amplitudes are usually lower than 3 mm. The mode-1 $Q_1$ constituent is overwhelmed by

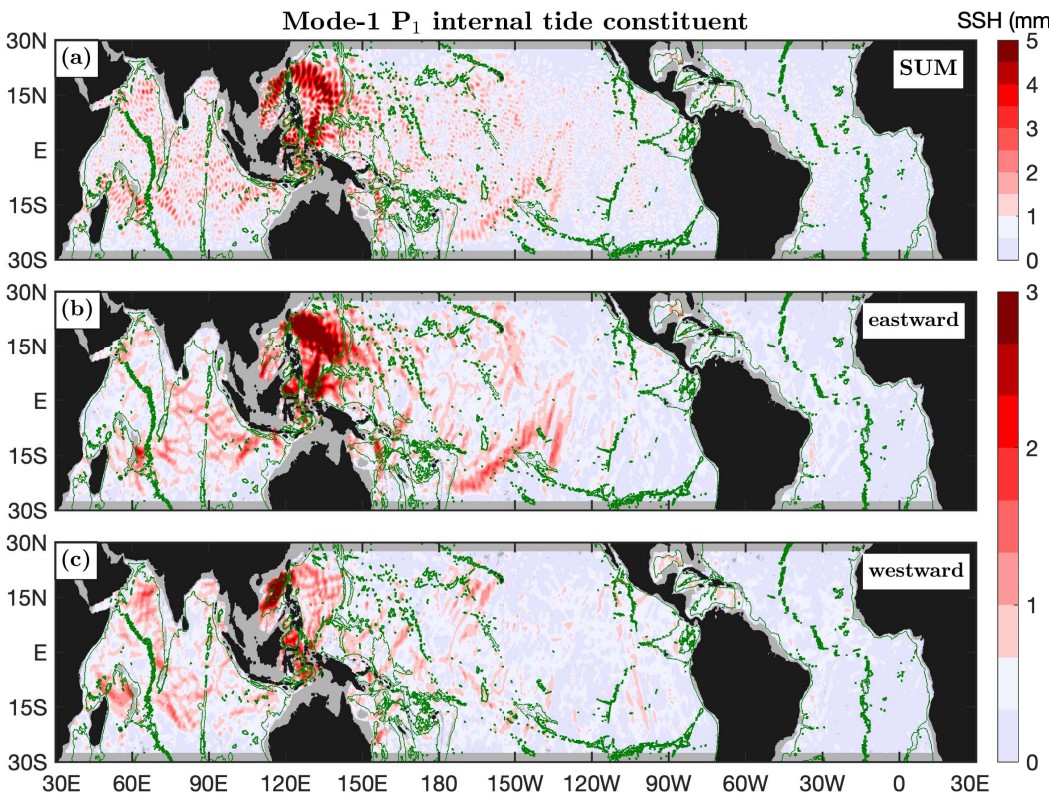

**Figure 17.** As in Figure 13 but for the mode-1 $P_1$ internal tide constituent.

model errors in most of the global ocean. Like other diurnal constituents, the Luzon Strait is the dominant generation source
of mode-1 $Q_1$ beams. The Luzon Strait radiates mode-1 $Q_1$ beams eastward into the western Pacific and westward into the
South China Sea. One can see strong $Q_1$ internal tides in the Indonesian Seas. This feature is consistent with our earlier model
evaluation (Section 4) that $Q_1$ can cause variance reduction in the Indonesian Sea and around the Luzon Strait (Figure 4). The
mode-1 $Q_1$ amplitudes are about 1/5 of the mode-1 $O_1$ amplitudes.

## 6 Regional Examples

The decomposed components show numerous long-range internal tidal beams, which contain important information on their
generation and propagation. To better extract the information, one should study them region by region or beam by beam. This
section showcases two examples: the semidiurnal beams off the Amazon shelf and the diurnal beams in the Arabian Sea.

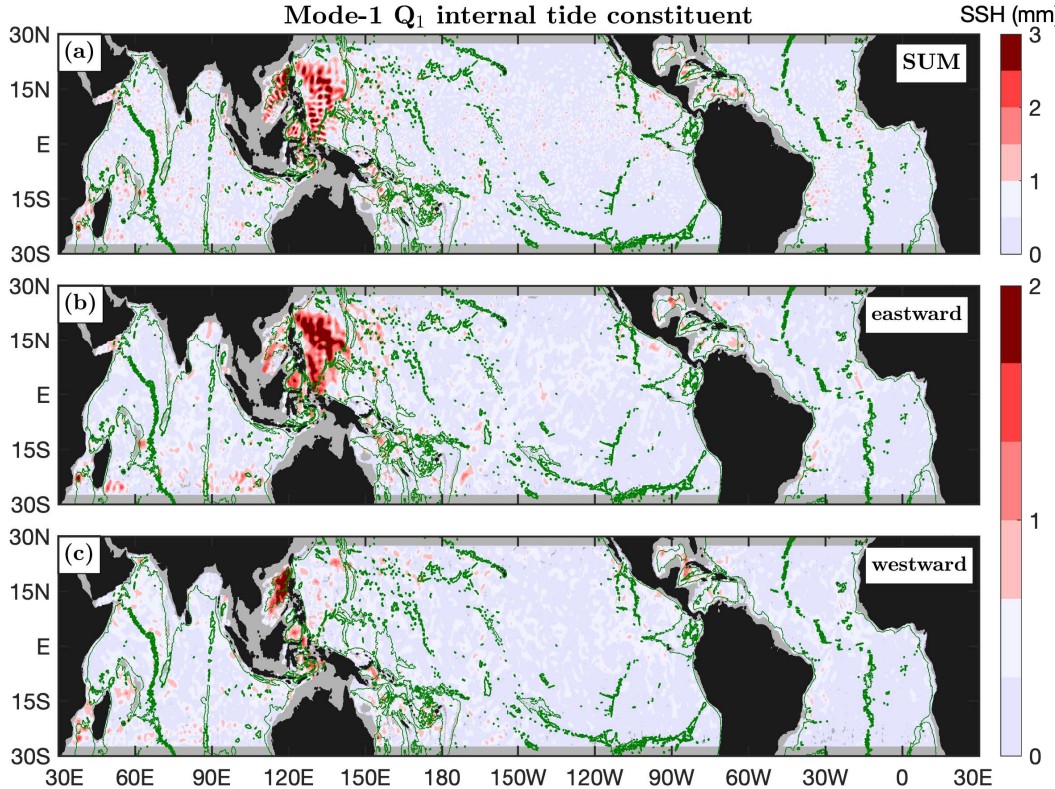

**Figure 18.** As in Figure 13 but for the mode-1 $Q_1$ internal tide constituent.

## 6.1 Semidiurnal internal tidal beams off the Amazon shelf

The Amazon shelf ($60°$ W–$35°$ W, $5°$ S–$20°$ N) is chosen as an example because of its strong internal tides and frequent occurrence of internal solitary waves (Magalhaes et al., 2016; Barbot et al., 2021; Egbert and Erofeeva, 2021; Tchilibou et al., 2022; de Macedo et al., 2023; Assene et al., 2024). However, previous studies mainly focused on the dominant $M_2$ internal tides. This figure shows internal tidal beams for 6 semidiurnal constituents including mode-1 $M_2$, $S_2$, $N_2$, and $K_2$ and mode-2 $M_2$ and $S_2$. Figure 19 shows the six internal tide constituents in the region. Each constituent is divided into northeastward and southwestward components. The northeastward component contains the largest waves at each grid point with propagation direction ranging $-45°$–$135°$, and the southwestward component contains (Figure S15) the largest waves ranging $135°$–$315°$. Note that the six constituents are shown using the same colormap but different ranges, because their amplitudes may vary by an order of magnitude. For all constituents, internal tides with low amplitudes are shown in light blue. Meanwhile, the $0°$ co-phase lines can also highlight internal tidal beams.

The results reveal outstanding internal tidal beams off the Amazon shelf. The four mode-1 constituents ($M_2$, $S_2$, $N_2$, and $K_2$) have similar spatial patterns. Among them, $M_2$, $S_2$, and $N_2$ show 6 isolated internal tidal beams propagating northeastward

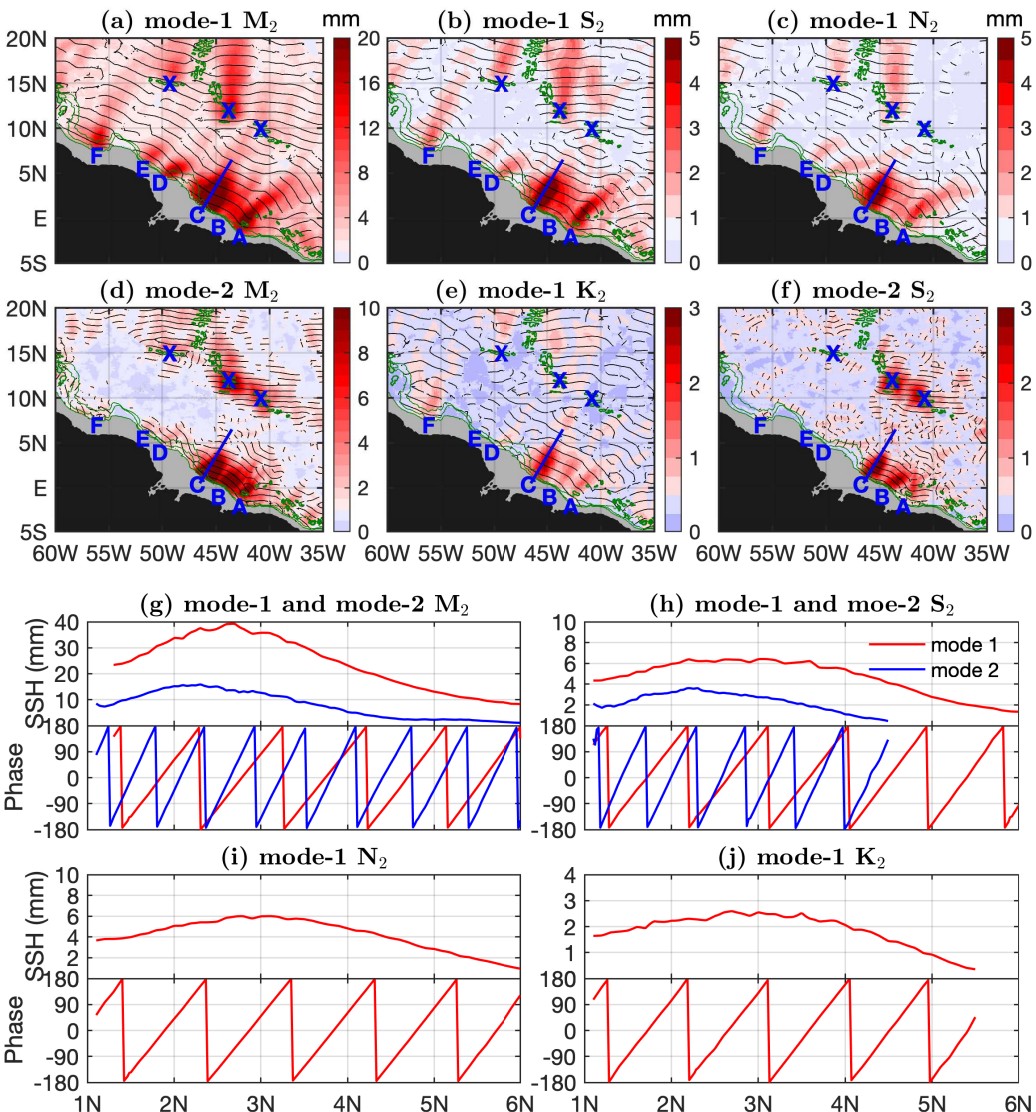

**Figure 19.** Northeastward semidiurnal internal tidal beams off the Amazon shelf. (a)-(f) Northeastward internal tide components $(-45°–135°)$. Black lines indicate the $0°$ co-phase charts. Green contours indicate the 1000-, 2000-, and 3000-m isobaths. Each constituent has up to 6 isolated beams off the Amazon shelf (labeled A–F). Blue lines highlight the strongest beams generated at the mouth of the Amazon River. Beams generated at the Mid-Atlantic Ridge are marked. (g)-(j) Along-beam amplitudes and phases. Note that the largest amplitudes are usually one wavelength away. It is likely an artitificial features caused by our large spatial windows for fitting plane waves and flurier 2D filtering.

from the Amazon shelf. They are labeled A–F (Figure 19). The $K_2$ constituent shows only 5 beams A–D and F. Its beam E is missing, most likely because the much weaker $K_2$ (lower than 2 mm) is affected by model errors. The co-phase lines are





parallel to one another and across the internal tidal beams. For these constituents, the strongest beams (blue lines) radiate from the mouth of the Amazon River. The along-beam amplitudes and phases are shown in Figures 19g–j. One can see that their amplitudes are smooth and their phases increase linearly with propagation. These features suggest that these beams are successfully extracted. Otherwise, they would show standing-wave features (half-wavelength fluctuations). Along the strongest beams, their amplitudes range from 40 mm for $M_2$ to 3 mm for $K_2$. The strongest $M_2$ beam can be tracked for about 700 km from the Amazon shelf to the Mid-Atlantic Ridge. However, the $S_2$, $N_2$, and $K_2$ beams disappear soon, likely because their lower amplitudes are masked by model errors. Our satellite observations reveal some beams generated over the Mid-Atlantic Ridge (Figure 19).

Our satellite observations are generally consistent with previous numerical simulations, which though mainly focused on $M_2$ internal tides. Isolated $M_2$ internal tidal beams are observed by Tchilibou et al. (2022, Figure 7) and Assene et al. (2024, Figure 2). For example, Tchilibou et al. (2022) identified 6 strong internal tidal beams off the Amazon shelf. Their generation sites are consistent with our satellite observations. One exception is that they suggest the strong beam (Figure 19a, beam C) is composed of two beams. Previous studies found that the strong beams overlap with internal solitary waves very well, in that internal solitary waves are generated by nonlinear internal tides. Tchilibou et al. (2022) showed the temporal variation of internal tides in two contrasting time periods September–November and March–June. Likewise, $M_2$ internal tide maps have been constructed using four seasonal subsets (Zhao, 2021) and decadal maps (Zhao, 2023a). It is interesting to compare the satellite observations and numerical models in future research. The 6 beams are spatially collocated for the four constituents, suggesting that they are generated by the same topographic features. Their superposition will lead to temporal variation of semidiurnal internal tides.

Our multiwave decomposed results also reveal mode-2 $M_2$ and $S_2$ internal tidal beams (Figures 19d, f). Both constituents show three beams A–C. There are no outstanding mode-2 beams at E–G. Figures 19g and 19h show the along-beam amplitudes and phases for mode-2 $M_2$ and $S_2$ (blue lines). It shows that the mode-2 $M_2$ amplitudes may be up to 15 mm, and the mode-2 $S_2$ amplitudes are up to 4 mm. Similarly, the mode-2 beams have large and smooth amplitudes and their phases increase linearly with propagation. The mode-2 $M_2$ constituent is larger than mode-1 $S_2$ and $N_2$ constituents; therefore, the missing beams D–F are not likely because of its weak signals. Rather, it is likely that the local specific topography favors the generation of mode-1 constituents, but not mode-2 constituents.

## 6.2 Diurnal beams in the Arabian Sea

We next give an example using the westward diurnal internal tidal beams in the Arabian Sea. In this region, the dominant $M_2$ internal tides have been well documented in previous studies (Kaur et al., 2024; Ma et al., 2021; Subeesh and Unnikrishnan, 2016; Subeesh et al., 2021; Zhao, 2021). Fortunately, our new model reveals both $O_1$ and $K_1$ internal tidal beams. Both mode-1 and mode-2 components are examined. Figure 20 shows the the four diurnal internal tide constituents. Each constituent is divided into eastward and westward component. The eastward component contains the largest waves at each grid point ranging $-90°$–$90°$ (Figure S16), and the westward component contains the largest waves ranging $90°$–$270°$. Overlain are the $0°$ and



Earth System
Science
Data

180° co-phase lines. The intervals between two neighboring co-phase lines are half one wavelength. Topographic features are indicated by the 1000-, 2000-, and 3000-m isobaths.

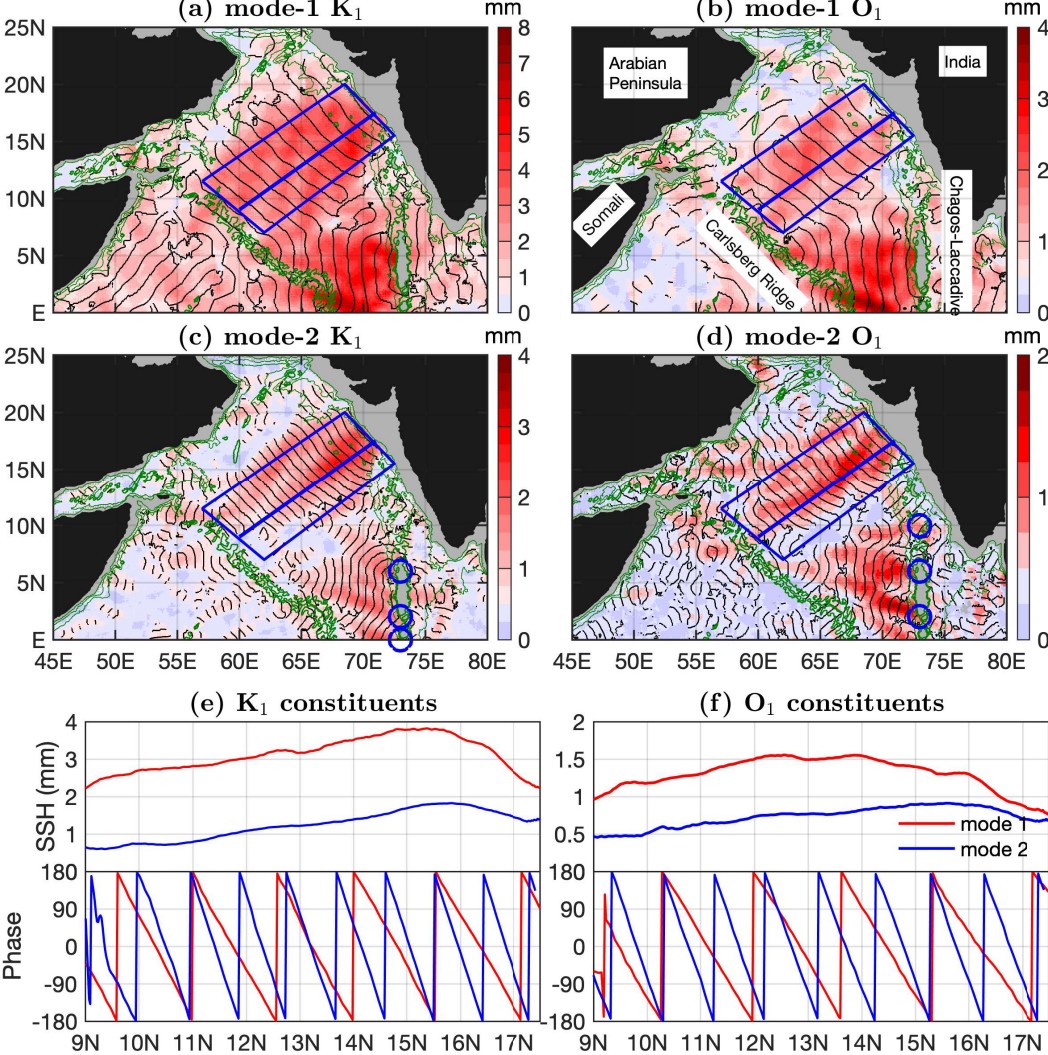

**Figure 20.** Westward diurnal internal tidal beams in the Arabian Sea. (a)-(d) Westward internal tide components (90°–270°). Black lines indicate the 0° and 180° co-phase charts. Internal tides with amplitudes lower than 0.5 mm are shown in light blue. Green contours indicate the 1000-, 2000-, and 3000-m isobaths. Blue circles mark some isolated mode-2 beams at the Chagos–Laccadive Ridge. (e, f) Along-beam amplitudes (averaged across the beam) and phases (along the central line).

Figure 20 shows two outstanding internal tidal beams for each of the four diurnal constituents. One set of beams originate at the Indian western shelf along 15°–20° N. The beams propagate toward 210° for about 2000 km long from the Indian coast to the Carlsberg Ridge. It takes about 6 (10) repeat cycles for mode-1 (mode-2) $K_1$ and $O_1$ internal tides to travel cross



the distance. Another set of beams radiate from the Chagos–Laccadive Ridge (along 73° E) and travel over 1000 km before disappearance. The mode-2 beams from the Chagos–Laccadive Ridge appear to be composed of several narrow beams (blue circles). Note that the two generation sites also generate strong $M_2$ internal tides (Subeesh et al., 2021; Ma et al., 2021). For comparison, the diurnal internal tidal beams are wider than semidiurnal beams. The southwestward mode-2 beams from the Indian western shelf are about 500 km wide.

Figures 20e and 20f show along-beam amplitudes and phases for the four diurnal constituents. Their smooth amplitudes and linear-increasing phases suggest that the beams have been successfully separated. The diurnal internal tidal beams have small amplitudes, ranging from 4 mm for mode-1 $K_1$ to 1 mm for mode-2 $O_1$ internal tides. This feature again suggests that our new internal tide model has low model errors. However, the even weaker $P_1$ and $Q_1$ constituents (lower than 1 mm) do not show well-defined internal tidal beams (Figures 17 and 18). The regional maps reveal isolated beams and pinpoint generation sites. Such an examination can be conducted constituent by constituent and region by region.

## 7    Summary

We have observed global internal tides by applying our newly improved mapping technique to 30 years of satellite altimetry data from 1993 to 2022 (Figure 1). Our new mapping technique consists of two rounds of plane wave analysis with a spatial bandpass filter in between (Zhao, 2022a, b). The new data record is 120 satellite years long, including all the nadir altimetry data made by 15 altimetry mission. Our main findings and conclusions are summarized as follows:

1. We have constructed a new internal tide model that contains 12 internal tide constituents (Figure 2): 8 mode-1 constituents ($M_2$, $S_2$, $N_2$, $K_2$, $K_1$, $O_1$, $P_1$, and $Q_1$) and 4 mode-2 constituents ($M_2$, $S_2$, $K_1$, and $O_1$). The combination of 30-year data record and new mapping technique significantly suppresses model errors down to lower than 1 mm (Figure 3), which makes it possible to map weak mode-2 and minor internal tide constituents.

2. We have decomposed the multiconstituent multimodal multidirectional internal tide field into a series of simple plane waves. In frequency, eight principal constituents are extracted. In the vertical direction, two lowest baroclinic modes are extracted for the four major constituents. In the horizontal direction, each internal tide constituent is decomposed into 5 waves of arbitrary directions at each grid point. All together, the internal tide field into 60 plane waves at each grid point (Figures S3–S14). The multiwave decomposition presents a new view of the global internal tide field.

3. We have validated the new internal tide model using independent altimetry data in 2023 (Figure 4). On global average, ten constituents (but for $K_2$ and $Q_1$) can cause positive variance reductions, because these constituents are sufficiently strong to overcome model errors. The weak $K_2$ and $Q_1$ constituents can overcome model errors near their sources in the western Pacific Ocean. ZHAO30yr performs much better than ZHAO20yr, a model developed using 20 years of altimetry data by an obsolete mapping technique.

4. The 12 internal tide constituents are examined in the new model. The decomposed components reveal numerous well-defined long-range internal tidal beams (Figures 7–18). These beams are associated with notable topographic features.

For all constituents, mode-2 beams are shorter and narrower than corresponding mode-1 beams. The decomposed components contain rich information on their generation, propagation, and dissipation.

5. We have studied in detail the semidiurnal internal tidal beams off the Amazon shelf (Figure 19). For mode-1 constituents ($M_2$, $S_2$, $N_2$, and $K_2$), six isolated beams propagating northeastward off the Amazon shelf recognized. Along the strongest beams from the mouth of the Amazon River, their amplitudes range from 40 mm for $M_2$ to 3 mm for $K_2$. For mode-2 constituents ($M_2$ and $S_2$), there are three isolated beams off the Amazon shelf. Our satellite observations are generally consistent with previous numerical simulations.

6. We have studied in detail the diurnal internal tidal beams in the Arabian Sea (Figure 20). For both mode-1 and mode-2 $O_1$ and $K_1$ constituents, westward beams radiate from the Indian western shelf and the Chagos–Laccadive Ridge. Their along-beam amplitudes range from 4 mm for mode-1 $K_1$ to 1 mm for mode-2 $O_1$. Isolated mode-2 $O_1$ and $K_1$ beams are observed at the Chagos–Laccadive Ridge.

   7. We have revealed that the global internal tide field is composed of numerous isolated internal tidal beams, which dis-
tribute across the ocean. One can acquire important information on their generation, propagation, and dissipation by tracking these long-range internal tidal beams.

There is large room for future model improvements. First, the model can be further refined by optimizing empirical mapping parameters. Due to the complexity and inhomogeneity of the global internal tide field, the parameters should be optimized region by region and constituent by constituent. Second, previous studies have shown that internal tides are subject to significant
variations over seasonal, inter-annual, and decadal timescales (Zhao, 2021, 2022a, 2023a). Future internal tide models should be of time-variable amplitudes and phases.

## 8   Data availability

http://doi.org/10.6084/m9.figshare.28078523 (Zhao, 2024).

*Author contributions.* The study was completed by the sole author.

*Competing interests.* The author has declared that there are no competing interests.

*Acknowledgements.* The model development was supported by the National Aeronautics and Space Administration (NASA) via projects NNX17AH57G and the National Science Foundation (NSF) via project OCE1947592.





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
