# Peer review of "A New-Generation Internal Tide Model Based on 30 Years of Satellite Sea Surface Height Measurements"

_Earth System Science Data, 2024_

## Referee Comment (RC1)

**Review on "A New-Generation Internal Tide Model Based on 30 Years of Satellite Sea Surface Height Measurements"**

Zhongxiang Zhao

January 2025

**1 General comments**

The article and data set describe a model representing the local sea surface height (SSH) signature of the spatio-temporally coherent internal tidal field in terms of a superposition of propagating plane waves (for given tidal constituent and baroclinic mode). Most notably, this new model (ZHAO30yr) incorporates an unprecedented collection of tidal constituents and vertical modes, mapped at higher resolution than in previous work by the author, and includes error estimates. The performance of the model is evaluated using independent altimetry data (using a variance reduction statistics) showing significant improvement over previous work by the author. These major advances were made possible mainly by the refinement of the authors' mapping technique over the years, and the inclusion of input data from more altimetry missions. In other words, ZHAO30yr is the culmination of the author's leading research in mapping internal tides sea surface signature from altimetry data. Yet, at the same time, ZHAO30yr can be seen as a new starting point for the many potential improvements suggested in this study.

One important feature of the article is the description of the spatial variability in a single component of the decomposed multiwave field. The author investigates two decompositions of the multiwave field: (i) by extracting the (locally) five largest propagating plane waves, and (ii) by selecting the largest of the previously extracted waves propagating in a given directional range (generally half circles). While I am convinced that the reconstructed multiwave field is a reasonable

representation of the underlying internal tidal SSH field (although all of the five fitted waves might not be significant everywhere), I have more trouble understanding the analysis of individual plane wave components as performed in the present study.

The article claims that such decompositions can be used to disentangle wave interference and further isolate internal tidal beams. This is an apparent contradiction, in that these beams are commonly viewed as an interference pattern resulting from the superposition of waves radiating from distinct sources (Rainville et al., 2010, see in particular their Fig. 9). The confusion is entertained by the claim that individual components of the multiwave decomposed SSH are not affected by multiwave interference (l215 in the article). On the contrary, interference patterns (with nodes and antinodes separated by half a wavelength) are clearly visible in the decomposed product (see Figs. 3 and 5 in Detailed comments). The above apparent inconsistencies make it challenging for me to understand what the spatial variability in a single component of the multiwave decomposed field actually represents (be it the locally largest component or the locally largest one in a directional range).

In any event, I think the article lacks a clear definition of what is considered as "internal tidal beams". Then, the results based on a single component of the decomposed multiwave field (sections 5 and 6) should be analyzed in a way that is consistent with the latter definition.

Another comment is that the general tone of the article is quite enthusiastic (perhaps too much so), somewhat obscuring the limitations of the model. As a result, non-specialist users of the data might end up being mislead. I would like to see the important limitations and shortcomings of the model explicitly stated where fit:

- ZHAO30yr only extracts the spatio-temporally coherent fraction of the ITs. Hence, the decay in amplitude of an observed internal tidal beam with distance from its source region cannot be directly related to dissipation (e.g. Zaron, 2015; Buijsman et al., 2017; Geoffroy & Nycander, 2022).

- ZHAO30yr cannot distinguish between modal ITs originating from barotropic-to-baroclinic conversion and the waves resulting from scattering of different modes (e.g. by interactions with topography).

- Physically, an internal tidal beam is the spatial expression of the interference of multiple waves radiating from distinct sources (Rainville et al., 2010). Thus, one cannot directly "pinpoint" the source locations of an observed beam. Moreover, it is not straightforward to gain information on the generation of the individual baroclinic waves forming a beam.

**2 ESSD's review criteria**

- *Read the manuscript: are the data and methods presented new? Is there any potential of the data being useful in the future? Are methods and materials described in sufficient detail? Are any references/citations to other data sets or articles missing or inappropriate? Is the article itself appropriate to support the publication of a data set?*

  The methods used in this publication have been used in previous studies (e.g. Zhao et al., 2016, 2019; Zhao, 2021, 2022, 2023, with an evolution throughout the years). I fail to see a significant difference in the methods compared with Zhao (2023), apart from the inclusion of more tidal constituents and vertical modes. Another important advance is the use of more altimetry missions as input data to build the model. The data will very likely be useful in the future. The methods are described in details, and more information can be found in the previously mentioned studies. However, I have troubles understanding a few key points in the methods and presentation of the results (see General comments). These may be misunderstandings, or simple issues with the terminology used in the paper. Nonetheless, I think these points should be clarified before publication. The paper generally does a good job citing appropriate references and other datasets. It could, however, do a better job at pointing at the (important) limitations of the model.

- *Check the data quality: is the data set accessible via the given identifier? Is the data set complete? Are error estimates and sources of errors given (and discussed in the article)? Are the accuracy, calibration, processing, etc. state of the art? Are common standards used for comparison? Is the data set significant – unique, useful, and complete?*

  The data is straightforward to access and use. The data set lacks error estimates for a single plane wave, as well as the mask for regions of large mesoscale variability where the model

likely fails (as explained in the article). In particular, each of the fitted plane wave should be compared with the local error estimate for a single wave ; as it is likely that not all of the five fitted waves are significant at any given location. The model, i.e. the sum of the five fitted plane waves for a given tidal constituent and baroclinic mode, is validated using a variance reduction statistics, and it is compared to previous iterations showing significant improvement.

- *Consider article and data set: are there any inconsistencies within these, implausible assertions or data, or noticeable problems which would suggest the data are erroneous (or worse). If possible, apply tests (e.g. statistics). Unusual formats or other circumstances which impede such tests in your discipline may raise suspicion. Is the data set itself of high quality?*

As stated above, I think some key points in the methods and presentation of the results should be clarified. Apart from these, the article properly describes the data set.

- *Check the presentation quality: is the data set usable in its current format and size? Are the formal metadata appropriate? Check the publication: is the length of the article appropriate? Is the overall structure of the article well structured and clear? Is the language consistent and precise? Are mathematical formulae, symbols, abbreviations, and units correctly defined and used? Are figures and tables correct and of high quality? Is the data set publication, as submitted, of high quality?*

The published data set is usable (but still lacks the single wave error estimates and mask for the regions of large mesoscale variability), and the metadata is appropriate. The publication is somewhat long but well written and, overall, of good quality.

- *Finally: By reading the article and downloading the data set, would you be able to understand and (re-)use the data set in the future?*

Yes, but at this stage only for the reconstructed multiwave field (i.e. the sum of the locally fitted plane waves for a given tidal constituent and baroclinic mode). I don't understand the author's interpretation of the spatial variability in a single wave component.

**3 Detailed comments**

**Introduction**

l25-27: It would be fair to cite Colosi and Munk (2006) as well.

l32: To be complete, you could mention semi-analytical methods as well ; especially the recent anisotropic estimates in Pollmann and Nycander (2023) and Geoffroy et al. (2025), which are particularly suited for a comparison with ZHAO30yr.

Paragraph 34-45: Please make it clear that ZHAO30yr only estimates the spatio-temporally coherent ITs.

l51: The "new" mapping procedure is confusing, this seems to be the same procedure as in Zhao (2022).

l56: Same "newly improved" is confusing.

l57-59: This is confusing. Internal tidal beams are generally considered as the expression of the interference pattern between plane waves originating from different sources (Rainville et al., 2010).

l60: ZHAO30yr only sees the coherent fraction of the IT field, thus any information about dissipation would be non-trivial to retrieve (the decay of a beam with propagation distance in ZHAO30yr can also be attributed to a loss of coherence). ZHAO30yr also cannot distinguish between modal internal tides (ITs) generated by the conversion of surface tides and those resulting from the scattering of different modes. This should be clearly stated.

l62: "background internal tides" is unclear, perhaps mention that this will be explained later on.

l63: Errors are lower than 1 mm on a global average.

l67: "published in Carrere et al. 2021" → "mentioned in Carrere et al. 2021".

l70: "are previously masked" → "were previously masked".

l78: "Appendix ??". The appendix appears to be missing (it is not included in the manuscript and supplementary material.) This sentence is unclear at this stage.

l84: "including all altimetry [. . . ]" already written at the beginning of the paragraph.

**Section 2.1**

l89: "Direct" → "Directly"

l95: add "leaked" internal tide signals ?

l98: Taking into account the spatial variations in $\lambda$, depending on location one can have $\lambda$ larger than the cutoff wavelength for the mesoscale field. Hence, this low-pass filtering of the mesoscale estimate does not consistently remove the leaked IT variance (resulting in a biased low IT variance in the results). Applying another (lowpass) filter in frequency might mitigate this, as suggested by Zaron and Ray (2018). In this paper, they also note that the diurnal ITs are more closely entangled with mesoscale dynamics (compared with semidiurnal ITs), so that you may have more leakage from mesoscale activity in your diurnal products.

**Section 2.2**

l106: Can you demonstrate that these 2 constituent pairs are well separated in ZHAO30yr ?

l113: Do you use the rigid-lid approximation, namely $\Phi(z = 0) = 0$ ? If not, the problem you solve is not trivial to me. Could you mention the boundary condition you use at the surface and how you normalize the eigenfunctions (and provide a reference) ? One example is Kelly (2016).

l115: I think the equation should read $\lambda_n = \omega c_n / \sqrt{\omega^2 - f^2}$, see for instance Eq. (A5) in Zhao et al. (2016).

l118: What about the temporal variation in $\lambda$ over seasonal, and inter-annual timescales ? It would be worth comparing this time variability with the bandpass width of the 2D filter.

l123: Can you justify that such close constituents are well separated in ZHAO30yr ?

**Section 3**

The method extracts 5 independent propagating waves with a target $\lambda$ and $\omega$. Signatures of other processes than ITs (sharing a similar $\lambda$ and $\omega$) might be picked up: the mesoscale correction is not perfect, and there may be other waves (e.g. westward propagating tropical instability waves as mentioned in Zhao (2019)). The main assumption here is that only ITs can show the observed spatio-temporal coherent structures over the 30 years of data. Do I get this right ? (If yes, I think this could be made clearer).

l131: Truncating the 2D spectrum (i.e. multiplying it by a rectangular window) will result in ringing, you might want to use a window with a smoother roll-off.

**Section 3.1**

Could you mention how do you bin the altimetry data in time ? What tolerance do you use to consider two measures collocated in time ? What would a typical snapshot look like in terms of point density within a fitting window ?

Eq. (2): If I am not mistaking, this equation represents waves emanating from an infinitely long line source. This might be a good representation far away from the actual sources (of finite length), but not so much close to generation sites (where I would expect the waves to spread cylindrically). See Fig 1 for an example mode-1 M2 plane wave (in a correspondingly sized fitting window) using values from Table 1 in the article. Could the extent of the spatial window (smaller than a wavelength here) negatively impact the fit ?

[Figure]

Figure 1: Mode-1 M2 plane wave as per Eq. (2) for a single $m$ in a 120 km fitting window using the mean parameters from Table 1.

l148-154: It took me several reads to understand this procedure, this could be rewritten to make it clearer. The wording is confusing: "target internal tidal wave" is a sum of "5 internal tidal waves", perhaps the latter could be simply called plane waves. Also, you could explicitly state at what stage $A_m$, $\phi_m$, and $\theta_m$ are obtained.

l150: Is the least squares fit performed in each compass direction ? If yes, it should be mentioned in the following sentence for the sake of clarity.

Each plane wave fit should be compared with the estimated error (using "background" ITs or some other error estimate). Fits resulting in an amplitude below the estimated error should be discarded (as in Zhao et al. (2016)). Also, the error associated with a single plane wave should be included in the dataset.

**Section 3.2**

l159: "in one overlapping 850 by 850 km" → "in overlapping 850 by 850 km windows" ?

l161: "is nontidal errors' → " is considered as noise" ?

l162: Is truncation done with a rectangular window ? Does this introduce ringing ? If yes, did you evaluate/mitigate it ?

The truncated spectra may still contain some background noise, not only the tidal peaks. Perhaps this background noise level could be estimated from the variance just outside the theoretical wavenumber range.

**Section 3.3**

l168: "S2 has an tidal aliasing" → "S2 has an aliasing issue"

l172: "wavenumber-frequency filtering": I understand that you only filter in wavenumber space (but you select the frequency during the least square fit).

l181: Could you point to the section where these results are presented ?

l184-185: I see that bandpass width times $\lambda$ overlaps for mode-1 and mode-2 waves. But it is not obvious to me why mode 1 can affect mode 2 but not the other way around. Could you elaborate ?

**Section 3.4**

l193: define major/minor constituents.

l200: This light blue color code for amplitudes lower than 1 mm does not reflect the spatial variability of the error and may be misleading. Can you think of a better representation including

the geographical variability of the error ? I realize discarding the results where the amplitude is smaller than the error may not be a very good solution neither since a large fraction of the global field would be masked (see Fig. 2). Perhaps you could discard data based on the variance reduction statistics computed in section 4.2 ?

[Figure]

Figure 2: Mode-1 M2 internal tide amplitude (sum of the 5 fitted plane waves), masked where the amplitude is smaller than the provided error for mode-1 semidiurnal ITs.

l201: Consider publishing your mask excluding the regions of large mesoscale variability alongside the data.

l202: "have largest amplitudes" → "have the largest amplitudes"

l203: same for "lowest"

**Section 3.5**

l210: "5 waves of arbitrary directions" → "arbitrary" may not be adapted here. Perhaps rewrite in something like "the 5 most prominent plane waves with empirically determined directions".

l215: I disagree, interference patterns with half-wavelength fluctuations are clearly visible in the

decomposed wave field. In Fig. 3, I attach detail maps of an area between Hawaii and the Aleutian Islands displaying a marked interference pattern in the North-South direction in the mode-1 M2 first (largest) plane wave (IW1). This can also be seen in your maps in supplementary material for mode-1 M2, N2 (and S2 to a lesser extent). Diurnal constituents seem affected as well (in particular, IW1 in mode-1 K1 west of India shows similar fluctuations in the East-West direction, see Fig. 5).

[Figure]

Figure 3: Left panel: Amplitude of the first plane wave for the mode-1 M2 internal tide, an interference pattern is clearly visible (horizontal darker and lighter rays with wavelength of about $\lambda/2$; Right panel: Zoom in the interference pattern.

l216: "new features that are previously masked" $\rightarrow$ "new features that were previously masked"

l215-224: This is confusing. Internal tidal beams are an interference pattern, I don't understand how you could locally represent a beam as a single plane wave as defined in Eq. (2) (for a single $m$, see Fig. 1).

l222: Do you mean the decomposed multiwave field ?

l223: Yes, looking at a decomposition by propagation direction might be a more consistent approach. Still, I don't quite understand how you could assimilate the amplitude of the largest plane wave in a directional range to a beam.

figure 2: Hard to read, panels are too small.

**Section 4.1**

l238: The corresponding mask is not included with the published data.

l240: "reliably" might be an overstatement, especially at the global scale (as you discuss in section 4.2).

Figure 3: The statistics seem to be computed including regions of large mesoscale variability, discarding these regions (as stated in the text) will improve the results.

**Section 4.2**

l244: "(amplitudes and phases)" $\rightarrow$ "(amplitude and phase of each constituent)"

l246-247: should read $T_n$, $f_n(t)$, $u_n(t)$, $A_n(x, y)$, and $phi_n(x, y)$.

l249: delete "Here" or $\rightarrow$ "Here,"

l254: "is the variance difference" $\rightarrow$ "is the difference in variance computed"

l256-257: confusing, rephrase.

l262: "real internal tides" $\rightarrow$ "predicted internal tides"

**Section 4.3**

figure 4: Hard to read, panels are too small.

l292: internal tidal beams are presents in all products, rephrase.

**Section 5.1**

l306-307: as well as in studies based on in situ observations, it would be fair to mention a few of these as well.

l310: "But" → "Note"

l312-313: These single-wave error estimates are not published with the data.

l316-318: unclear, rephrase.

l325: "Section 6.1 has shown" → "Section 6.1 shows"

Figure 8: delete "[too many circles]" in caption

l329: This is unclear. The detection of mode-2 M2 ITs with altimetry might be strongly influenced by the stratification structure, but not so much the underlying true wave field (other parameters set the generation, e.g. the topographic wavelength (Llewellyn Smith & Young, 2002)). You should also mention that mode-2 waves tend to be more incoherent.

l342: "The mode-1 and mode-2 beams" → "The detected mode-1 and mode-2 beams". Item (1), and perhaps item (2), can be partly explained by mode-2 waves being more incoherent than mode-1 waves (e.g. Rainville & Pinkel, 2006). Also, a possible explanation for item (2) could be that the interference of the shorter mode-2 wavelengths results in narrower beams compared with mode 1.

l346: item (3), linear theory is instructive here: it predicts very small conversion where the topographic length scale does not match the wavelength of the wave (Llewellyn Smith & Young, 2002). In other words, surface tide conversion into mode 2 is maximized at topographic features with length scales close to the shorter mode-2 wavelength.

l347: Again this is confusing, the beams are not generated at a point location, they are the result of the interference of multiple (line) sources.

l349: This is wrong, mid-ocean ridges also generate significant mode-2 M2 ITs. See global map of the M2 mode-2 generation from linear theory in Geoffroy et al. (2024).

l350-351: The model also have important limitations that should be clearly mentioned (only coherent ITs, interference pattern present in the amplitude of the individual fitted plane waves, separation of barotropic conversion and scattering processes).

**Section 5.2**

l361: "; however," → ", however,"

l365: Could (1) be explained by weaker S2 barotropic tide as in (2) ?

**Section 5.3**

l382: "summed filed" → "summed field"

l382: "beams are masked by multiwave interference", rewrite.

**Section 5.5**

l410-411: eastern/western hemisphere relative to 120 deg W ? (State reference longitude to avoid confusion.)

l422: "Figure 13b, circle", which circle ?

l423: "Figure 13b, circle", which circles ? (Use different colors to distinguish from the Mona Passage.)

l435: "pinpoint" is confusing.

l437: Unclear, rewrite.

l438: Already stated above, delete sentence (and typo in "K2").

Fig. 14, 15, 16: mention blue circles in caption ?

**Section 5.6**

l441: "but that the K1" delete "that" ?

l446: "Another similar feature" add "with mode-1 K1".

**Section 6.1**

l494: The beams are actually clearly visible from the summed plane wave product (see Fig. 4). The directional decomposition offers a cleaner view of the beams.

Figure 19: Title of (h), typo in "mode-2 S2".

Last line of caption, "flurier 2D filtering" → "Fourier 2D filtering".

Last line of caption: On the contrary, this last observation seems consistent with the literature: The maximum in SSH occurring at some distance away from the generation site(s) has been observed and modeled in previous studies. This has been related to the first surface bouncing of the tidal

[Figure]

Figure 4: Mode-1 M2 internal tide amplitude (sum of the 5 fitted plane waves) close the Amazon mouth.

beam (Merrifield & Holloway, 2002; Carter et al., 2008). This could also be interpreted as the expression of interference between multiple plane waves (Rainville et al., 2010)).

l501: Could you explain ? Is this because of the superposition with waves (reflecting or originating) from the Mid-Atlantic Ridge ?

l505: It would be good to remind that the decay in amplitude of a spotted beam with propagation distance is partly due to the growing fraction of incoherent ITs (not being taken into account in your model).

l514: Since you mention possible comparisons, the semi-analytical estimates from Pollmann and Nycander (2023), and Geoffroy et al. (2025) are also particularly suited for a comparison with ZHAO30yr.

l515: "temporal variation of semidiurnal ITs", unclear. do you refere to the beat originating

from the sum of the constituents close to the M2 frequency ?

**Section 6.2**

l528: Typo "the the four diurnal". Also, you rather show the two first modes of the two diurnal constituents O1 and K1.

l533-535: This beam is actually solely due to IW1 in the plane wave decomposed field (see Fig. 5). Furthermore, an interference pattern with half-wavelength fluctuations is clearly visible within the area of the beam. Hence, Fig. 20 a is challenging to interpret. Nonetheless, the evolution of phase and amplitude along the putative trajectory look convincing.

[Figure]

Figure 5: Left panel: Amplitude of the first plane wave for the mode-1 K1 internal tide (IW1), an interference pattern is clearly visible (both vertical and horizontal darker and lighter rays with wavelength of about $\lambda/2$; Right panel: IW1 minus the westward component of the directionally decomposed internal tidal field.

l535: "repeat cycles", unclear. Do you refer to a satellite mission in particular ?

l537: "beams [...] composed of narrow beams" is unclear, rephrase.

l541: That's the 2 first modes of 2 constituents.

l545: "pinpoint", again, not exactly.

**Summary**

l549: delete "new", it has been published already.

l554: "down to lower than 1mm" add "on a global average".

l559: "the internal tidal field into" → "the internal tidal field is decomposed into"

l560: I would be more nuanced on the results of the multiwave decomposition (specifically for the presence of interference patterns in a single plane wave field).

l564: reference for ZHAO20yr ?

l569: To be nuanced: the beam generation does not occur at a single source point (Rainville et al., 2010). Moerover, incoherence acts to decrease the detected beams' amplitude with propagation distance, this is not straightforward to disentangle from dissipation.

l571: "off the Amazon shelf recognized" → "off the Amazon shelf have been recognized"

l573: "For mode-2 constituents" → "For M2 and S2 mode-2 internal tides"

l575-577: The mode-1 K1 beam you identified west of India corresponds to the first (largest) fitted plane waves in this region. This is inconsistent with the commonly accepted view of a beam resulting from the interference between multiple plane waves emanating from distinct generations sites.

l579: The beams have already been identified in previous studies. Also, you already stated this in point 4.

**References**

Buijsman, M. C., Arbic, B. K., Richman, J. G., Shriver, J. F., Wallcraft, A. J., & Zamudio, L. (2017). Semidiurnal internal tide incoherence in the equatorial pacific. *Journal of Geophysical Research: Oceans*, *122*(7), 5286-5305. Retrieved from `https://agupubs.onlinelibrary.wiley.com/doi/abs/10.1002/2016JC012590` doi: https://doi.org/10.1002/2016JC012590

Carter, G. S., Merrifield, M. A., Becker, J. M., Katsumata, K., Gregg, M. C., Luther, D. S., ... Firing, Y. L. (2008). Energetics of m2 barotropic-to-baroclinic tidal conversion at the hawaiian islands. *Journal of Physical Oceanography*, *38*(10), 2205 - 2223. Retrieved from

https://journals.ametsoc.org/view/journals/phoc/38/10/2008jpo3860.1.xml    doi:
10.1175/2008JPO3860.1

Colosi, J. A., & Munk, W. (2006). Tales of the venerable honolulu tide gauge. *Journal of Physical Oceanography*, *36*(6), 967 - 996. Retrieved from https://journals.ametsoc.org/view/journals/phoc/36/6/jpo2876.1.xml    doi: 10.1175/JPO2876.1

Geoffroy, G., Kelly, S. M., & Nycander, J. (2025). Tidal conversion into vertical normal modes by continental margins. *Geophysical Research Letters*, *52*(1), e2024GL112865. Retrieved from https://agupubs.onlinelibrary.wiley.com/doi/abs/10.1029/2024GL112865 (e2024GL112865 2024GL112865) doi: https://doi.org/10.1029/2024GL112865

Geoffroy, G., & Nycander, J. (2022). Global mapping of the nonstationary semidiurnal internal tide using argo data. *Journal of Geophysical Research: Oceans*, *127*(4), e2021JC018283. doi: 10.1029/2021JC018283

Geoffroy, G., Pollmann, F., & Nycander, J. (2024). Tidal conversion into vertical normal modes by near-critical topography. *Journal of Physical Oceanography*, *54*(9), 1949 - 1970. doi: 10.1175/JPO-D-23-0255.1

Kelly, S. M. (2016). The vertical mode decomposition of surface and internal tides in the presence of a free surface and arbitrary topography. *Journal of Physical Oceanography*, *46*(12), 3777 - 3788. Retrieved from https://journals.ametsoc.org/view/journals/phoc/46/12/jpo-d-16-0131.1.xml doi: 10.1175/JPO-D-16-0131.1

Llewellyn Smith, S. G., & Young, W. R. (2002). Conversion of the barotropic tide. *Journal of Physical Oceanography*, *32*(5), 1554 - 1566. doi: 10.1175/1520-0485(2002)032¡1554:COTBT¿2.0.CO;2

Merrifield, M. A., & Holloway, P. E. (2002). Model estimates of m2 internal tide energetics at the hawaiian ridge. *Journal of Geophysical Research: Oceans*, *107*(C8), 5-1-5-12. doi: 10.1029/2001JC000996

Pollmann, F., & Nycander, J. (2023). Resolving the horizontal direction of internal tide generation: global application for the m2 tide's first mode. *J. Phys. Oceanogr.*, *53*(5), 1251 - 1267. doi: 10.1175/JPO-D-22-0144.1

Rainville, L., Johnston, T. M. S., Carter, G. S., Merrifield, M. A., Pinkel, R., Worcester, P. F., & Dushaw, B. D. (2010). Interference pattern and propagation of the m2 internal tide south of the hawaiian ridge. *Journal of Physical Oceanography*, *40*(2), 311 - 325. Retrieved from https://journals.ametsoc.org/view/journals/phoc/40/2/2009jpo4256.1.xml doi: 10.1175/2009JPO4256.1

Rainville, L., & Pinkel, R. (2006). Propagation of low-mode internal waves through the ocean. *Journal of Physical Oceanography*, *36*(6), 1220 - 1236. Retrieved from https://journals.ametsoc.org/view/journals/phoc/36/6/jpo2889.1.xml doi: 10.1175/JPO2889.1

Zaron, E. D. (2015). Nonstationary internal tides observed using dual-satellite altimetry. *Journal of Physical Oceanography*, *45*(9), 2239 - 2246. Retrieved from https://journals.ametsoc.org/view/journals/phoc/45/9/jpo-d-15-0020.1.xml doi: 10.1175/JPO-D-15-0020.1

Zaron, E. D., & Ray, R. D. (2018). Aliased tidal variability in mesoscale sea level anomaly maps. *Journal of Atmospheric and Oceanic Technology*, *35*(12), 2421 - 2435. Retrieved from https://journals.ametsoc.org/view/journals/atot/35/12/jtech-d-18-0089.1.xml doi: 10.1175/JTECH-D-18-0089.1

Zhao, Z. (2019). Mapping internal tides from satellite altimetry without blind directions. *Journal of Geophysical Research: Oceans*, *124*(12), 8605-8625. Retrieved from https://agupubs.onlinelibrary.wiley.com/doi/abs/10.1029/2019JC015507 doi: https://doi.org/10.1029/2019JC015507

Zhao, Z. (2021). Seasonal mode-1 m2 internal tides from satellite altimetry. *Journal of Physical Oceanography*, *51*(9), 3015 - 3035. Retrieved from https://journals.ametsoc.org/view/journals/phoc/51/9/JPO-D-21-0001.1.xml doi: 10.1175/JPO-D-21-0001.1

Zhao, Z. (2022). Satellite estimates of mode-1 m2 internal tides using nonrepeat altimetry missions. *Journal of Physical Oceanography*, *52*(12), 3065 - 3076. Retrieved from https://journals.ametsoc.org/view/journals/phoc/52/12/JPO-D-21-0287.1.xml doi: 10.1175/JPO-D-21-0287.1

Zhao, Z. (2023). Mode-1 $n_2$ internal tides observed by satellite altimetry. *Ocean Science*, *19*(4), 1067–1082. Retrieved from `https://os.copernicus.org/articles/19/1067/2023/` doi: 10.5194/os-19-1067-2023

Zhao, Z., Alford, M. H., Girton, J. B., Rainville, L., & Simmons, H. L. (2016). Global observations of open-ocean mode-1 m2 internal tides. *Journal of Physical Oceanography*, *46*(6), 1657 - 1684. Retrieved from `https://journals.ametsoc.org/view/journals/phoc/46/6/jpo-d-15-0105.1.xml` doi: 10.1175/JPO-D-15-0105.1

Zhao, Z., Wang, J., Menemenlis, D., Fu, L.-L., Chen, S., & Qiu, B. (2019). Decomposition of the multimodal multidirectional m2 internal tide field. *Journal of Atmospheric and Oceanic Technology*, *36*(6), 1157 - 1173. Retrieved from `https://journals.ametsoc.org/view/journals/atot/36/6/jtech-d-19-0022.1.xml` doi: 10.1175/JTECH-D-19-0022.1

---

## Author Comment (AC1)

**1      General comments**

The article and data set describe a model representing the local sea surface height (SSH) signature of the spatio-temporally coherent internal tidal field in terms of a superposition of propagating plane waves (for given tidal constituent and baroclinic mode). Most notably, this new model (ZHAO30yr) incorporates an unprecedented collection of tidal constituents and vertical modes, mapped at higher resolution than in previous work by the author, and includes error estimates. The performance of the model is evaluated using independent altimetry data (using a variance reduction statistics) showing significant improvement over previous work by the author. These major advances were made possible mainly by the refinement of the authors' mapping technique over the years, and the inclusion of input data from more altimetry missions. In other words, ZHAO30yr is the culmination of the author's leading research in mapping internal tides sea surface signature from altimetry data. Yet, at the same time, ZHAO30yr can be seen as a new starting point for the many potential improvements suggested in this study.

I appreciate your time for the quick and thorough review.

I agree that "ZHAO30yr is the culmination of the author's leading research in mapping internal tides sea surface signature from altimetry data ... ZHAO30yr can be seen as a new starting point for the many potential improvements suggested in this study."

One important feature of the article is the description of the spatial variability in a single component of the decomposed multiwave field. The author investigates two decompositions of the multiwave field: (i) by extracting the (locally) five largest propagating plane waves, and (ii) by selecting the largest of the previously extracted waves propagating in a given directional range (generally half circles). While I am convinced that the reconstructed multiwave field is a reasonable representation of the underlying internal tidal SSH field (although all of the five fitted waves might not be significant everywhere), I have more trouble understanding the analysis of individual plane wave components as performed in the present study.

This manuscript was carefully revised following your suggestions and comments. Please see my point-by-point response blow.

**(A)** The article claims that such decompositions can be used to disentangle wave interference and further isolate internal tidal beams. This is an apparent contradiction, in that these beams are commonly viewed as an interference pattern resulting from the superposition of waves radiating from distinct sources (Rainville et al., 2010, see in particular their Fig. 9). The confusion is entertained by the claim that individual components of the multiwave decomposed SSH are not affected by multiwave interference (l215 in the article). On the contrary, interference patterns (with nodes and antinodes separated by half a wavelength) are clearly visible in the decomposed product (see Figs. 3 and 5 in Detailed comments). The above apparent inconsistencies make it challenging for me to understand what the spatial variability in a single component of the multiwave decomposed field actually represents (be it the locally largest component or the locally largest one in a directional range). In any event, I think the article lacks a clear definition of what is considered as "internal tidal

beams". Then, the results based on a single component of the decomposed multiwave field (sections 5 and 6) should be analyzed in a way that is consistent with the latter definition.

Rainville et al (2010) did not say that "… beams are commonly viewed as an interference pattern resulting from the superposition of waves radiating from distinct sources." On the contrary, Rainville et al (2010) said that "… direct observation of these propagating waves is complicated … by the presence of interference patterns," and suggested that "… waves from multiple sources and their interference pattern have to be taken into account to correctly interpret in situ observations and satellite altimetry." Their Figure 9 is shown below. In panels (a) and (b), one can see misleading "apparent" beams (green lines) in the multiwave interference pattern. But they are not real internal tidal beams, which are specifically shown in panels (c)–(f). Therefore, "apparent" beams caused by multiwave interference are not the internal tidal beams I mapped in this study. ZHAO30yr (and its predecessors) is developed to extract real internal tidal beams by plane wave analysis and multiwave decomposition.

[Figure]

Figure 9 in Rainville et al. (2010). Baroclinic SSH for mode 1 from (a) the model and from (b) the superposition of four line sources on the ridge at (c) Kauai and (d) Kaulakahi Channels and at the islands of (e) Nihoa and (f) Hawaii. Green lines in (b) mark misleading "apparent" beams, but no real internal tidal beams. Zhao30yr is developed to extract real internal tidal beams as shown in (c)–(f).

In ZHAO30yr, I decompose the multiwave internal tide field as shown in panel (b) into 5 waves in different directions at each grid point. Then I decompose internal tidal waves by propagation direction. The decomposed components show isolated beams and provide useful information on their generation, propagation, and dissipation. The directionally decomposed results as shown in Figures 7–18 reveal numerous real internal tidal beams that were previously masked by multiwave interference.

The Referee found half-wavelength wiggles in Figure 3 below in this document. But Figure 3 shows the first plane wave in my model (labelled *IW1*), which is a bad way of examining internal tidal beams. As I show in the study, one should examine the directionally decomposed components (labelled *southmax*, *northmax*, *eastmax*, or *westmax*) following Figures 7–18. For comparison, I replotted Figure 3 (labelled Figure X3) using the directionally decomposed components. Figure X3 clearly shows southward and northward beams and much weaker half-wavelength wiggles. Figure X3 suggests that ZHAO30yr has separated southward and northward internal tides. ZHAO30yr can reduce, if not fully remove, the effect of multiwave interference.

Figure 5 tells another story. As shown in Figure 20 in my paper, the along-beam SSH amplitudes range 2–4 mm for mode-1 $K_1$ and 1–2 mm for mode-2 $K_1$. They are *very weak* signals and easily affected by model errors (i.e., 0.5±0.3 mm). Note that my fitting window is only 120 km by 120 km. I can better map $K_1$ internal tides using larger fitting windows (e.g., 250 km by 250 km), considering that their wavelengths are longer than 250 km in the Arabian Sea. Despite the weak signal, my Figure 20 clearly shows long-range $K_1$ beams and linear-increasing phases. With this said, I admit that ZHAO30yr is currently imperfect, due to the inhomogeneity of satellite ground tracks and small fitting windows used in this study.

(B) Another comment is that the general tone of the article is quite enthusiastic (perhaps too much so), somewhat obscuring the limitations of the model. As a result, non-specialist users of the data might end up being mislead. I would like to see the important limitations and shortcomings of the model explicitly stated where fit:

ZHAO30yr has inherit limitations, mainly stemming from the large sampling intervals in both time and space of nadir-looking altimetry data. In the revised manuscript, I added Section 9 to specifically point out shortcomings of the new model including 30-year coherent component only and un-optimized mapping parameters.

- (C) ZHAO30yr only extracts the spatio-temporally coherent fraction of the ITs. Hence, the decay in amplitude of an observed internal tidal beam with distance from its source region cannot be directly related to dissipation (e.g. Zaron, 2015; Buijsman et al., 2017; Geoffroy & Nycander, 2022).

  Due to the missing incoherent component, ZHAO30yr cannot fully answer the question of where internal tides dissipate. But the along-beam decay provides useful information on their dissipation.

- (D) ZHAO30yr cannot distinguish between modal ITs originating from barotropic-to-baroclinic conversion and the waves resulting from scattering of different modes (e.g. by interactions with topography).

This question is interesting. ZHAO30yr was not designed to answer this question.

- **(E)** Physically, an internal tidal beam is the spatial expression of the interference of multiple waves radiating from distinct sources (Rainville et al., 2010). Thus, one cannot directly "pinpoint" the source locations of an observed beam. Moreover, it is not straightforward to gain information on the generation of the individual baroclinic waves forming a beam.

  Please see my reply to comment (A), where I explain the difference between "apparent" and real internal tidal beams. ZHAO30yr (and its predecessors) was developed to resolve multiwave interference and real internal tidal beams.

  My goal is to decompose and extract real internal tidal beams. For example, Figure 19 (and Figure X4 in this document) shows well-defined beams on the Amazon continental shelf and at the Mid-Atlantic Ridge. Their generation sites can be clearly seen. Throughout the revised manuscript, "pinpoint" was replaced with "locate."

**2    ESSD's review criteria**

- *Read the manuscript: are the data and methods presented new? Is there any potential of the data being useful in the future? Are methods and materials described in sufficient detail? Are any references/citations to other data sets or articles missing or inappropriate? Is the article itself appropriate to support the publication of a data set?*

The methods used in this publication have been used in previous studies (e.g. Zhao et al., 2016, 2019; Zhao, 2021, 2022, 2023, with an evolution throughout the years). I fail to see a significant difference in the methods compared with Zhao (2023), apart from the inclusion of more tidal constituents and vertical modes. Another important advance is the use of more altimetry missions as input data to build the model. The data will very likely be useful in the future. The methods are described in details, and more information can be found in the previously mentioned studies. However, I have troubles understanding a few key points in the methods and presentation of the results (see General comments). These may be misunderstandings, or simple issues with the terminology used in the paper. Nonetheless, I think these points should be clarified before publication. The paper generally does a good job citing appropriate references and other datasets. It could, however, do a better job at pointing at the (important) limitations of the model.

This manuscript was carefully revised following your suggestions and comments. Some confusing points were clarified. Please see my point-by-point response.

I understand the Referee "fails to see a significant difference in the methods compared with Zhao (2023)." It is because my goal is not to present another new method but develop a new internal tide model (ZHAO30yr) by applying this recently improved method to global satellite altimetry data.

- *Check the data quality: is the data set accessible via the given identifier? Is the data set complete? Are error estimates and sources of errors given (and discussed in the article)? Are the accuracy, calibration, processing, etc. state of the art? Are common standards used for comparison? Is the data set significant – unique, useful, and complete?*

(F) The data is straightforward to access and use. The data set lacks error estimates for a single plane wave, as well as the mask for regions of large mesoscale variability where the model likely fails (as explained in the article). In particular, each of the fitted plane wave should be compared with the local error estimate for a single wave; as it is likely that not all of the five fitted waves are significant at any given location. The model, i.e. the sum of the five fitted plane waves for a given tidal constituent and baroclinic mode, is validated using variance reduction statistics, and it is compared to previous iterations showing significant improvement.

I uploaded the data (model errors and geographic mask) and made them freely available to the public (doi:10.6084/m9.figshare.28559978.v1). They are (1) the geographic mask for regions of strong mesoscale activities (and thus large model errors) and (2) the 5-wave decomposed errors including 4 files for mode-1 and -2 $M_2$ and mode-1 and -2 $K_1$.

- *Consider article and data set: are there any inconsistencies within these, implausible*

*assertions or data, or noticeable problems which would suggest the data are erroneous (or worse). If possible, apply tests (e.g. statistics). Unusual formats or other circumstances which impede such tests in your discipline may raise suspicion. Is the data set itself of high quality?*

As stated above, I think some key points in the methods and presentation of the results should be clarified. Apart from these, the article properly describes the data set.

In the revised manuscript, many points were clarified following your suggestions and comments. Please see my point-by-point response.

- *Check the presentation quality: is the data set usable in its current format and size? Are the formal metadata appropriate? Check the publication: is the length of the article appropriate? Is the overall structure of the article well structured and clear? Is the language consistent and precise? Are mathematical formulae, symbols, abbreviations, and units correctly defined and used? Are figures and tables correct and of high quality? Is the data set publication, as submitted, of high quality?*

The published data set is usable (but still lacks the single wave error estimates and mask for the regions of large mesoscale variability), and the metadata is appropriate. The publication is somewhat long but well written and, overall, of good quality.

Please also see my reply to comment (F).

- *Finally: By reading the article and downloading the data set, would you be able to understand and (re-)use the data set in the future?*

Yes, but at this stage only for the reconstructed multiwave field (i.e. the sum of the locally fitted plane waves for a given tidal constituent and baroclinic mode). I don't understand the author's interpretation of the spatial variability in a single wave component.

The model is at regular spatial grid, it is straightforward to check the data and plot some figures (see Figures 1–5 below in this document).

New users may be confused with single-wave components or internal tidal beams, because these variables are not provided in previous internal tide models. Users need to write their own codes to do detailed analysis. I am more than happy to help would they ask any specific questions.

**3    Detailed comments**

**Introduction**

l25-27: It would be fair to cite Colosi and Munk (2006) as well.

Colosi and Munk (2006) is cited now.

l32: To be complete, you could mention semi-analytical methods as well; especially the recent anisotropic estimates in Pollmann and Nycander (2023) and Geoffroy et al. (2025), which are particularly suited for a comparison with ZHAO30yr.

The semi-analytical internal tide modes are mentioned and three references cited.

Paragraph 34-45: Please make it clear that ZHAO30yr only estimates the spatio-temporally coherent ITs.

Section 9 was added to describe major limitations. One is that ZHAO30yr contains only the 30-year phase-locked internal tide components.

l51: The "new" mapping procedure is confusing. This seems to be the same procedure as in Zhao (2022).

 "New" was replaced with "recently improved." The procedure was described in Zhao (2022).

l56: Same "newly improved" is confusing.

Throughout the paper "newly improved" was replaced with "recently improved."

l57-59: This is confusing. Internal tidal beams are generally considered as the expression of the interference pattern between plane waves originating from different sources (Rainville et al., 2010).

Please see my reply to comment (A).

l60: ZHAO30yr only sees the coherent fraction of the IT field, thus any information about dissipation would be non-trivial to retrieve (the decay of a beam with propagation distance in ZHAO30yr can also be attributed to a loss of coherence). ZHAO30yr also cannot distinguish between modal internal tides (ITs) generated by the conversion of surface tides and those resulting from the scattering of different modes. This should be clearly stated.

Please see my replies to comments (C) and (D).

l62: "background internal tides" is unclear, perhaps mention that this will be explained later on.

The term "background internal tides" is described in my previous paper Zhao (2023b), which is cited here now. The method will be explained in Section 4.1.

l63: Errors are lower than 1 mm on a global average.

"On a global average" was added in several places.

l67: "published in Carrere et al. 2021" → "mentioned in Carrere et al. 2021".

Changed as suggested.

l70: "are previously masked" → "were previously masked".

Fixed as suggested.

l78: "Appendix ??". The appendix appears to be missing (it is not included in the manuscript and supplementary material.) This sentence is unclear at this stage.

This sentence was removed. There once was an Appendix but dropped in later revision.

l84: "including all altimetry [. . . ]" already written at the beginning of the paragraph.

This part "including all altimetry . . . Marine Service" was removed.

**Section 2.1**

l89: "Direct" → "Directly"

Fixed.

l95: add "leaked" internal tide signals?

Added.

l98: Taking into account the spatial variations in $\lambda$, depending on location one can have $\lambda$ larger than the cutoff wavelength for the mesoscale field. Hence, this low-pass filtering of the mesoscale estimate does not consistently remove the leaked IT variance (resulting in a biased low IT variance in the results). Applying another (lowpass) filter in frequency might mitigate this, as suggested by Zaron and Ray (2018). In this paper, they also note that the diurnal ITs are more closely entangled with mesoscale dynamics (compared with semidiurnal ITs), so that you may have more leakage from mesoscale activity in your diurnal products.

Zaron and Ray (2018) is cited here now. The mesoscale correction could be further explored to improve internal tide models in future studies.

**Section 2.2**

**(G)** l106: Can you demonstrate that these 2 constituent pairs are well separated in ZHAO30yr?

The two constituent pairs are well separated for the following reasons: (1) The two constituent pairs are evaluated using independent altimetry data (Figure 4); (2) They follow overall the same scaling factors as their barotropic tides (Figure 21). I admit that ZHAO30yr is far from perfect yet.

l113: Do you use the rigid-lid approximation, namely $\Phi\,(z = 0) = 0$? If not, the problem you solve is not trivial to me. Could you mention the boundary condition you use at the surface and how you normalize the eigenfunctions (and provide a reference)? One example is Kelly (2016).

The free-surface condition is used in the calculation following Kelly (2016).

l115: I think the equation should read $\lambda_n = \omega\, c_n\, / \sqrt{\omega^2 - f^2}$, see for instance Eq. (A5) in Zhao et al. (2016).

Double-checked. Note that its left-hand side is wavelength $\lambda_n$ not wave speed $c_p$.

l118: What about the temporal variation in $\lambda$ over seasonal, and inter-annual timescales? It would be worth comparing this time variability with the bandpass width of the 2D filter.

The seasonal and decadal variations of wavelengths do not affect my 2D bandpass filter, because my bandpass is wide (see Table 1). For example, my bandpass width for mode-1 $M_2$ is [0.75, 1.5] times local wavenumber. In contrast, the seasonal and decadal variations of wavelength are usually <10%, that is, [0.9 1.1] times local wavenumber. I have calculated and compared results with different wavelengths. Their differences are negligible.

l123: Can you justify that such close constituents are well separated in ZHAO30yr?

Please see my reply to comment (G).

**Section 3**

The method extracts 5 independent propagating waves with a target $\lambda$ and $\omega$. Signatures of other processes than ITs (sharing a similar $\lambda$ and $\omega$) might be picked up: the mesoscale correction is not perfect, and there may be other waves (e.g. westward propagating tropical instability waves as mentioned in Zhao (2019)). The main assumption here is that only ITs can show the observed spatio-temporal coherent structures over the 30 years of data. Do I get this right? (If yes, I think this could be made clearer).

No. It is not because of their spatiotemporal coherent structures. No other processes share *both* similar $\lambda$ and $\omega$ with internal tides. Eddies and Tropical instability waves have different $\lambda$ or/and $\omega$ with internal tides. Empirical internal tide models have errors, because the internal tide field is irregularly sampled in both space and time by satellite altimetry. Would we have spatially and temporally regular satellite observations, mapping internal tides is not a problem anymore!

l131: Truncating the 2D spectrum (i.e. multiplying it by a rectangular window) will result in ringing, you might want to use a window with a smoother roll-off.

The ringing effect (artificial wiggles) mainly occurs in the boundary layer (see Figure 4 in Zhao et al 2019 JTech). For each window, we throw away values in the outer 100-km boundary layer and keep only the inner region. Now Zhao et al. (2019) is cited.

**Section 3.1**

Could you mention how do you bin the altimetry data in time? What tolerance do you use to consider two measures collocated in time? What would a typical snapshot look like in terms of point density within a fitting window?

In this study, I did not bin altimetry data in time. All data from 1993 to 2022 from all altimetry missions are used. Each measurement goes to the wave fit, although it may collocate with SSH measurements made by other altimetry missions. By plane wave analysis, I do not care what a snapshot looks like.

Eq. (2): If I am not mistaking, this equation represents waves emanating from an infinitely long line source. This might be a good representation far away from the actual sources (of finite length), but not so much close to generation sites (where I would expect the waves to spread cylindrically).

Figure X1 below shows that the southward mode-1 $M_2$ internal tide spreads cylindrically. There are many such examples in the global ocean. This feature suggests that ZHAO30yr can detect cylindrical spreading (maybe underestimate amplitudes), though the waves are determined by plane wave analysis.

[Figure]

Figure X1. Southward mode-1 $M_2$ internal tides from the Lombok Strait. Colors indicate SSH amplitude (in mm). Black lines indicate the 0° and 180° co-phase contours. The beam spreads cylindrically with propagation.

See Fig 1 for an example mode-1 M2 plane wave (in a correspondingly sized fitting window) using values from Table 1 in the article. Could the extent of the spatial window (smaller than a wavelength here) negatively impact the fit?

The size and extent of the fitting window impact the wave fit. We have a trade-off here. A larger fitting window better reduces model errors and resolves horizontal propagation directions. A smaller fitting window gives better spatial resolution and larger amplitude. What is the best fitting window? It depends on many factors including the strength of internal tides, the multiwave interference, the strength of mesoscale eddies, and the proximity to land. The mapping parameters could be optimized region by region in future studies.

[Figure]

Figure 1: Mode-1 M2 plane wave as per Eq. (2) for a single m in a 120 km fitting window using the mean parameters from Table 1.

l148-154: It took me several reads to understand this procedure, this could be rewritten to make it clearer. The wording is confusing: "target internal tidal wave" is a sum of "5 internal tidal waves", perhaps the latter could be simply called plane waves. Also, you could explicitly state at what stage $A_m$, $\phi_m$, and $\theta_m$ are obtained.
l150: Is the least squares fit performed in each compass direction? If yes, it should be mentioned in the following sentence for the sake of clarity.

This paragraph was re-written. Plane wave analysis has long been documented in my previous papers dated back to Zhao et al. (2016). In my previous papers, I usually have a figure to demonstrate this procedure. To avoid repetition and help readers, one sentence was added "Examples can be found in Zhao (2014, Figure 3) and Zhao et al. (2016, Figure 2)."

Each plane wave fit should be compared with the estimated error (using "background" ITs or some other error estimate). Fits resulting in an amplitude below the estimated error should be discarded (as in Zhao et al. (2016)). Also, the error associated with a single plane wave should be included in the dataset.

For the first question, please see my reply to comment (F). For the second question, please see my reply to comment (H).

**Section 3.2**

l159: "in one overlapping 850 by 850 km" → "in overlapping 850 by 850 km windows"?

Changed as suggested.

l161: "is nontidal errors" → "is considered as noise"?

Changed as suggested.

l162: Is truncation done with a rectangular window? No. It is a square window. Does this introduce ringing? Yes. If yes, did you evaluate/mitigate it? I discard filtered results in the outer 100-km boundary layer. The truncated spectra may still contain some background noise, not only the tidal peaks. Yes. That is why the model has errors. Perhaps this background noise level could be estimated from the variance just outside the theoretical wavenumber range. No. It theoretically does not make sense and practically does not work.

**Section 3.3**

l168: "$S_2$ has an tidal aliasing" → "$S_2$ has an aliasing issue"

Changed. "Sun-synchronous altimetry missions … have an aliasing issue with the $S_2$ tide."

l172: "wavenumber-frequency filtering": I understand that you only filter in wavenumber space (but you select the frequency during the least square fit).

"Wavenumber-frequency filtering" was replaced with "the 3-step mapping procedure."

l181: Could you point to the section where these results are presented?

The results will be shown in the latter part of this paper. "(Figure 2; Sections 4 and 7)" was added in this sentence to be specific.

l184-185: I see that bandpass width times λ overlaps for mode-1 and mode-2 waves. But it is not obvious to me why mode 1 can affect mode 2 but not the other way around. Could you elaborate?
Firstly, I chose their bandpass widths (see Table 1) such that no overlap for mode-1 and mode-2 waves.

Second, the crosstalk is because the satellite altimetry data are irregular both spatially and temporally. Would the internal tide field is spatiotemporally regularly sampled by satellite altimetry, the mode-1 and mode-2 waves could be easily separated.

Third, it is because mode-1 and mode-2 waves have different amplitudes. Assuming the mode-1 and mode-2 internal tide amplitudes are 15 and 5 mm, respectively, and they may leak 10% of their amplitude to the other. One can find that mode-2 leaks to mode-1 by 0.5 mm (5 mm times 10%), which is only 3.3% of the mode-1 amplitude. However, mode-1 leaks to mode-2 by 1.5 mm (15 mm times 10%), which is 30% of the mode-2 amplitude.

**Section 3.4**

l193: define major/minor constituents.

Now this sentence reads "In the first round of plane wave analysis, a fitting window of 120 km is used for major constituents ($M_2$, $S_2$, $K_1$, and $O_1$) and 160 km for minor constituents ($N_2$, $K_2$, $P_1$, and $Q_1$)."

l200: This light blue color code for amplitudes lower than 1 mm does not reflect the spatial variability of the error and may be misleading. Can you think of a better representation including the geographical variability of the error?

There may be a misunderstanding here. My Figure 2 shows the SSH amplitude of the 12 internal tide constituents, not model errors. My Figure 3 shows errors. In Figure 3, one can see the spatial variability of model errors.

(H) I realize discarding the results where the amplitude is smaller than the error may not be a very good solution neither since a large fraction of the global field would be masked (see Fig. 2). Perhaps you could discard data based on the variance reduction statistics computed in section 4.2?

I did not discard any internal tides. My goal is to present a complete set of internal waves to avoid discarding useful information. Users can discard data by their own criteria.

Figure 2 was plotted wrong and any conclusion drawn from it makes no sense. I repeated the calculation and plotted Figure X2. Figure X2 shows that mode-1 $M_2$ internal tides are dominantly greater than model errors. Blue regions indicate where model errors are greater than signals. Figure X2 shows that one can discard weak internal tides according to model errors if needed.

[Figure]

Figure 2: Mode-1 M2 internal tide amplitude (sum of the 5 fitted plane waves), masked where the amplitude is smaller than the provided error for mode-1 semidiurnal ITs. Wrong! See Figure X2.

[Figure]

Figure X2. A corrected version of Figure 2. Blue indicates regions where model errors are greater than mode-1 $M_2$ internal tides.

l201: Consider publishing your mask excluding the regions of large mesoscale variability alongside the data.

Please see my reply to comment (F).

l202: "have largest amplitudes" → "have the largest amplitudes"

Fixed as suggested.

l203: same for "lowest"

Fixed as suggested.

**Section 3.5**

l210: "5 waves of arbitrary directions" → "arbitrary" may not be adapted here. Perhaps rewrite in something like "the 5 most prominent plane waves with empirically determined directions".

Now it reads "… 5 plane waves with empirically determined directions …"

l215: I disagree, interference patterns with half-wavelength fluctuations are clearly visible in the decomposed wave field. In Fig. 3, I attach detail maps of an area between Hawaii and the Aleutian Islands displaying a marked interference pattern in the North-South direction in the mode-1 M2 first (largest) plane wave (IW1). This can also be seen in your maps in supplementary material for mode-1 M2, N2 (and S2 to a lesser extent). Diurnal constituents seem affected as well (in particular, IW1 in mode-1 K1 west of India shows similar fluctuations in the East-West direction, see Fig. 5).

This argument is based on a bad figure. Figure 3 shows the first plane wave in my model (*IW1* in

ZHAO30yr). That is a bad way to show internal tidal beams. As demonstrated in Figures 7–18 of this study, one should examine the directionally decomposed components (*southmax*, *northmax*, *eastmax*, or *westmax* in ZHAO30yr) for internal tidal beams.

As an example, Figure X3 was plotted to present internal tidal beams in this region. Figure X3a shows a complicated interference pattern. Figures X3b and X3c show mode-1 $M_2$ internal tidal beams originating from Alaska and Hawaii, respectively. The beams show weak half-wavelength wiggles, but the beams are evidenced by their amplitudes and co-phase lines.

[Figure]

Figure 3: Left panel: Amplitude of the first plane wave for the mode-1 M2 internal tide, an interference pattern is clearly visible (horizontal darker and lighter rays with wavelength of about $\lambda/2$; Right panel: Zoom in the interference pattern. See Figure X3 for a better way to show internal tidal beams.

[Figure]

Figure X3. As in Figure 3 but for (a) the 5-wave sum, (b) southmax component, and (c) northmax component. Colors indicate the SSH amplitude (in mm). Black lines indicate the 0° and 180° co-phase contours. Panel (a) is complicated due to multiwave interference. Panels (b) and (c) clearly show internal tidal beams, although their amplitudes may have half-wavelength wiggles. But the half-wavelength wiggles in (b) and (c) are much weaker than those in (a).

l216: "new features that are previously masked" → "new features that were previously masked"

Changed as suggested.

l215-224: This is confusing. Internal tidal beams are an interference pattern, I don't understand how you could locally represent a beam as a single plane wave as defined in Eq. (2) (for a single m, see Fig. 1).

Please see my reply to comment (A).

l222: Do you mean the decomposed multiwave field?

Changed. Now this sentence reads "This study will demonstrate that isolated internal tidal beams should be examined using the directionally decomposed components (Section 5)."

l223: Yes, looking at a decomposition by propagation direction might be a more consistent approach. Still, I don't quite understand how you could assimilate the amplitude of the largest plane wave in a directional range to a beam.

Users need to write their own codes to make further analysis. It is a little complicated for new users. I am more than happy to help would users have any specific questions (email me!).

figure 2: Hard to read, panels are too small.

Replotted. Now Figure 2 has larger panels.

**Section 4.1**

l238: The corresponding mask is not included with the published data.

Please see my reply to comment (F).

l240: "reliably" might be an overstatement, especially at the global scale (as you discuss in section 4.2).

"reliably" was dropped here.

Figure 3: The statistics seem to be computed including regions of large mesoscale variability, discarding these regions (as stated in the text) will improve the results.

Yes. But excluding eddy-rich regions does not change the numbers much.

**Section 4.2**

l244: "(amplitudes and phases)" → "(amplitude and phase of each constituent)"

Changed as suggested.

l246-247: should read $T_n$, $f_n(t)$, $u_n(t)$, $A_n(x, y)$, and $phi_n(x, y)$.

Fixed as suggested. Thanks!

l249: delete "Here" or → "Here,"

"Here" was deleted.

l254: "is the variance difference" → "is the difference in variance computed"

Changed as suggested.

l256-257: confusing, rephrase.

One sentence was changed to "Variance reductions for the 12 internal tide constituents are computed respectively following the same procedure."

l262: "real internal tides" → "predicted internal tides"

Changed. Now it reads "predicted real internal tides".

**Section 4.3**

figure 4: Hard to read, panels are too small.

Figure 4 was replotted to make panels larger.

l292: internal tidal beams are presents in all products, rephrase.

"Detect internal tidal beams" was replaced with "better resolve internal tidal beams."

**Section 5.1**

l306-307: as well as in studies based on in situ observations, it would be fair to mention a few of these as well.

Numerical models, semi-analytical models, and in situ measurements are mentioned here.

l310: "But" → "Note"

Changed as suggested.

l312-313: These single-wave error estimates are not published with the data.

Please see my reply to comment (F).

l316-318: unclear, rephrase.

In the revised manuscript, I dropped "Nevertheless, low model errors are key to track their along-beam amplitude and phase changes (Zhao, 2016)."

l325: "Section 6.1 has shown" → "Section 6.1 shows"

Changed as suggested.

Figure 8: delete "[too many circles]" in caption

Fixed.

l329: This is unclear. The detection of mode-2 M2 ITs with altimetry might be strongly influenced by the stratification structure, but not so much the underlying true wave field (other parameters set the generation, e.g. the topographic wavelength (Llewellyn Smith & Young, 2002)). You should also mention that mode-2 waves tend to be more incoherent.

One sentence was added "Mode-2 $M_2$ internal tides tend to become more incoherent and undetectable, because they are easily affected by the time-varying ocean environment." Another sentence was added "Mode-2 $M_2$ internal tides are mainly associated with rough bottom topography, because they are generated in the tide-topography interaction."

l342: "The mode-1 and mode-2 beams" → "The detected mode-1 and mode-2 beams". Item (1), and perhaps item (2), can be partly explained by mode-2 waves being more incoherent than mode-1 waves (e.g. Rainville & Pinkel, 2006). Also, a possible explanation for item (2) could be that the interference of the shorter mode-2 wavelengths results in narrower beams compared with mode 1.

Changed as follows: "The mode-1 and mode-2 beams" was replaced with "The satellite observed mode-1 and mode-2 beams". Also, one sentence was added "It is partly because mode-2 waves become more incoherent than mode-1 waves after leaving their generation sources (Rainville and Pinkel, 2006)."

l346: item (3), linear theory is instructive here: it predicts very small conversion where the topographic length scale does not match the wavelength of the wave (Llewellyn Smith & Young, 2002). In other words, surface tide conversion into mode 2 is maximized at topographic features with length scales close to the shorter mode-2 wavelength.

I agree. The reference is cited now.

l347: Again this is confusing, the beams are not generated at a point location, they are the result of the interference of multiple (line) sources.

Yes, I can locate the generation sites of mode-2 internal tidal beams (see Figure 19). Please also see my reply to comment (A).

l349: This is wrong, mid-ocean ridges also generate significant mode-2 M2 ITs. See global map of the M2 mode-2 generation from linear theory in Geoffroy et al. (2024).

Mid-ocean ridges do not generate mode-2 $M_2$ internal tides continuously like linear sources. Instead, mode-2 $M_2$ internal tides are generated at isolated hotspots over the ridges (see Figure X4 or Figure S15 below). Thus, I say that they are mainly generated over isolated seamounts.

l350-351: The model also has important limitations that should be clearly mentioned (only coherent ITs, interference pattern present in the amplitude of the individual fitted plane waves, separation of barotropic conversion and scattering processes).

Please see my replies to comments (B) and (C).

**Section 5.2**

l361: "; however," → ", however,"

Fixed as suggested.

l365: Could (1) be explained by weaker S2 barotropic tide as in (2)?

Now it reads "likely because the $S_2$ barotropic and internal tides are weak, and the latter are easily masked by model errors."

**Section 5.3**

l382: "summed filed" → "summed field"

Typo was fixed.

l382: "beams are masked by multiwave interference", rewrite.

Changed. Now it reads "For the same reason, mode-1 $N_2$ beams can be seen in the 5-wave summed field (Figure 11a), but they are smeared by multiwave interference."

**Section 5.5**

l410-411: eastern/western hemisphere relative to 120 deg W? (State reference longitude to avoid confusion.)

The division between eastern and western hemisphere is unclear and may cause confusion. Therefore, I replaced them using the Pacific and Indian Oceans and the Atlantic Ocean. Now it reads "… is their geographic distribution: All diurnal constituents are strong in the Indian and Pacific Oceans and weak in the Atlantic Ocean."

l422: "Figure 13b, circle", which circle?

l423: "Figure 13b, circle", which circles? (Use different colors to distinguish from the Mona Passage.)

The three circles in Figure 13b were plotted in cyan, magenta and blue, respectively. Corresponding changes were made.

l435: "pinpoint" is confusing.

"Pinpoint" was replaced with "locate."

l437: Unclear, rewrite.

Now it reads "This feature raises the question of what special topographic and tidal conditions combined induce these isolated beams. Answering this question may improve our understanding of the generation of internal tides and their variation with global ocean changes."

l438: Already stated above, delete sentence (and typo in "K2").

Changed as suggested.

Fig. 14, 15, 16: mention blue circles in caption?

All were changed as suggested. For Figure 14, it reads "Blue circles mark two isolated beams in the eastern Pacific Ocean."

**Section 5.6**

l441: "but that the K1" delete "that"?

Fixed as suggested.

l446: "Another similar feature" add "with mode-1 K1".

Changed as suggested.

**Section 6.1**

l494: The beams are actually clearly visible from the summed plane wave product (see Fig. 4). The directional decomposition offers a cleaner view of the beams.

Now it reads "Compared to the multiwave summed products, the decomposed products present a much clearer view of the isolated internal tidal beams off the Amazon shelf."

Figure 19: Title of (h), typo in "mode-2 S2".

Typo was fixed.

Last line of caption, "flurier 2D filtering" → "Fourier 2D filtering".

Typo was fixed.

Last line of caption: On the contrary, this last observation seems consistent with the literature: The maximum in SSH occurring at some distance away from the generation site(s) has been observed and modeled in previous studies. This has been related to the first surface bouncing of the tidal beam (Merrifield & Holloway, 2002; Carter et al., 2008). This could also be interpreted as the expression of interference between multiple plane waves (Rainville et al., 2010)).

Thank you for pointing out these reasonable explanations. This feature has puzzled me for a long time. I will conduct a thorough investigation in the future. Now it reads "Maximum SSH amplitudes along all beams do not appear on the Amazon shelf, but about one wavelength away from their source. It is likely an artificial feature caused by the large spatial windows used in plane wave fitting and Fourier bandpass filtering. It is also likely because the distances are needed for internal tidal rays bounce to the sea surface for the first time (Merrifield and Holloway, 2002; Carter et al., 2008)."

[Figure]

Figure 4: Mode-1 M2 internal tide amplitude (sum of the 5 fitted plane waves) close the Amazon mouth.

l501: Could you explain? Is this because of the superposition with waves (reflecting or originating) from the Mid-Atlantic Ridge?

The half-wavelength wiggles are mainly because of the superposition with waves originating (not reflecting) from the Mid-Atlantic Ridge. Please see my Figure S15 in the supplementary file (Figure X4 below). It shows no half-wavelength variability in the directionally decomposed components. Isolated internal tidal beams at the Mid-Atlantic Ridge can be clearly seen (blue crosses).

[Figure]

Figure X4. Semidiurnal internal tidal beams off the Amazon shelf. Each constituent is divided into northeastward (−45°–135°) and southwestward (135°–315°) components. Black lines indicate the 0° co-phase charts. Green contours indicate the 1000-, 2000-, and 3000-m isobaths. Isolated beams off the Amazon shelf are labeled A–F. Blue lines highlight the strongest beams generated at the mouth of the Amazon River. Beams generated at the Mid-Atlantic Ridge are marked. (Figure S15 in the supplementary file).

l505: It would be good to remind that the decay in amplitude of a spotted beam with propagation distance is partly due to the growing fraction of incoherent ITs (not being taken into account in your model).

Yea. Please see my reply to comment (C).

l514: Since you mention possible comparisons, the semi-analytical estimates from Pollmann and Nycander (2023), and Geoffroy et al. (2025) are also particularly suited for a comparison with ZHAO30yr.

Changed. Now it reads "It is interesting to compare the satellite observations, numerical models, semi-analytical, and in situ measurements in future research (Nycander 2025; Pollmann and Nycander 2023; Zaron and Elipot, 2024; Geoffroy et al. 2025)."

l515: "temporal variation of semidiurnal ITs", unclear. do you refer to the beat originating from the sum of the constituents close to the M2 frequency?

Slightly changed. Now this sentence reads "Their superposition will lead to the temporal variability of semidiurnal internal tides (strong beats and weak beats)."

**Section 6.2**

l528: Typo "the the four diurnal". Also, you rather show the two first modes of the two diurnal constituents O1 and K1.

Now it reads "Figure 20 shows the lowest two modes of the diurnal $O_1$ and $K_1$ constituents."

l533-535: This beam is actually solely due to IW1 in the plane wave decomposed field (see Fig. 5). Furthermore, an interference pattern with half-wavelength fluctuations is clearly visible within the area of the beam. Hence, Fig. 20 a is challenging to interpret. Nonetheless, the evolution of phase and amplitude along the putative trajectory look convincing.

[Figure]

Figure 5: Left panel: Amplitude of the first plane wave for the mode-1 K1 internal tide (IW1), an interference pattern is clearly visible (both vertical and horizontal darker and lighter rays with wavelength of about $\lambda/2$; Right panel: IW1 minus the westward component of the directionally decomposed internal tidal field.

See Figure 20 in my paper, the along-beam SSH amplitudes range 2–4 mm for mode-1 $K_1$ and 1–2 mm for mode-2 $K_1$. They are very weak SSH signals and easily affected by model errors (0.5±0.3 mm). ZHAO30yr is imperfect, due mainly to the combined effect of the small fitting window employed and the inhomogeneity of satellite ground tracks.

l535: "repeat cycles", unclear. Do you refer to a satellite mission in particular?

"Repeat cycles" was replaced with "repeat diurnal tidal cycles."

l537: "beams [...] composed of narrow beams" is unclear, rephrase.

The first "beams" was replaced with "internal tides". Now this sentence reads "The mode-2 internal tides from the Chagos–Laccadive Ridge appear to be composed of several isolated narrow beams (Figure 20, blue circles)."

l541: That's the 2 first modes of 2 constituents.

Slightly changed. Now it reads "for the first two modes of the $K_1$ and $O_1$ constituents."

l545: "pinpoint", again, not exactly.

"pinpoint" was dropped here.

**Summary**

l549: delete "new", it has been published already.

Changed as suggested.

l554: "down to lower than 1mm" add "on a global average".

Changed as suggested.

l559: "the internal tidal field into" → "the internal tidal field is decomposed into"

Fixed as suggested.

l560: I would be more nuanced on the results of the multiwave decomposition (specifically for the presence of interference patterns in a single plane wave field).

Please see my reply to comment (A).

l564: reference for ZHAO20yr?

Two references were added: Zhao et al (2016) and Carrere et al (2021).

l569: To be nuanced: the beam generation does not occur at a single source point (Rainville et al., 2010). Moerover, incoherence acts to decrease the detected beams' amplitude with propagation distance, this is not straightforward to disentangle from dissipation.

Please see my replies to comments (A) and (C).

l571: "off the Amazon shelf recognized" → "off the Amazon shelf have been recognized"

Fixed as suggested.

l573: "For mode-2 constituents" → "For M2 and S2 mode-2 internal tides"

Changed as suggested.

l575-577: The mode-1 K1 beam you identified west of India corresponds to the first (largest) fitted plane waves in this region. This is inconsistent with the commonly accepted view of a beam resulting from the interference between multiple plane waves emanating from distinct generations sites.

Please see my reply to comment (A).

l579: The beams have already been identified in previous studies. Also, you already stated this in point 4.

Changed. Now it reads "We have revealed isolated internal tidal beams for all 12 internal tide constituents."

**References**

Buijsman, M. C., Arbic, B. K., Richman, J. G., Shriver, J. F., Wallcraft, A. J., & Zamudio, L. (2017). Semidiurnal internal tide incoherence in the equatorial pacific. Journal of Geophysical Research: Oceans, 122 (7), 5286-5305. Retrieved from https://agupubs.onlinelibrary.wiley.com /doi/abs/10.1002/2016JC012590 doi: https://doi.org/10.1002/2016JC012590 [cited]

Carter, G. S., Merrifield, M. A., Becker, J. M., Katsumata, K., Gregg, M. C., Luther, D. S., . . . Firing, Y. L. (2008). Energetics of m2 barotropic-to-baroclinic tidal conversion at the Hawaiian islands. Journal of Physical Oceanography, 38 (10), 2205-2223. Retrieved from https://journals. ametsoc.org/view/journals/phoc/38/10/2008jpo3860.1.xml doi: 10.1175/2008JPO3860.1 [cited]

Colosi, J. A., & Munk, W. (2006). Tales of the venerable honolulu tide gauge. Journal of Physical Oceanography, 36 (6), 967– 996. Retrieved from https://journals.ametsoc.org/view/journals /phoc/36/6/jpo2876.1.xml doi:10.1175/JPO2876.1 [cited]

Geoffroy, G., Kelly, S. M., & Nycander, J. (2025). Tidal conversion into vertical normal modes by continental margins. Geophysical Research Letters, 52 (1), e2024GL112865. Retrieved from https://agupubs.onlinelibrary.wiley.com/doi/abs/10.1029/2024GL112865 (e2024GL112865 2024GL112865) doi: https://doi.org/10.1029/2024GL112865 [cited]

Geoffroy, G., & Nycander, J. (2022). Global mapping of the nonstationary semidiurnal internal tide using argo data. Journal of Geophysical Research: Oceans, 127 (4), e2021JC018283. doi: 10.1029/2021JC018283 [cited]

Geoffroy, G., Pollmann, F., & Nycander, J. (2024). Tidal conversion into vertical normal modes by near-critical topography. Journal of Physical Oceanography, 54 (9), 1949-1970. doi: 10.1175/JPO-D-23-0255.1 [cited]

Kelly, S. M. (2016). The vertical mode decomposition of surface and internal tides in the presence of a free surface and arbitrary topography. Journal of Physical Oceanography, 46 (12), 3777-3788. Retrieved from https://journals.ametsoc.org/view/journals/phoc/46/12/jpo-d-16-0131.1.xml doi: 10.1175/JPO-D-16-0131.1 [cited]

Llewellyn Smith, S. G., & Young, W. R. (2002). Conversion of the barotropic tide. Journal of Physical Oceanography, 32 (5), 1554-1566. doi: 10.1175/1520-0485(2002)032¡1554:COTBT¿2.0.CO;2 [cited]

Merrifield, M. A., & Holloway, P. E. (2002). Model estimates of m2 internal tide energetics at the hawaiian ridge. Journal of Geophysical Research: Oceans, 107 (C8), 5-1-5-12. doi: 10.1029/2001JC000996 [cited]

Pollmann, F., & Nycander, J. (2023). Resolving the horizontal direction of internal tide generation: global application for the m2 tide's first mode. J. Phys. Oceanogr., 53 (5), 1251-1267. doi: 10.1175/JPO-D-22-0144.1 [cited]

Rainville, L., Johnston, T. M. S., Carter, G. S., Merrifield, M. A., Pinkel, R., Worcester, P. F., & Dushaw, B. D. (2010). Interference pattern and propagation of the m2 internal tide south of the hawaiian ridge. Journal of Physical Oceanography, 40 (2), 311 - 325. Retrieved from https://journals.amet soc.org/view/journals/phoc/40/2/2009jpo4256.1.xml doi: 10.1175/2009JPO4256.1 [cited]

Rainville, L., & Pinkel, R. (2006). Propagation of low-mode internal waves through the ocean. Journal of Physical Oceanography, 36 (6), 1220-1236. Retrieved from https://journals.ametsoc.org /view/journals/phoc/36/6/jpo2889.1.xml doi: 10.1175/JPO2889.1 [cited]

Zaron, E. D. (2015). Nonstationary internal tides observed using dual-satellite altimetry. Journal of Physical Oceanography, 45 (9), 2239-2246. Retrieved from https://journals.ametsoc.org /view/journals/phoc/45/9/jpo-d-15-0020.1.xml doi: 10.1175/JPO-D-15-0020.1 [cited]

Zaron, E. D., & Ray, R. D. (2018). Aliased tidal variability in mesoscale sea level anomaly

maps. Journal of Atmospheric and Oceanic Technology, 35(12), 2421 - 2435. Retrieved from https://journals.ametsoc.org/view/journals/atot/35/12/jtech-d-18-0089.1.xml doi: 10.1175/JTECH-D-18-0089.1 [cited]

Zhao, Z.(2019). Mapping internal tides from satellite altimetry without blind directions. Journal of Geophysical Research: Oceans, 124(12), 8605-8625. Retrieved from https://agupubs.onlinelibrary. wiley.com/doi/abs/10.1029/2019JC015507 doi: https://doi.org/10.1029/2019JC015507 [cited]

Zhao, Z. (2021). Seasonal mode-1 m2 internal tides from satellite altimetry. Journal of Physical Oceanography, 51(9), 3015-3035. Retrieved from https://journals.ametsoc.org/view/journals/phoc /51/9/JPO-D-21-0001.1.xml doi: 10.1175/JPO-D-21-0001.1 [cited]

Zhao, Z. (2022). Satellite estimates of mode-1 m2 internal tides using nonrepeat altimetry missions. Journal of Physical Oceanography, 52(12), 3065-3076. Retrieved from https://journals.ametsoc. org/view/journals/phoc/52/12/JPO-D-21-0287.1.xml doi: 10.1175/JPO-D-21-0287.1 [cited]

Zhao, Z. (2023). Mode-1 $N_2$ internal tides observed by satellite altimetry. Ocean Science, 19(4), 1067–1082. Retrieved from https://os.copernicus.org/articles/19/1067/2023 doi: 10.5194/os-19-1067-2023 [cited]

Zhao, Z., Alford, M. H., Girton, J. B., Rainville, L., & Simmons, H. L. (2016). Global observations of open-ocean mode-1 M2 internal tides. Journal of Physical Oceanography, 46(6), 1657-1684. Retrieved from https://journals.ametsoc.org/view/journals/phoc/46/6/jpo-d-15-0105.1.xml doi: 10.1175/JPO-D-15-0105.1 [cited]

Zhao, Z., Wang, J., Menemenlis, D., Fu, L.-L., Chen, S., & Qiu, B. (2019) Decomposition of the multimodal multidirectional m2 internal tide field. Journal of Atmospheric and Oceanic Technology, 36(6), 1157-1173. Retrieved from https://journals.ametsoc.org/view/journals/atot /36/6/jtech-d-190022.1.xml doi: 10.1175/JTECH-D-19-0022.1 [cited]

---

## Author Comment (AC2)

I consider this journal focused toward documenting data sets that are useful to the earth science community. As the new 30-year internal tide model is likely to be quite useful, I am happy to see it so thoroughly documented. I have some suggestions about items that appear confused or confusing or might improve the presentation.

Thank you very much for your time and help.

1. I think it is too much to show all 12 derived constituents in large 3-panel figures. This covers Figures 7 through 18, a lot of figures and a lot of page space. Each figure is described in the text and most readers, if like me, will eventually find this tedious. Cannot some of this go into the Supplement? Are they all so important to justify so much page space? "A picture is worth a thousand words" (Line 296), but it can still be too many, unless there is good reason to show all of these.

I have seriously considered the suggestion of moving some figures to the supplementary file. I decided to keep its current arrangement for the following reasons. First, Figures 7–18 present the core product of ZHAO30yr and show new internal tide features that have never been seen before. It is better to keep the completeness. Second, page space is not as critical as before, because papers in this journal (Earth System Science Data) are published online. Both the paper and its supplementary file will be in digital.

2. In each figure there are two panels (b and c) that decompose the wave field into two directions. These are interesting to see, but I am less clear about what they mean and what Dr Zhao thinks they mean. At Line 58 it is claimed that this decomposition "reveals numerous long-range internal tidal beams." But so does the original (not decomposed) wave field. It is my understanding that the apparent beams (horizontal, not vertical!) represent mostly interference patterns (e.g., Rainville & Pinkel, 2006), and this must still be true in all three of the figure panels. I still think the decompositions are useful, because they (presumably) reflect in some way how much wave energy at a location is arriving from which direction. But the justification for these decompositions and the assertions about "beams" needs some more careful thinking.

I thank the Referee for pointing out the differences between "apparent" and real internal tidal beams. Yes. The multiwave interfered internal tide fields give "apparent" internal tidal beams. In contrast, the directionally decomposed fields give real internal tidal beams. This concept may be challenging for new users, because previous internal tide models do not give resolved internal tidal beams.

The multiwave interfered field and the directionally decomposed field seemingly have similar beam patterns. But the latter should be used in the calculation of energy, energy flux, and phase speed. My goal of Figures 7–18 is to demonstrate numerous isolated internal tidal beams previously masked.

I agree with the Referee that, in Figures 7–18, panels (b) and (c) show multiwave interference of some degree. It is partly because the multiwave decomposition is not perfect. Note that the multiwave interference can be reduced by dividing the summed field into 4 components. For example, in one of my previous papers, I have four directionally decomposed components: eastward (directional range is -45°–45°), northward (45°–135°), westward (135°–225°), and southward (225°–315°). In addition, a larger fitting window also helps reduce half-wavelength wiggles.

3. There is some apparent confusion about "sun-synchronous" measurements and how these impact the S2 waves - Lines 169-172. The statement that "signals caused by solar radiance have longer spatial scales" is not the most important point. That is not the problem with sun-synchronous measurements. The problem is that all the altimeter data measure the tide at the same phase, so you are trying to solve for a sine wave when you have measurements at only one phase. I think the reason S2 is recovered here is because there is a lot of CryoSat data. If the fitting were tried with only the sun-synch data, it would be less good and maybe fail, I suspect. Also, (Line 170), Ubelmann et al. solved for S2, but they did not discuss the results, and it is not clear how successful their S2 solution was.

This paragraph was thoroughly rewritten. "There are likely three reasons for why Sun-synchronous missions do not ruin our mapping of $S_2$ internal tides. (1) Our mapping procedure extracts internal tides not only by their frequencies in time but also by their wavelengths in space. Measurements by Sun-synchronous missions still provide useful spatial information on $S_2$ internal tides. (2) Our 30-year-long data record itself can significantly reduce model errors. (3) A large fraction of our data is from non-Sun-synchronous missions, which greatly reduces model errors."

One item that does potentially point to small S2 errors (caused by lack of phase sampling in sun-synchronous measurements) can be seen by comparing M2, N2, and S2 results - Figures 7, 9, and 11. Patterns in N2 look more like M2 than do patterns in S2, even though S2 forcing is closer to M2 forcing. For example, look at the southward components in panels (c) in the North Pacific. S2 appears different. I suspect this is because of the problem of so many altimeters that do not sample S2 well enough.

In an earlier version of my manuscript, I had one section describing scaling factors. It was dropped before submission. Challenged by this comment, I think it is better to add it back (Section 7 Scaling factors). In this section, I examined the spatial patterns of the barotropic and internal tide constituents using TPXO8 and ZHAO30yr. I found similarities between 4 pairs of internal tide constituents ($M_2$–$N_2$, $K_1$–$P_1$, $S_2$–$K_2$, and $Q_1$–$O_1$). $M_2$ and $N_2$ have similar spatial pattern with a correlation coefficient of ~0.69. $S_2$ and $K_2$ have similar spatial patterns with a correlation coefficient of ~0.57. The similarities stem from their barotropic tide similarities.

4. Related to that, I suspect that another reason Zhao20 is inferior to Zhao30 for S2 is because there was far less CryoSat data in the old solution. This affects the explanation Lines 287-291.

ZHAO20yr-$S_2$ was developed excluding SSH measurements made by Sun-synchronous altimetry missions. It is poor, because (1) it used much fewer SSH data (1993–2012, excluding Sun-synchronous altimetry missions), and (2) it was mapped using my old mapping procedure.

5. The Abstract needs to acknowledge that the mapping is only the "phase-locked" component. This is clear from the Intro, but it also should be stated in the Abstract.

One sentence was added to the Abstract that "ZHAO30yr only extracts the 30-year phase-locked internal tides, lacking the incoherent component caused by the time-varying ocean environment."

6. Since non-repeat altimeter data is used, it is important in Section 2.1 to state what Mean Sea Surface model was used. Also, Line 86, polar tide should be pole tide.

One sentence was added to specify: "Specifically, the mean sea surface model used in this satellite altimetry product is CNES-CLS15 (Pujol et al., 2018)."

It should be "pole tide." Fixed.

7. Table 1, column "Bandpass width" has units?

A note (b) was added to explain that "bandpass width multiplying local wavenumber K (lon, lat) yields bandpass cut-off wavenumbers."

8. Line 123. I think this statement should be removed unless it can be backed up with evidence.

This statement was removed. A new sentence was added that "… therefore, it is challenging to separate these two constituents. In this paper, I extract reasonable $K_1$ and $P_1$ internal tides empirically using 30 years of altimetry data."

9. Line 139: For plane-wave fitting, perhaps Ray and Cartwright (2001, GRL) should be mentioned here?

A sentence was added here. It reads "Plane wave analysis evolves from the two-dimensional plane-wave fit method (Ray and Cartwright, 2001), but plane wave analysis can determine multiple waves in different propagation directions and thus resolving multiwave interference (Zhao and Alford, 2009)."

10. Section 4.1 on model errors describes an interesting approach to determine errors, with interesting results. Also, I notice that mode-2 errors are larger than mode-1 errors, even in absolute terms, not just relative. I like this section.

Figure 2 shows that mode-1 and mode-2 errors are overall on the same level. The latter is slightly larger than the former by 0.1 mm.

11. In Figure 3 caption, please state what the numbers in upper left mean. In Figure 4 caption, please state units (yes, the figure gives mm^2, but it is tiny in upper right and easy to miss). In Figure 8 caption, what is "too many circles"?

The captions of Figures 3, 4 and 8 were changed/fixed as suggested.

12. Line 437: Answering WHAT question?

This paragraph was rewritten. One sentence was added "This feature raises the question of what special topographic and tidal conditions combined induce these isolated beams."

13. Figure 19 caption. I do not understand the sentence "Note that the largest..." Also, fix apparent typo on "flurier".

This sentence was rewritten. It reads "Note that maximum SSH amplitudes along all beams do not appear on the Amazon shelf, but about one wavelength away from their source. It is an artificial feature caused by the large spatial windows used in plane wave fitting and Fourier bandpass filtering."

Typo was fixed. It should be "Fourier."

More typos:

78 - ??

Fixed. There once was an Appendix in the manuscript but was dropped in later revision.

Figure 19h, "moe"

Typo was fixed.

504 - should "model" be "data"?

Changed. Now this sentence reads "For comparison, the $S_2$, $N_2$, and $K_2$ beams disappear sooner, likely because their lower amplitudes are masked by the still large errors in our models."

728 - "using"

Typo was fixed.

---

## Referee Report (RR1)

**2nd review on "A New-Generation Internal Tide Model Based on 30 Years of Satellite Sea Surface Height Measurements: Multiwave Decomposition and Isolated Beams"**

by Zhongxiang Zhao

April 2025

**1 General comments**

I thank Dr Zhao for his dedication in answering my numerous comments point-by-point. After reading his response alongside the tracked changes version of the manuscript, I consider he did a great job answering my earlier interrogations. I also think the revised manuscript gained in clarity and does not require another round of review. Still, there are three points I would like to raise (I let the author and editor judge the necessity of addressing these):

The first one concerns the denomination of the "apparent" and "real" or "isolated internal tidal beams", i.e. the pattern originating from the interference of multiple internal tidal waves versus a single internal tidal wave originating from a finite-length source. I am unsure referring to the latter as a "beam" is correct (note I am not a native english speaker). Instead, I think a "real internal tidal beam" should simply be called an internal tidal wave. In my opinion, this would greatly improve the clarity of the text and be more consistent with the literature.

Second, as pointed out by the author, my plot (Figure 2) and comments relative to the SSH error were wrong. In the published dataset, the SSH error is given in unit mm while the SSH amplitude is in cm (I assumed everything was in cm). The units are indeed correctly specified in the data, however it requires special attention from the user (at least for users not relying on matlab) that

could be easily spared. Thus, I suggest using the same unit for both the SSH amplitude and error in the published data.

Lastly, Dr Zhao made available the error estimates for individual waves as well as the geographic mask corresponding to regions of strong mesoscale activity (http://doi.org/10.6084/m9.figshare.28559978.v1). However, this update is not visible from the main dataset page (ttp://doi.org/10.6084/m9.figshare.28078523.v1). Could a reference be added there ? (Maybe it is pending, in that case please ignore this comment).

**2 Detailed comments**

I list below a few suggestions for minor changes in the main text (lines are referenced to the tracked changes version).

l315: change "better resolve" for "better resolves".

l352: "Mode-2 M2 internal tides are mainly associated with rough bottom topography, because they are also generated in the tide-topography interaction" drop ", because they are also generated in the tide-topography interaction" (confusing).

l353: change "Mode-2 M2 internal tides mainly occur at low latitudes" for "Mode-2 M2 internal tides are mainly detected at low latitudes". Mode-2 waves not being seen by altimetry does not mean that they are not there. The stratification structure may cause the surface signature to be very small, still the waves can have non-negligible expressions at depth.

l534: change "the distances are needed for the internal tidal rays bounce to the sea surface for the first time" to "the distances are needed for the internal tidal rays to bounce at the sea surface for the first time".

l547: "strong beats and weak beats" is unclear. In my original comment I was thinking of the beat translating into e.g. spring and neap tides.

l588-603: "Figure??". Figure 21 (I assume) not correctly referenced throughout the paragraph.

l640: same, "Figure??" instead of Figure 21.

l647: change "Incoherent internal tides can be mapped from the de-correlation of covariance" to "Incoherent internal tides can be described statistically, and the associated variance has been mapped using realistic numerical simulations, as well as satellite altimetry and in situ observations." Note a

recent pre-print by K. Shimizu improves upon the simple statistical model used in Zaron (2015) and Geoffroy and Nycander (2022) (https://egusphere.copernicus.org/preprints/2025/egusphere-2024-4192/). Also, Zaron (2017, 2022) might be worthy additions.

l652: "ZHAO30yr has significantly reduced model errors to lower than 1 mm" add "on a global average".

l653: change "the minor and mode-2 internal tide constituents" to "the minor constituents and mode-2 waves".

l654: same.

**References**

Geoffroy, G., & Nycander, J. (2022). Global mapping of the nonstationary semidiurnal internal tide using argo data. *Journal of Geophysical Research: Oceans*, *127*(4), e2021JC018283. doi: 10.1029/2021JC018283

Zaron, E. D. (2015). Nonstationary internal tides observed using dual-satellite altimetry. *Journal of Physical Oceanography*, *45*(9), 2239 - 2246. Retrieved from https://journals.ametsoc.org/view/journals/phoc/45/9/jpo-d-15-0020.1.xml doi: 10.1175/JPO-D-15-0020.1

Zaron, E. D. (2017). Mapping the nonstationary internal tide with satellite altimetry. *Journal of Geophysical Research: Oceans*, *122*(1), 539-554. Retrieved from https://agupubs.onlinelibrary.wiley.com/doi/abs/10.1002/2016JC012487 doi: https://doi.org/10.1002/2016JC012487

Zaron, E. D. (2022). Baroclinic tidal cusps from satellite altimetry. *Journal of Physical Oceanography*, *52*(12), 3123 - 3137. Retrieved from https://journals.ametsoc.org/view/journals/phoc/52/12/JPO-D-21-0155.1.xml doi: 10.1175/JPO-D-21-0155.1

---

## Author Response (AR2)

**Editor**

The manuscript has been significantly improved in the first round of discussion. Nonetheless, there is still room for some improvements, as per the referees' suggestions.

Thank you very much for your time and help!

Dear Author,

The referees have acknowledged the improvements you brought in the revised version and your dedication in addressing their comments. Nonetheless, as you can see, they provide some useful suggestions for a few further adjustments, which I kindly encourage you to consider.

This manuscript was further revised following suggestions from both referees. Please see my point-to-point response.

Concerning the comment by Referee #2 about the number of figures, which they leave to my decision. I would tend to agree on their point that the whole description could be somewhat lengthy for some readers, but on the other hand of they are important for you I am also perfectly fine with them.

In response to Referee #2's concern on the number of figures, I moved Figures 2, 5, and 6 to supplementary materials and dropped Figure 21. Among them, Figure 2 presents the 12 constituents. Figures 5 and 6 show that the new model is better than an old version. Figure 21 shows scaling factors. They are not so important as Figures 7–18 (now Figure 4–15). In the end, the revised manuscript has only 17 figures.

**Referee #1**

I thank Dr Zhao for his dedication in answering my numerous comments point-by-point. After reading his response alongside the tracked changes version of the manuscript, I consider he did a great job answering my earlier interrogations. I also think the revised manuscript gained in clarity and does not require another round of review.

Thank you very much for your time and help!

Still, there are three points I would like to raise (I let the author and editor judge the necessity of addressing these):

(A) The first one concerns the denomination of the "apparent" and "real" or "isolated internal tidal beams", i.e. the pattern originating from the interference of multiple internal tidal waves versus a single internal tidal wave originating from a finite-length source. I am unsure referring to the latter as a "beam" is correct (note I am not a native English speaker). Instead, I think a "real internal tidal beam" should simply be called an internal tidal wave. In my opinion, this would greatly improve the clarity of the text and be more consistent with the literature.

Thank you very much for the clarification. I agree that "beam" may have other denominations. The term "internal tidal beam" has long been used since our 2009 paper *New Altimetric Estimates of Mode-1 $M_2$ Internal Tides in the Central North Pacific Ocean* (https://doi.org/10.1175/2009JPO3922.1). In this paper, we suggested that multiwave interference must be resolved to correctly interpret in situ observations and satellite altimetry. Later, we used the term in our 2016 paper *Global Observations of Open-Ocean Mode-1 $M_2$ Internal Tides* (https://doi.org/10.1175/JPO-D-15-0105.1). It has been cited ~240 times (Google scholar) without public objection to our usage of the word. Since then, I used the term in all my following journal papers. I think that it is better for me to keep consistent. I do not object other researchers to use the term for different meanings, if only those researchers are consistent in their own publications.

Second, as pointed out by the author, my plot (Figure 2) and comments relative to the SSH error were wrong. In the published dataset, the SSH error is given in unit mm while the SSH amplitude is in cm (I assumed everything was in cm). The units are indeed correctly specified in the data, however it requires special attention from the user (at least for users not relying on matlab) that could be easily spared. Thus, I suggest using the same unit for both the SSH amplitude and error in the published data.

Thank you for pointing out that the inconsistency between the units of SSH amplitudes (in centimeters) and amplitude errors (in millimeters) could cause misunderstanding. In the revision, I re-created the model error files. Now, both amplitudes and errors are in centimeters (uploaded to the figshare data server).

Lastly, Dr Zhao made available the error estimates for individual waves as well as the geographic mask corresponding to regions of strong mesoscale activity (http://doi.org/10.6084/m9.figshare.28559978.v1). However, this update is not visible from the main dataset page (http://doi.org/10.6084/m9.figshare.28078523.v1). Could a reference be added there? (Maybe it is pending, in that case please ignore this comment).

I also noticed that the two datasets (models and errors) do not display side by side at my figshare website, although both Digital Object Identifiers (DOIs) work well. But I cannot manage it, which is likely programmed by the figshare server. To compensate for this inconvenience, I added the hyperlink of the model errors (http://doi.org/10.6084/m9.figshare.28559978.v3) or the citation (Zhao 2025) in three places of the revised manuscript. A serious reader will not miss these data files.

**Referee #2**

Overall, the paper is improved, and I appreciate the additions that point out limitations with the new model.

Thank you very much for your time and help!

I still have a few comments, the first is in response to a new section (Section 7) on "scaling factors". As the Author Reply notes, this section was originally in his draft paper but then removed in the first submitted version. It has now been added back. I agree with his original thinking, and I question its utility because I don't think it adds much that is useful.

Section 7 was dropped in the revised manuscript. It was used to explain why the $M_2$–$N_2$ pair is more similar than the $M_2$–$S_2$ pair to address a question brought up in the first-round discussion.

First, the scaling factors are strangely in the form of rational fractions: (1/3), (1/5), etc. The barotropic Q1-O1 factor is quoted as 1/5. This makes no geophysical sense. The Q1/O1 ratio in the astronomical force is 0.1926 = 1/5.192. It is then stated that the internal tides have the same factors, but they are not listed. I question this. E.g, the O1-Q1 histogram (Fig 21h) shows a clear bias toward larger Q1 values (likely from noise), and any kind of orthogonal regression of that data, which accounts for errors in both variables, would give a scale factor different than 1/5. Same for the histograms of P1-K1 and K2-S2. The end result is that these factors supposedly imply that "the new internal tide model is trustworthy," but I don't follow the logic of that. My recommendation is that the section be taken back out.

Section 7 shows that the eight internal tide constituents have scaling factors that are consistent with their barotropic counterparts. I rounded the scaling factors to their closest integer ratios for simplicity. In fact, as Referee #2 points out, the Q1/O1 ratio is 1/5.192 not 1/5. I agree with Referee #2 that this section can be taken out (and left to future dedicated study).

My original review stated that Figures 7-18 were too many figures and most readers would find the description of all of them "tedious". I still think that. Dr Zhao disagrees and thinks all the figures are necessary. I leave it to the editor to decide. Dr Zhao is correct that the journal is no longer cutting down trees to print all these figures, so that is good.

I respect your opinion that this manuscript is somewhat long. In the revised manuscript, I moved Figures 2, 5, and 6 to supplementary materials to make room for Figures 7–18 (now Figures 4–15). Now the revised manuscript has 17 figures (instead of 21 figures).

My final point is in response to his replies to both reviewers regarding interference of wave patterns and the formation of internal tide "beams". It dawns on me that the reviewers and Dr Zhao are using the term "beam" with different meanings. We were using the term to describe a concentrated 2-D focus of wave energy. I think Dr Zhao is using "beam" to describe essentially a wavefront, so he talks about beams even for plane waves (or waves spreading out in a cylindrical pattern). Maybe our objections about "beams" and "interference" would make more sense if this difference in meaning is accounted for.

Thank you for the clarification. Please see my reply to comment (A) of Referee #1. In this manuscript, all my figures (e.g., Figures 16 and 17) clearly illustrate what I mean by an "internal tidal beam", which is consistent with my previous papers.